# Structural basis for high selectivity of a rice silicon channel Lsi1

Yasunori Saitoh [1,7], Namiki Mitani-Ueno[2,7], Keisuke Saito [3,4,7], Kengo Matsuki [5], Sheng Huang[2], Lingli Yang [1], Naoki Yamaji[2], Hiroshi Ishikita [3,4], Jian-Ren Shen [1,5], Jian Feng Ma [2✉] & Michihiro Suga [1,5,6✉]

Silicon (Si), the most abundant mineral element in the earth's crust, is taken up by plant roots in the form of silicic acid through Low silicon rice 1 (Lsi1). Lsi1 belongs to the Nodulin 26-like intrinsic protein subfamily in aquaporin and shows high selectivity for silicic acid. To uncover the structural basis for this high selectivity, here we show the crystal structure of the rice Lsi1 at a resolution of 1.8 Å. The structure reveals transmembrane helical orientations different from other aquaporins, characterized by a unique, widely opened, and hydrophilic selectivity filter (SF) composed of five residues. Our structural, functional, and theoretical investigations provide a solid structural basis for the Si uptake mechanism in plants, which will contribute to secure and sustainable rice production by manipulating Lsi1 selectivity for different metalloids.

[1] Research Institute for Interdisciplinary Science, Okayama University, Tsushima Naka 3-1-1, Kita, Okayama 700-8530, Japan. [2] Institute of Plant Science and Resources, Okayama University, Chuo 2-20-1, Kurashiki 710-0046, Japan. [3] Research Center for Advanced Science and Technology, The University of Tokyo, 4-6-1 Komaba, Meguro-Ku, Tokyo 153-8904, Japan. [4] Department of Applied Chemistry, Graduate School of Engineering, The University of Tokyo, 7-3-1 Hongo, Bunkyo-Ku, Tokyo 113-8654, Japan. [5] Graduate School of Natural Science and Technology, Okayama University, 3-1-1 Tsushima Naka, Okayama 700-8530, Japan. [6] Japan Science and Technology Agency, PRESTO, 4-1-8 Honcho, Kawaguchi, Saitama 332-0012, Japan. [7] These authors contributed equally: Yasunori Saitoh, Namiki Mitani-Ueno, Keisuke Saito. ✉email: maj@rib.okayama-u.ac.jp; michisuga@okayama-u.ac.jp

All plants rooting in soil contain a significant amount of silicon (Si) in their bodies[1–3]. Although Si has not been recognized as an essential element for plant growth, its beneficial effects have been observed in many plant species. Si is especially essential for the high and stable production of rice (*Oryza sativa*), which is able to accumulate Si in the shoots to up to 10% Si of dry weight[3]. This high Si accumulation helps plants to overcome various biotic (e.g., pest, disease) and abiotic (e.g., lodging, nutrient imbalance, metal toxicity) stresses[1–4]. Due to its importance in rice production, Si has been recognized as an "agronomically essential element", in Japan and Si fertilizers are routinely applied to the paddy field[3].

Plant roots take up Si from soil solution as silicic acid $Si(OH)_4$, an uncharged monomeric molecule at a pH below 9. High accumulation of Si in rice shoots is achieved by the cooperation of two different transporters for silicic acid; Low Si rice 1 and 2 (Lsi1 and Lsi2)[5–7]. Lsi1 and Lsi2 are polarly localized at the distal and proximal sides of both exodermis and endodermis of the roots, respectively, and are responsible for Si uptake[5–7]. Knockout of either *Lsi1* or *Lsi2* results in a significant decrease in Si uptake and rice grain yield[5–7]. Lsi1 belongs to the Nodulin 26-like intrinsic proteins (NIPs) subfamily in the aquaporin (AQP) family[6], while Lsi2 belongs to the ion transporter superfamily[7]. NIPs are unique members of AQP because they are only present in plants but not in animals. Furthermore, among NIPs, only a small group (NIP III), including Lsi1, shows transport activity for silicic acid[8–10], while other members transport boric acid, arsenite, and glycerol[11,12] (Fig. 1a and Supplementary Table 1). These findings indicate that Lsi1 has a distinct selectivity for silicic acid, but the structural basis for this high selectivity is unknown.

In the present study, we show the crystal structure of rice Lsi1 at 1.8 Å resolution and compare it with that of water-specific AQP1[13] and glycerol permeable aquaglyceroporin GlpF[14], and other known AQP structures. The structure of Lsi1 reveals unique transmembrane (TM) helical orientations, the selectivity filter (SF), and water molecules in the channel that are distinct from the other structurally characterized AQPs. Mutational studies based on the high-resolution structure and theoretical calculations uncover the principles of silicic acid permeability.

## Results

**The overall structure of Lsi1.** To obtain the crystal of Lsi1, we used fluorescence-detection size-exclusion chromatography (FSEC)[15]. However, we failed to obtain crystals of full-length Lsi1, so we screened a large variety of mutant Lsi1 and found that a mutant starting at Leu47 and ending at Arg264 (ΔN44/ΔC24/K50R/C66A/T93V/C139A/K232R/T253V/K264R) gave rise to crystals (Supplementary Figs. 1 and 2). This Lsi1 variant, Lsi1$_{cryst}$, is functional in transporting Ge (Si analog) based on assay in Sf9 insect cells (Supplementary Fig. 3). Compared to the C-terminally EGFP-tagged full-length Lsi1 (CE-Lsi1), both CE-Lsi1$_{cryst}$ and the N- and C- terminally truncated constructs (CE-Lsi1_ΔNC) showed a slightly reduced Ge transport activity (about 60% of wild type) (Supplementary Fig. 3). Crystals of Lsi1$_{cryst}$ diffracted to 1.6 Å resolution but suffered from lattice-translocation defects[16]. We corrected X-ray intensities with the approach of Wang et al.[16] and solved the structure of Lsi1$_{cryst}$ at 1.8 Å resolution (Table 1). The structure of Lsi1$_{cryst}$ reveals a tetrameric fold similar to other AQPs from bacteria[14], plants[17,18], and animals[13,19]. Each monomer contains six transmembrane helices (TM1-TM6), five connecting loops (loop A- loop E), and two half helices (HB and HE) with N and C-terminus located on the cytoplasmic side of the membrane (Fig. 1b, c). The high-resolution electron density map allowed us to build all side-chain residues (Ala46 through Arg264) unambiguously. About 120

water molecules per monomer were also identified in the high-resolution structure (Figs. 1d and 2e). The channel pore exists in each monomer with a constriction on the extracellular side, similar to the other AQP structures[13,14,17].

However, there are notable differences between water-specific AQPs and aquaglyceroporins in the loop regions and the tilting of TM helices. Compared to other AQP structures, including AQP1[13] (PDB 1J4N) and GlpF[14] (PDB 1FX8), Lsi1$_{cryst}$ has a shift in TM1, TM4, TM5, and HE at the extracellular side towards the center of the channel, whereas its TM2 shifts towards a pseudo-*c*4 axis of the tetramer (Fig. 2a, b and Supplementary Fig. 4). A few residues unique to the Si permeable AQPs (Gly88$_{TM2}$, Val173, Thr206, Ser207$_{TM5}$, and Gly216$_{LE1}$) can explain such shifts well. AQP1 has bulky residues Phe58$_{TM2}$ and His182$_{TM5}$ in SF, whereas they are replaced by smaller ones in Lsi1$_{cryst}$, Gly88$_{TM2}$, and Ser207$_{TM5}$ (Fig. 3a, b). Two bulky residues in SF of GlpF, Trp48$_{TM2}$, and Phe200$_{LE1}$, are also replaced by Gly residues in Lsi1$_{cryst}$, Gly88$_{TM2}$, and Gly216$_{LE1}$ (Fig. 3a, c). The smaller residues in Lsi1$_{cryst}$ (Gly88$_{TM2}$, Ser207$_{TM5}$, and Gly216$_{LE1}$) alleviate steric restriction and allow the shifts of the TM1, TM2, TM5, and HE. In addition, extracellular loop A of Lsi1$_{cryst}$ (Gly73 to Ser80) is shorter than other AQPs, which may be related to the close approach of TM1 and TM2. AQP1 has Gly147 in TM4 and Gly181 in TM5, making a close contact between TM4 and TM5 possible, whereas the equivalent residues in Lsi1$_{cryst}$ are Val173 and Thr206, which disable the close approach in the center of the bilayer region. Instead, loop C interacts with the tips of TM4 and TM5, assisting

**Table 1 Crystallographic data collection and refinement statistics.**

| Data collection | |
|---|---|
| Wavelength (Å) | 1.112181 |
| Space group | $P12_11$ |
| Cell dimensions | $a = 89.5$ Å, $b = 91.4$ Å, $c = 166.1$ Å, $\beta = 102.1°$ |
| Resolution (Å) | 40−1.80 (1.91−1.80) |
| No. of unique reflections | 237,514 (37,334) |
| Completeness (%) | 96.9 (94.7) |
| *R*-factor (%) | 12.0 (104.5) |
| Multiplicity | 3.2 (3.0) |
| CC1/2 | 99.6 (78.4) |
| Mean I/σ (I) | 5.73 (1.02) |
| Refinement | |
| Resolution (Å) | 20−1.80 (1.86−1.80) |
| $R_{work}/R_{free}$ (%) | 0.2475 (0.4022)/0.2758 (0.4179) |
| Willson B-factor (Å$^2$) | 23.62 |
| No. of non-H atoms | 14,182 |
| Macromolecules | 12,920 |
| Ligands | 278 |
| Water | 984 |
| Protein residues | 1,731 |
| Average B-factor (Å$^2$) | 28.6 |
| Macromolecules | 27.0 |
| Ligands | 63.3 |
| Water | 39.8 |
| *RMSDs* | |
| Bond length (Å) | 0.017 |
| Bond angles (°) | 1.57 |
| *Ramachandran plot (%)* | |
| Favored | 95.8 |
| Allowed | 4.2 |
| Disallowed | 0.0 |
| PDB code | 7CJS |

Values in parenthesis are those of the highest resolution shell.

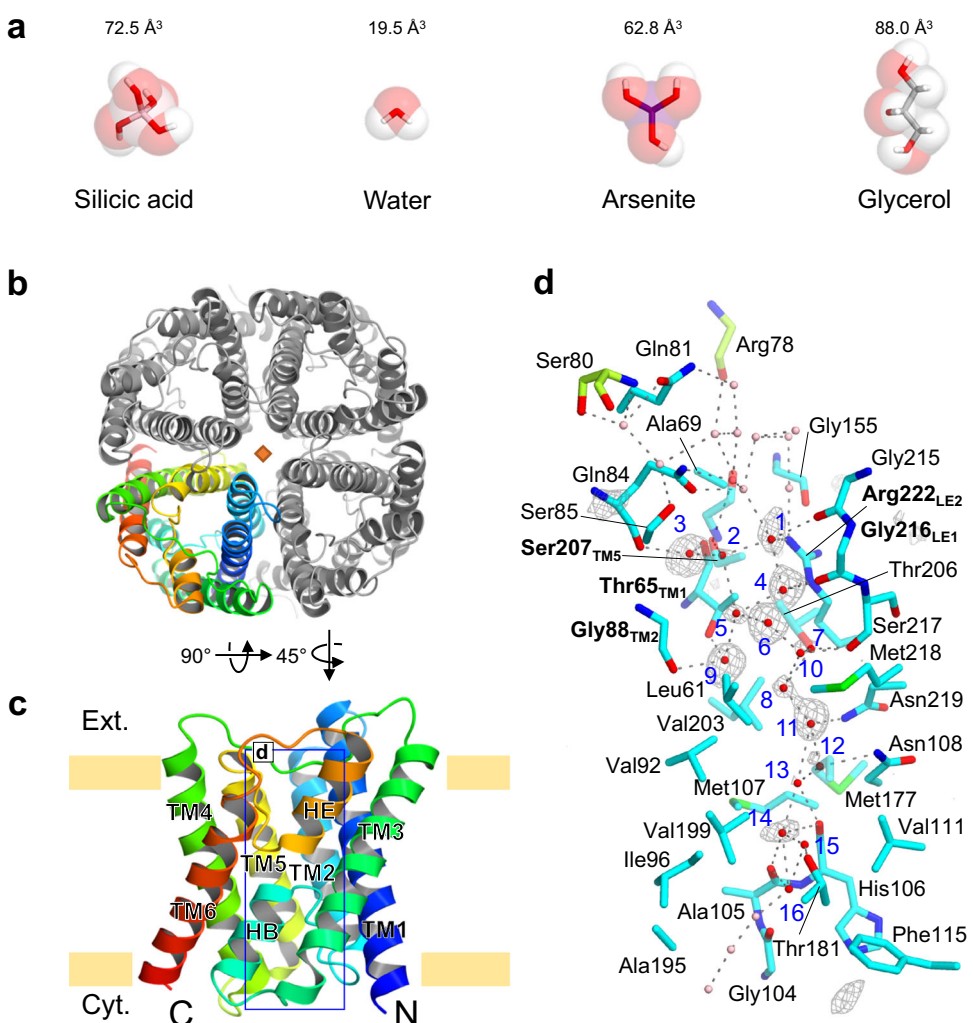

**Fig. 1 Overall structure of the rice Si channel Lsi1 at a resolution of 1.8 Å. a** Structures and the van der Waals volumes of water, glycerol, and metalloid (Si, As) transported by aquaporin. **b** The overall structure of the Lsi1$_{cryst}$ tetramer is viewed from the extracellular side. A pseudo-*C*4 symmetric axis is represented by a diamond shape. **c** Side view of the monomeric Lsi1$_{cryst}$, rainbow-colored with the N terminus in blue. The khaki bars indicate the membrane boundaries. **d** A close-up view of the channel region (chain C). Hydrogen bonding network (dot lines) shows interactions of the water molecules and the residues facing the channel's pore. Key residues comprising the ar/R SF are highlighted by bold letters. Two residues (Arg78 and Ser80) in green color are from an adjacent subunit. Water molecules positioned in the channel (red) and extra/intracellular regions (pink) are shown. Water molecules in the channel are labeled in blue, and with the omit map (gray) contoured at +3.5σ.

their close approach at the extracellular side (Supplementary Fig. 5b). Also, one water molecule found in the center of bilayer regions stabilizes the arrangement of TM4 and TM5 by hydrogen bond interactions with Asn176, Thr206, and carbonyls of Val173 and Ala202 (Supplementary Fig. 5a). This water also stabilizes a conformation of Met218 in loop E that creates the hydrophobic face near the Asn-Pro-Ala (NPA) motif. Thus, Lsi1$_{cryst}$ has different TM bundle orientations in which TM1 and HE tilt largely towards the center of the channel compared to the other structurally characterized AQPs[13,14,17–19] (Supplementary Fig. 4). These results reveal a structure of Lsi1$_{cryst}$ that is distinct from other AQPs, presumably associated with the high selectivity of Lsi1 for silicic acid since the orientation of TM helices links to the orientation of residues facing the channel.

**The channel and the selectivity filter**. The transport substrate specificity of AQPs is proposed to be determined by the aromatic/ Arg (ar/R) SF[20], which is located below the extracellular vestibule and creates the narrowest portion of the channel. SF comprises four residues in TM2, TM5, and loop E (LE1 and LE2). SF of animal, bacterial, and archaeal AQPs are classified into two types, the water-specific AQP type[13] (Phe$_{TM2}$/His$_{TM5}$/Cys$_{LE1}$/Arg$_{LE2}$) or the glycerol-permeable aquaglyceroporin type[14] (Trp$_{TM2}$/Gly$_{TM5}$/ Phe$_{LE1}$/Arg$_{LE2}$), both of which contain two bulky amino acid residues in the TM2/TM5/LE1 portion (Fig. 3). However, SF of Lsi1 contains three small residues (Gly$_{TM2}$/Ser$_{TM5}$/Gly$_{LE1}$/ Arg$_{LE2}$)[10]. Based on the crystal structure, the conduction pore is ~30 Å long and constricts to ~3.5 Å at its narrowest region (Fig. 2c, d). Each monomer has 33 water molecules in the extracellular vestibule, 25 water molecules in the intracellular vestibule, and 16 water molecules in the channel (Fig. 1d and Supplementary Fig. 6c). The extracellular vestibule is hydrophilic and binds more water molecules, whereas the intracellular side is hydrophobic and bounds fewer water molecules. The contrasting environments may affect the energetic barrier to pass SF in removing hydrated water molecules or promoting hydration and releasing the transported silicic acid. This hypothesis is supported by molecular dynamics (MD) simulations of silicic acid desolvation as shown in Supplementary Fig. 6a, b). The 16 water molecules (Wat1-Wat16) in the channel were independently

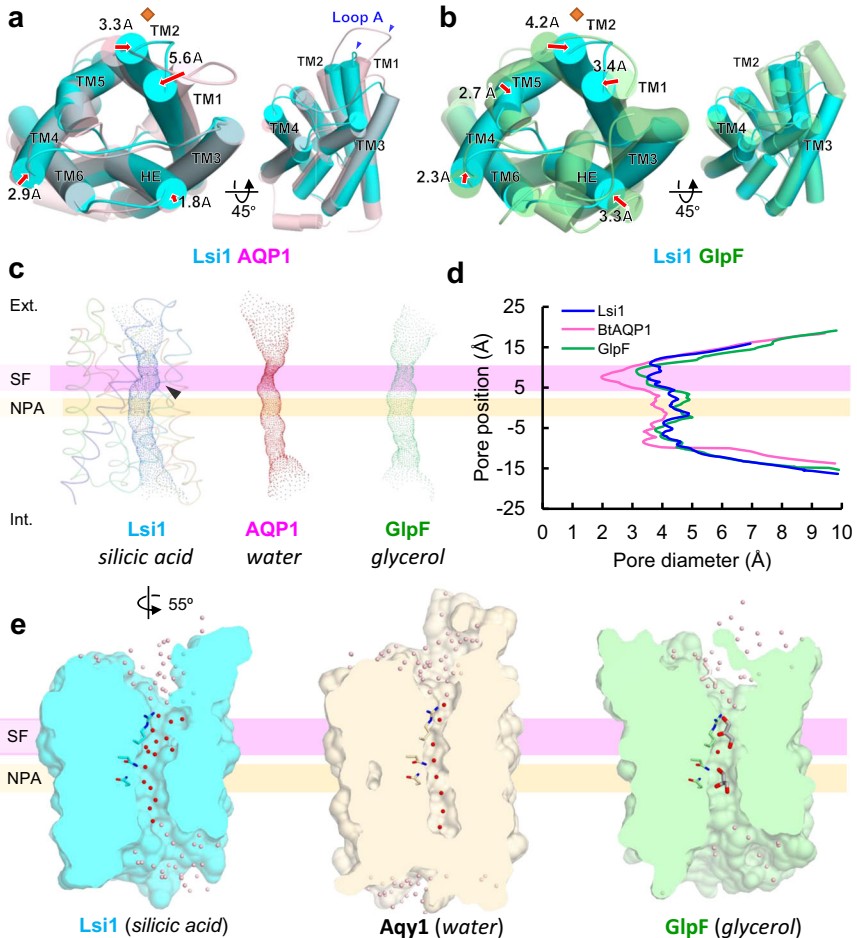

**Fig. 2 Comparison of the structures of Si channel Lsi1 and other AQPs.** Superposition of the structure of Lsi1cryst (cyan) with AQP1 (pink, PDB 1J4N) (**a**), and GlpF (light green, PDB 1FX8) (**b**). Top view from the extracellular side (left) and a rotated view by 45° against the membrane normal (right). Arrowheads indicate loop A and red arrows represent the shift of TM helices (in Å). A diamond shape represents a pseudo-*C4* symmetric axis. Channel profile (**c**) and diameters (**d**) along the pore for Lsi1cryst, AQP1, and GlpF, calculated using the program HOLE2, are shown. The regions for SF and NPA motifs are colored in plum and khaki, respectively. An arrowhead indicates the distortion in the channel. **e** Cross-section of the channel of Lsi1cryst, Aqy1 (wheat, PDB 3ZOJ), and GlpF with a 55° rotation relative to (**c**). Water molecules in the channel (red), extra/intracellular regions (pink), and glycerol molecules are shown. In **e**, N-terminal and C-terminals of Aqy1 were omitted for clarity. The color coding shown here is used for all Figures unless otherwise noted.

identified in two non-crystallographic symmetry-related Lsi1cryst tetramers, indicating that they are intrinsically associated with the monomer. Wat7 and Wat8 had relatively higher temperature factors, and their positions were too closely spaced (2.3 Å) to be simultaneously occupied, suggesting that a single water molecule occupies in rapid equilibrium between adjacent sites. Similarly, several pairs of two adjacent water molecules, Wat6-Wat10, Wat11-Wat12, Wat12-Wat13, Wat13-Wat14, and Wat14-Wat15, may be occupied by a single water molecule. By contrast, Wat3 and Wat9 have typical hydrogen bonds interactions shorter than 2.8 Å with the low-temperature factors, suggesting their stable binding to the channel.

A striking feature of the channel of Lsi1cryst is that a large number of water molecules at the extracellular half of the pore are not in a single file, which is brought by the hydrophilic SF in the broad channel (Figs. 1d and 2e). In chain A, three water molecules in SF (Wat4, Wat5, and Wat6) and two water molecules beneath it (Wat7 and Wat8) are separated by more than 3.4 Å along the channel, which no longer contributes to the hydrogen bond interactions. In addition, since Wat9 donates hydrogen bonds to the carbonyls, it cannot transfer protons to water molecules Wat5 and Wat8. It has been proposed that

strongly correlated movements of the well-oriented single-file water in orthodox AQP family proteins prevent proton transfer via the grotthuss mechanism[21]. Lsi1 must have a different mechanism that prevents the fast proton transport since the water molecules in SF of Lsi1 are not single-file. The breakage of the connectivity between the SF waters and nearby water molecules may prevent proton translocation. The water molecules at the NPA motifs and the intracellular half, by contrast, are in a single file like the other AQP structures (Figs. 1d and 2e). The NPA motifs stabilize each other creating a positive electrostatic potential that functions as a barrier against proton transport across the membrane[21–23]. Therefore, the NPA motifs and water molecules nearby function to prevent protons in the rice Si channel like the other AQPs[21–23].

Unlike AQP1 and GlpF, the channel of Lsi1cryst has a broader pore diameter at the constricted region with a slight distortion (An arrowhead in Fig. 2c). The shift of HE provides this channel's distortion, which could prevent the transport of substrates larger than silicic acid, such as silicic acid oligomer, as they are unlikely to rotate freely in this distortion. The narrowest region of the Lsi1 structure is located at SF, as observed in other AQPs. Two carbonyls of Gly215 and Gly216LE1 create the constrictions that

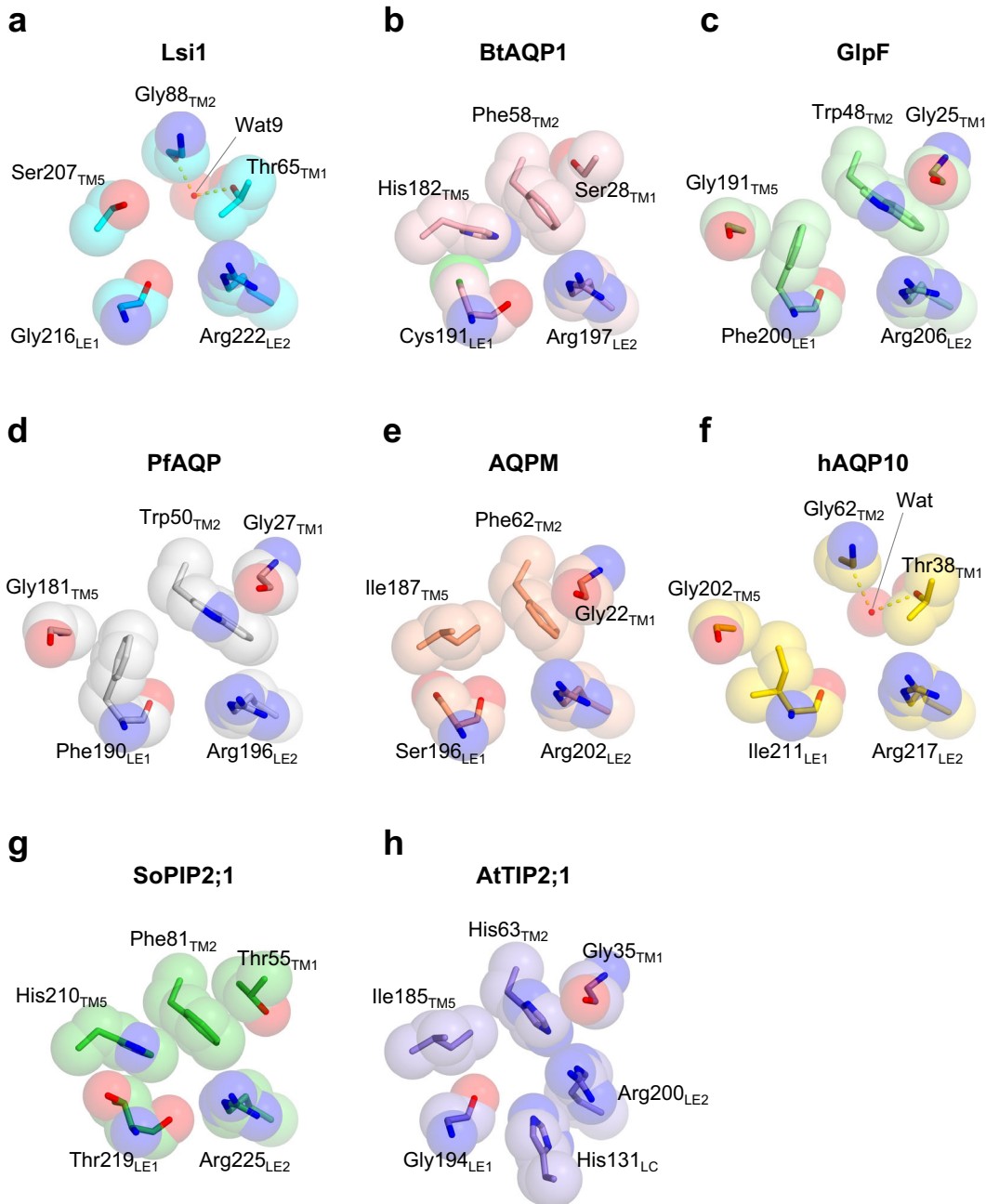

**Fig. 3 Comparison of SFs of Lsi1 and other AQPs.** SF of Lsi1cryst (**a**, cyan, PDB 7CJS), BtAQP1 (**b**, pink, PDB 1J4N), GlpF (**c**, light green, PDB 1FX8), PfAQP (**d**, white, PDB 3C02), AQPM (**e**, khaki, PDB 2F2B), hAQP10 (**f**, yellow, PDB 6F7H), SoPIP2;1 (**g**, green, PDB 1Z98), and AtTIP2;1 (**h**, light purple PDB 5I32). The view directions are the same as Fig. 2a.

point towards the channel. SF of AQP1 is narrower and hydrophilic (Phe58$_{TM2}$/His182$_{TM5}$/Cys191$_{LE1}$/Arg197$_{LE2}$), and that of GlpF is wider and amphipathic (Trp48$_{TM2}$/Gly191$_{TM5}$/Phe200$_{LE1}$/Arg206$_{LE2}$), whereas that of Lsi1cryst is the widest and hydrophilic (Thr65$_{TM1}$/Gly88$_{TM2}$/Ser207$_{TM5}$/Gly216$_{LE1}$/Arg222$_{LE2}$) (Fig. 3). The most striking feature of the Lsi1cryst's SF arises from an additional "fifth residue" Thr65 in TM1 (Thr65$_{TM1}$) and a water molecule hydrogen-bonded to Thr65$_{TM1}$ (Wat9) (Fig. 3a). The significance of Thr65$_{TM1}$ in SF has not been recognized since, in canonical AQPs, a bulky hydrophobic residue in TM2 (Phe58$_{TM2}$ in AQP1 and Trp48$_{TM2}$ in GlpF, respectively) shields the residue equivalent to Thr65$_{TM1}$. In Lsi1cryst, however, Gly88$_{TM2}$ exposes Thr65$_{TM1}$ to the channel, thereby making SF wide and hydrophilic. Thr65$_{TM1}$ donates a strong hydrogen bond

to the carbonyl of Leu61 and acts as a suitable hydrogen acceptor from a nearby water molecule Wat9 (Fig. 4a). Wat9 donates hydrogen bonds to both Thr65$_{TM1}$ and carbonyl of Gly88$_{TM2}$, thereby pointing its oxygen atom towards the channel (Fig. 4). Similarly, Wat3 can donate hydrogen bonds to both carbonyls of Gln84 and Thr65$_{TM1}$, thereby directing its oxygen atom towards the channel as well (Fig. 4). Wat3 and Wat9 are located on one side of the channel and are separated by 6 Å, facing oppositely to carbonyls of Gly215 and Ser217 (Figs. 1d and 4). Therefore, unlike other AQPs, the channel of Lsi1cryst has two polar faces. The additional polar face's water molecules likely act as hydrogen bond acceptors during the substrate transport in a similar fashion to the consecutive carbonyls protruding the channel. This notion is supported by the lower temperature factors of Wat3 and Wat9.

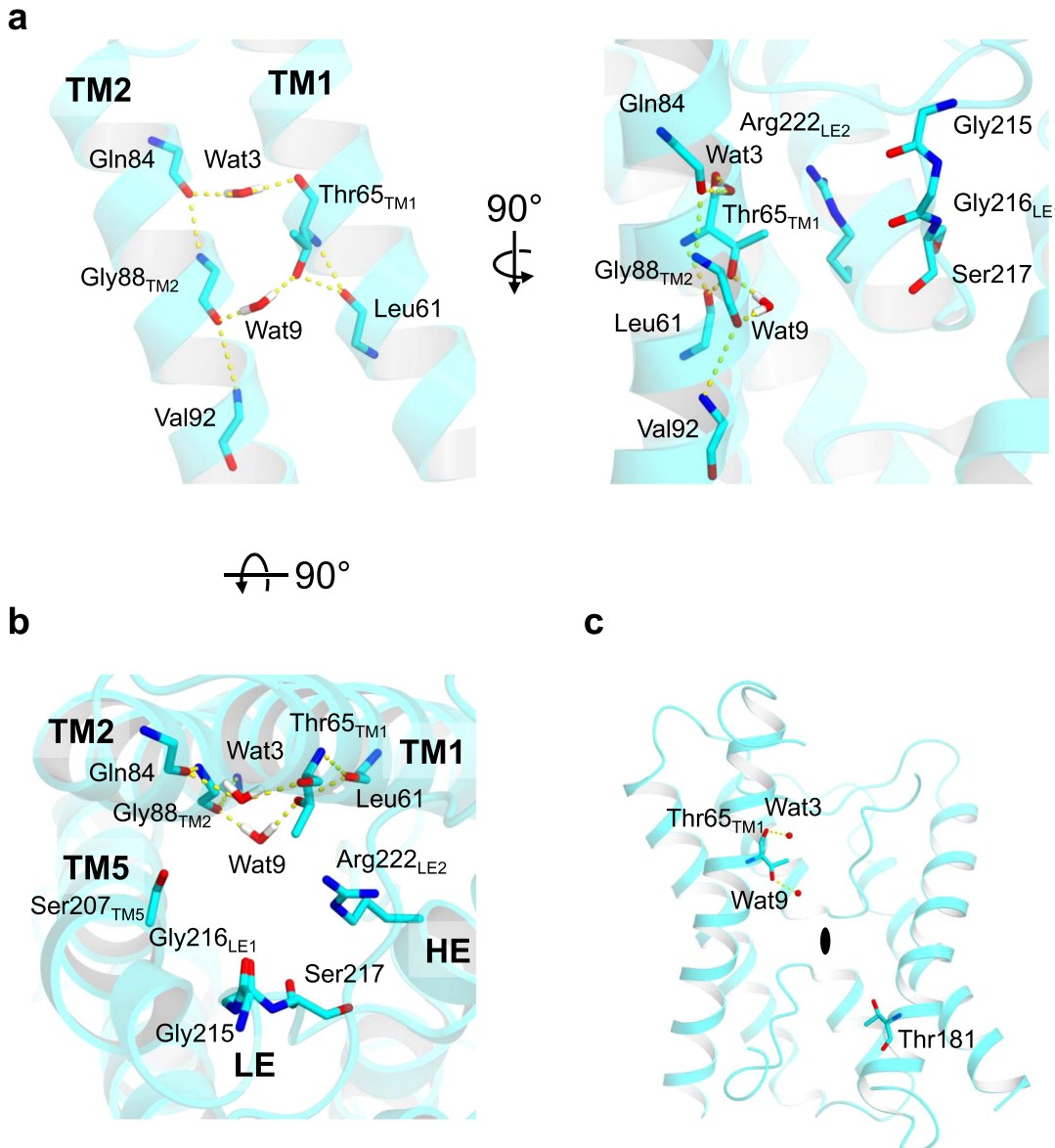

**Fig. 4 Unique water molecules and two Thr residues in the Lsi1 structure.** Hydrogen bonding interactions between the two water molecules comprising SF (Wat3 and Wat9) and TMs (TM1 and TM2) are shown with view directions facing the channel (**a**) or from the extracellular side (**b**). **c** A panel is showing the positions of Thr65 and Thr181. Hydrogen bond interactions that make the oxygen atoms of the waters pointing towards the channel are shown in yellow dashes.

Interestingly, the channel of Lsi1$_{cryst}$ has Thr181 in a pseudo-$c2$-symmetrically related position (N- and C- terminal three TM repeats) corresponding to the Thr65$_{TM1}$. Thr181 makes the channel hydrophilic likewise by having hydrogen bonds with the carbonyl of Met177 and Wat15 (Fig. 1d). These two Thr residues (Thr65$_{TM1}$ and Thr181) highlight the unique characteristic of the Si-channel Lsi1.

To examine the role of the Thr65$_{TM1}$ in transport substrate specificity, we generated ten variants of Lsi1. We investigated their transport activity for germanic acid (Ge) as a Si analog and arsenite (As) in *Xenopus* oocytes. The transport activities for both Ge and As were unaffected by the substitutions of Thr65$_{TM1}$ to Ala, Gly, and Ser (Fig. 5a, b), although expression of the Ala mutant was lower than wild type or the other two mutants in *Xenopus* oocytes (Supplementary Fig. 7a, b). One possible interpretation is that water molecules occupying free space created by the substitutions can compensate for a polar environment made by hydroxyl of Thr65$_{TM1}$ and Wat9. In

contrast, the activity was substantially decreased or lost by the other substitutions (Fig. 5a, b). By T65V$_{TM1}$ substitution, which increases the hydrophobicity of SF but keeps its size unchanged, transport activities for Ge and As were decreased. On the other hand, the T65I$_{TM1}$ substitution that mimics the size of Thr residue plus Wat9 severely decreased the activity for Ge, whereas the activity for As was similar to the T65V$_{TM1}$ mutant. These results suggested that Thr65$_{TM1}$, together with Wat9, constitute SF and play a key role in the specificity of transport substrate.

We also examined the role of the Thr181 in the substrate specificity by generating five site-directed mutants (T181S, T181V, T181N, T181Q, and T181D) and determined transport activity of Ge and As likewise in Sf9 cells. While the substitutions of Thr181 to Ser, Val, Asn, and Asp unaffected or slightly decreased the Ge transport activity, they slightly increased activity for As transport (Fig. 5c, d and Supplementary Fig. 7c, d). T181Q substitution, which narrows the size of the channel, completely abolished the Ge transport activity, whereas the As transport

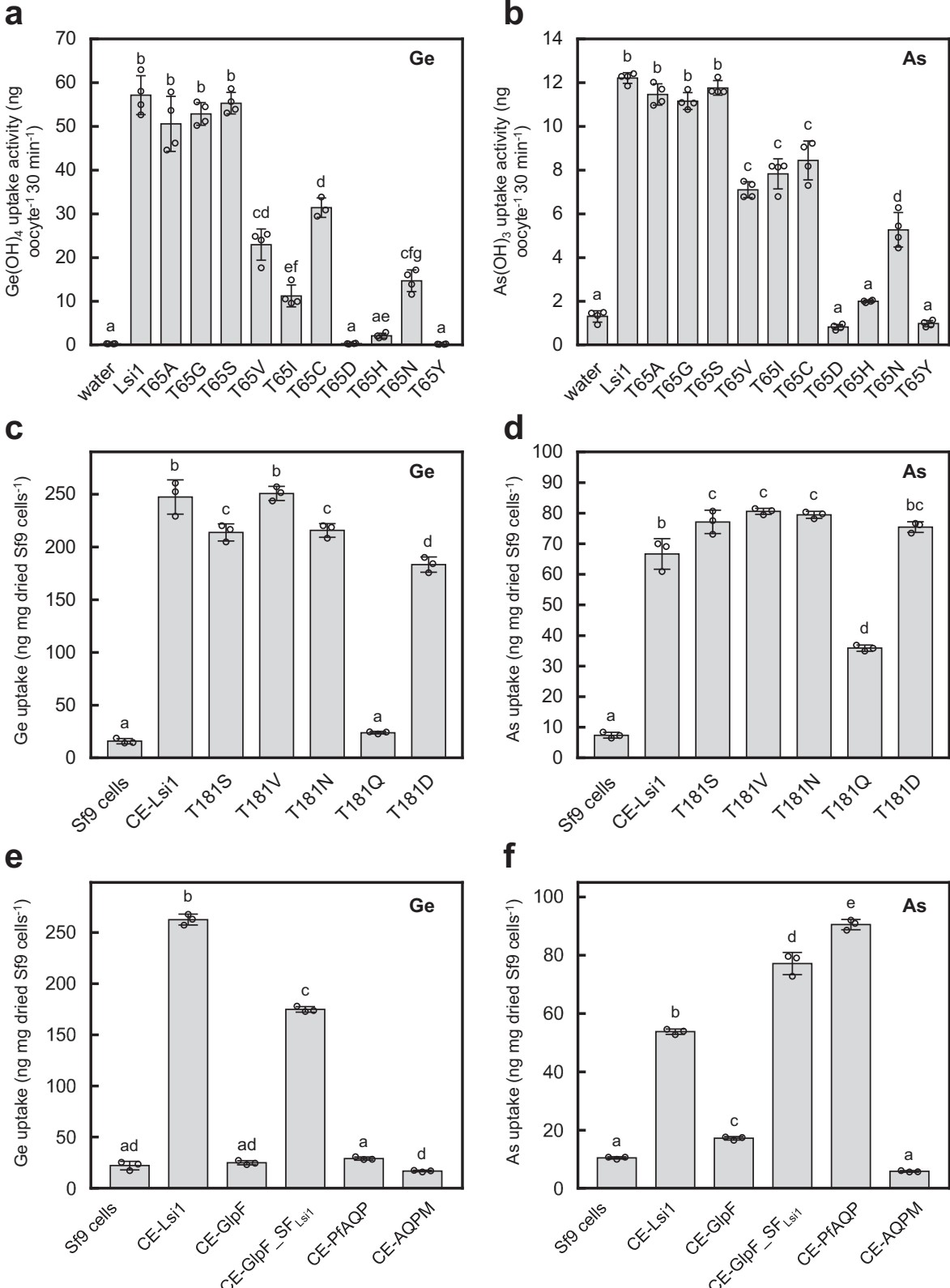

**Fig. 5 Effect of Lsi1 mutation on the transport activity for Ge and As.** Transport activity of germanic acid (Ge) (**a**, Si analog) and arsenite (As) (**b**) in *Xenopus* oocytes. The oocytes were injected with water (as a control) or cRNA of Lsi1 WT or $Thr65_{TM1}$ mutants. Oocytes were exposed to a solution containing Ge or As for 30 min. Transport activity of Ge (**c**, **e**) and As (**d**, **f**) in Sf9 cells. C-terminally EGFP-tagged Lsi1 or Thr181 mutants (**c**, **d**) or other aquaglyceroporins (**e**, **f**) were expressed in the cells. The cells were exposed to a solution containing Ge or As for 5 min. In **a**−**f**, different letters above the columns indicate statistically significant differences at $P < 0.01$ by Tukey−Kramer's test, and the test was two-sided. Values are means ± s.d., $n = 3$ for T65C in (**a**) and $n = 4$ for the others, $n$ is independent experiments.

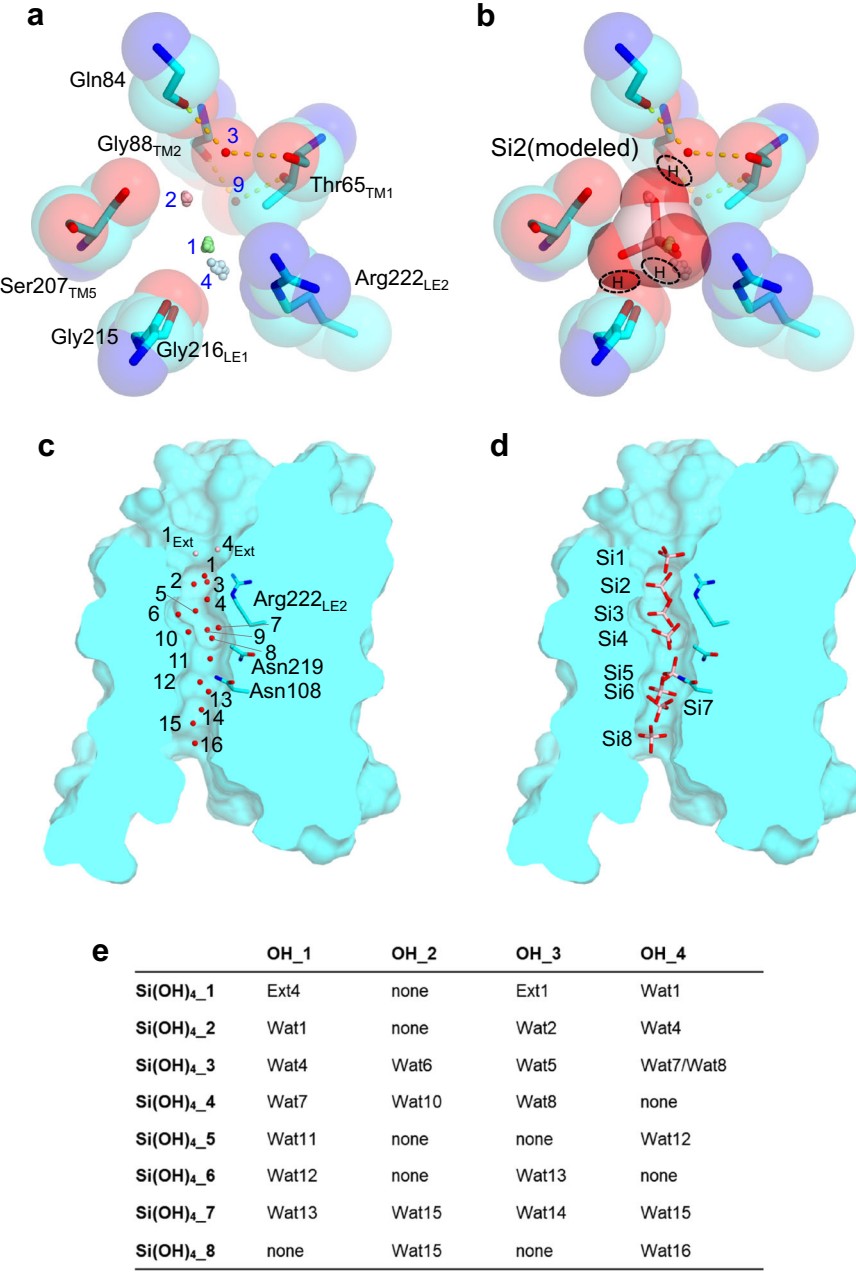

**Fig. 6 Structural determinant for the silicic acid permeability and modeled eight silicic acid positions in the Lsi1 channel.** SF of Lsi1$_{cryst}$ (**a**), and a possible conformation of silicic acid during the passage through SF (**b**). Water molecules in the channel (Wat1, Wat2, Wat3, Wat4, and Wat9) are shown as spheres and numbered in blue. Among them, three (Wat1, Wat2, and Wat4) are superposition from the eight Lsi1$_{cryst}$ protomers of the two tetramers. Putative hydrogens of hydroxyl groups of the silicic acid in the position Si2 are encircled by dashed lines. The view direction is the same as Fig. 2a. Water molecules used for the assignment of possible silicic acid positions (**c**) and the assigned eight silicic acids (Si1-Si8) in the channel (**d**) are shown in a cross-section view. Water molecules in the channel region (red spheres) and the extracellular region (pink spheres) are labeled in numbers. Arg222$_{LE2}$ in the SF region and two Asn residues in the NPA motifs are shown as the stick. **e** Water molecules employed for the assignment of four hydroxyl groups in silicic acid are listed. Ext in **c** and **e** indicates water molecules at the extracellular side.

activity remained about 35% of native Lsi1 (Fig. 5d). It should be noted that the effect of Thr181 mutations is moderate compared to Thr65$_{TM1}$ and its function is still not clear. Nevertheless, the results suggest that the specificity of transport substrate can be modified by manipulating the two Thr residues (Thr65$_{TM1}$ and Thr181) identified in the present study.

**Transport specificity of Lsi1 for silicic acid.** What is the structural basis of Lsi1 yielding the high selectivity for silicic acid rather than smaller glycerol? Our structural analysis showed that the oppositely located two polar faces found in the channel might be responsible for this difference in the transport substrate specificity (Fig. 3a). Since silicic acid is a hydrophilic tetrahedron, the two polar faces can provide an energetically preferable environment by surrounding it. In contrast, linear carbohydrates are amphipathic, and the two polar faces in Lsi1$_{cryst}$ may impair conductivity for glycerol. Indeed, SF of GlpF has an amphipathic property that fits well for glycerol with hydrophobic interactions by Trp48$_{TM2}$ and Phe200$_{LE1}$ and hydrogen bonds formed by Arg206$_{LE2}$ and carbonyl of Phe200$_{LE1}$ with the glycerol's hydroxyl

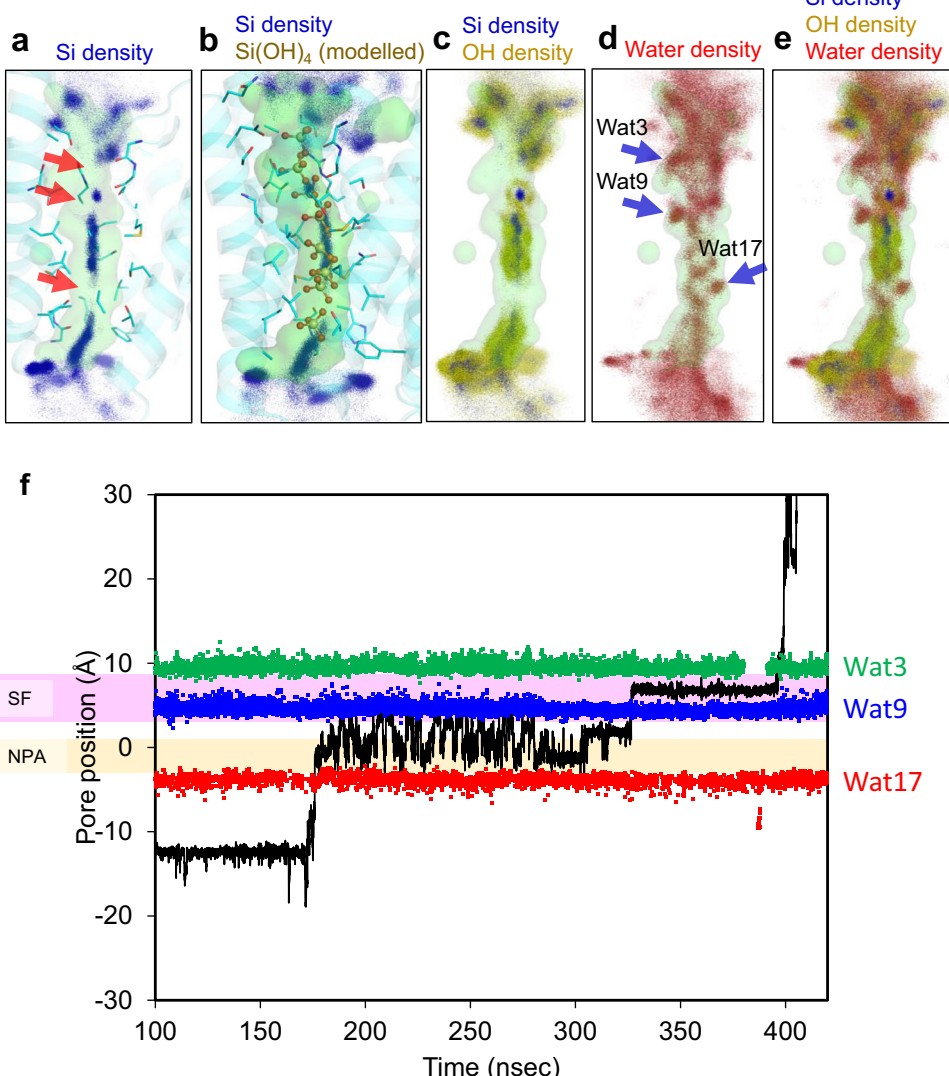

**Fig. 7 MD simulation of Lsi1 with silicic acid.** Distribution of Si atoms of silicic acid (**a**) and its overlay with modeled $Si(OH)_4$ molecules (Si1-Si8) (**b**), and distribution of Si atoms as well as oxygen atoms of silicic acid (**c**), or oxygen atoms of water (**d**) in the 0−300 ns MD simulation. In **a**, structural bottlenecks found in the channel are indicated by red arrows. In **d**, Wat3, Wat9, and Wat17 locations are indicated by blue arrows. Distributions of (**c**) and (**d**) are merged in (**e**). **f** The MD trajectory of the silicic acid permeation during the 100−420 ns. Positions of silicic acid (black), Wat3 (green), Wat9 (blue), and Wat17 (red) in the channel are plotted. Enlarged views of (**f**) and the distributions of $Si(OH)_4$ molecules at the beginning and end of the simulation are provided in Supplementary Fig. 11.

groups (Fig. 3c). To test this possibility, we focused on gain-of-function mutants of the structurally characterized aquaglyceroporin GlpF[14] in an attempt to mimic the Si-permeability of Lsi1. Wild-type GlpF, PfAQP[24], and AQPM[25] did not show transport activity for Ge in Sf9 cells or oocytes (Fig. 5e and Supplementary Figs. 7c, d, 8e). However, with the substitutions of their SF residues into Lsi1 type, GlpF variant GlpF_SF$_{Lsi1}$ (G25T$_{TM1}$, W48G$_{TM2}$, G191S$_{TM5}$, and F200G$_{LE1}$) showed a higher Ge transport activity (Fig. 5e and Supplementary Fig. 8e). Interestingly, GlpF_SF$_{Lsi1}$ also gained the As transport activity but showed the decreased glycerol transport activity (Fig. 5f and Supplementary Fig. 8c, f).

Deshmukh et al. reported that number of residues connecting two NPA motifs is essential for the Si permeability[26]. Since the spacing is 108 residues in Lsi1 and 132 residues in the Si-permeable GlpF variant GlpF_SF$_{Lsi1}$, the spacing itself is unlikely to be critical for the Si specificity. This is supported by a recent study on Lsi1 in tomatoes that possesses 109 residues in the

spacing but shows transport activity for Si[27]. Instead, loop C extensively stabilizes the consecutive carbonyls (Gly215-Ser217) and SF of Lsi1$_{cryst}$ via hydrogen bond interactions (Supplementary Fig. 5b). Taken together, the two polar faces of the channel provided by the unique SF have an essential role in the Si specificity in Lsi1.

**The silicic acid transport mechanism.** We failed to detect any Si- or Ge- derived anomalous signals from the Lsi1$_{cryst}$ crystals soaked in a buffer at saturated concentrations. However, we tentatively predicted eight positions of silicic acid (Si1 through Si8) that possibly occupy the channel in Si transport, based on the positions of waters identified in the channel (Wat1 through Wat16) (Fig. 6a). We postulated that hydroxyl groups of the transported silicic acid should favor the hydrophilic environment that waters bound. We first investigated the modeled silicic acid locations using quantum mechanical/molecular mechanical (QM/MM)

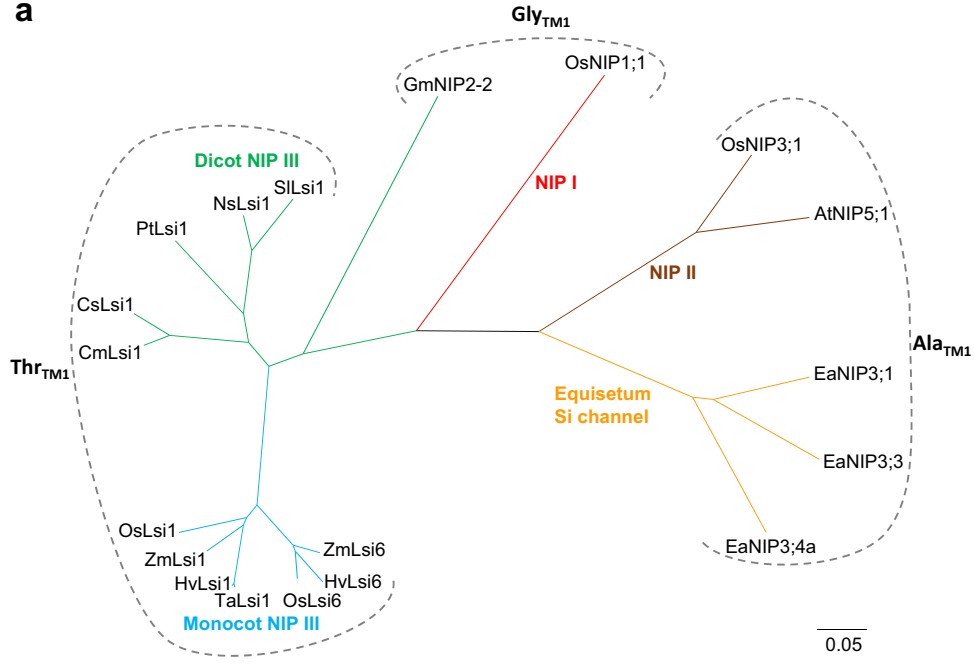

**Fig. 8 Analysis of selectivity filters in NIP subfamilies.** Phylogenetic tree of plant NIP subfamilies (**a**), and summary of the residues comprising the selectivity filer and Thr181 (**b**). The phylogenetic tree was constructed using the neighbor-joining algorithm by the software MEGAX[51] after the sequence alignment using Clustal Omega[52] with 1,000 bootstrap trials. The 0.05 scale indicates substitution distance.

|  | TM1 | TM2 | TM5 | LE1 | LE2 | Thr181 |
|---|---|---|---|---|---|---|
| Monocot NIP III | T | G | S | G | R | T |
| Dicot NIP III without Leguminosae | T | G | S | G | R | T |
| Leguminosae NIP III | G | G | S | G | R | S |
| NIP I | (G) | W | V | A | R | I |
| NIP II | A | A | I | G | R | V |
| Equisetum Si channel | A | S | T | A | R | N |

geometry optimization (Supplementary Fig. 9). Displacement of the QM/MM model ranges from 0.2 to 0.9 Å relative to the modeled silicic acid (Supplementary Fig. 9). The number of hydrogen-bonding interactions in the channel is mainly unchanged after the QM/MM optimization, suggesting that silicic acid can stably occupy the positions through Si1 to Si8.

Intriguingly, the modeled Si2 occupies SF, without any steric hindrance, with the three hydroxyl groups overlapping with three water molecules (Wat1, Wat2, and Wat4), thereby donating hydrogen bonds to Wat3 and two carbonyls (Gly215, and Gly216 $_{LE1}$) (Fig. 6a, b). Si2 thus likely represents a transient conformation during the passage through SF. Two hydroxyl groups go through a crevice formed by Arg222$_{LE2}$ and the two carbonyls (Gly215 and Gly216$_{LE1}$), another one goes through a crevice by the carbonyls and Ser207$_{TM5}$, and the other one goes through a crevice by Ser207 $_{TM5}$ and waters (Wat3 and Wat9) (Fig. 6b). In this way, all hydroxyl groups of the silicic acid can form hydrogen bonds with the carbonyls or waters positioned

along the channel and with Ser207$_{TM5}$ and Arg222$_{LE2}$ to compensate for the energetic cost of dehydration. The other four silicic acids (Si1, Si3, Si4, and Si7) overlap three or four hydroxyl groups with the waters identified, and the other three (Si5, Si6, and Si8) also do so in two hydroxyl groups (Fig. 6c−e). This assignment identifies plausible hydrogen bonding partners of the silicic acid and provides orientations of the former four positions. However, there may be other interpretations because we based our hypothesis on the structure without any substrates. The predicted silicic acids suggested that water molecules can replace one or two hydrogen-bonding interactions with the channel after the passage of SF to promote silicic acid migration.

We also performed a 450 ns MD simulation of the Lsi1 structure in the presence of ~1 M Si(OH)$_4$ to investigate the transport mechanism (Fig. 7 and Supplementary Figs. 10, 11). Two silicic acids permeated the channel during the simulation. During the MD simulation, an average of 11.6 ± 2.6 water molecules and 0.7 ± 0.7 Si(OH)$_4$ molecules occupied in the

channel region, where we found 16 water molecules in the crystal structure (Supplementary Fig. 10f). The simulation results also suggested three structural bottlenecks where silicic acid densities were low during the simulation and silicic acid cannot move freely (Red arrows in Fig. 7a). Two bottlenecks were in SF; one was consistent with the modeled Si2, and the other was between the Si3 and Si4. The third one was near the Thr181 in between the Si7 and Si8. The simulation suggested that each bottleneck has a cavity which the oxygen atoms of silicic acid cannot occupy. Instead, water molecules, Wat3, Wat9, and Wat17 (Wat17 was not identified in the crystal structure and has hydrogen bonding interaction with Thr181 in the MD simulation, see Supplementary Fig. 10b), stably exist within the cavities (Blue arrows in Fig. 7d). When the $Si(OH)_4$ moved across the Wat3 and Wat17 sites, Wat3/Wat17 were dissociated from this site. On the other hand, Wat9 was independent of the $Si(OH)_4$ movement and remained bound to the site (Fig. 7f and Supplementary Figs. 10f and 11). We calculated how often these water molecules occupy the cavities throughout the MD simulation. Their occupancies were 60% for Wat3, 94% for Wat9, and 44% for Wat17. Their exchange rate suggested that Wat9 bound to the site longer (the average exchange time, $\tau = 1.5$ ns) than Wat3 and Wat17 ($\tau \sim 0.3$ ns) (Supplementary Table 2). In this case, Wat3 formed hydrogen bonding interactions with the carbonyls of Gln84 or $Thr65_{TM1}$ in most cases, whereas Wat9 mainly formed two hydrogen bonds with $Thr65_{TM1}$ and $Gly88_{TM2}$. MD snapshots also suggested that Wat3 and Wat9 could accept hydrogen bonds from silicic acid consistent with our proposal from the crystal structure (Supplementary Fig. 10c, e). The silicic acid permeation trajectories and the MD snapshots support that Wat3 and Wat9 remain in the vicinity of their positions during the silicic acid passage while Wat17 will be displaced (Fig. 7f and Supplementary Figs. 10, 11). The channel diameter calculation including Wat9 results in a narrower channel diameter (Fig. 2c and Supplementary Fig. 12). These results indicate that during silicic acid permeation, Wat3 and Wat9 act as part of the channel lumen, narrowing the channel and strictly selecting the orientation of silicic acid, highlighting the importance of the high-resolution structure's ability to visualize most water molecules in the channel.

## Discussion

Plant AQPs have five subfamilies; the plasma membrane intrinsic proteins (PIP) subfamily, the tonoplast intrinsic proteins (TIP) subfamily, the NIP subfamily, the small basic intrinsic proteins subfamily (SIP), and the X intrinsic proteins (XIP) subfamily[28]. In the present study, we determined the high-resolution crystal structure of the rice Si channel Lsi1 belonging to the unique NIP subfamily. We compared the $Lsi1_{cryst}$ structure with other known plant AQP structures, SoPIP2;1 and AtTIP2;1. The SoPIP2;1 structure in the open and closed states have been reported[17]. The striking feature of the SoPIP2;1 structure is that loop D changes its conformation to open or occlude the pore in response to phosphorylation or pH change. In the SoPIP2;1 structure, loop D folds below the pore and occludes it in the closed conformation, whereas loop D flips largely towards TM4 and TM5 of the adjacent protomer, thereby opening the pore in the open conformation. The overall structure of Lsi1 is very similar to that of SoPIP2;1 (The RMSD values are 1.1 Å for the open and 1.2 Å for the closed structures, respectively), except for loop D (Supplementary Fig. 13a). Loop D in the Lsi1 structure flips towards TM2 of the adjacent protomer (Supplementary Fig. 13b). SF of the SoPIP2;1 is narrow and hydrophilic ($Phe81_{TM2}$/$His210_{TM5}$/$Thr219_{LE1}$/$Arg225_{LE2}$) similar to other structures of the water-specific AQPs (Fig. 3g). Thr55, which corresponds to the "fifth

residue" of the Lsi1 structure, is covered by the $Phe81_{TM2}$ and therefore does not face the pore.

On the other hand, AtTIP2;1 is water and ammonia permeable. While the Lsi1 structure is similar to the AtTIP2;1 structure (the RMSD value is 1.4 Å)[18], their loops A and C are quite different. In the Lsi1 structure, loop A orients to the pore's distal side, and loop C is displaced up to 5 Å towards the pore relative to the AtTIP2;1 structure (Supplementary Fig. 14). SF of AtTIP2;1 is narrow and hydrophilic ($His63_{TM2}$/$His131_{LC}$/$Ile185_{TM5}$/$Gly194_{LE1}$/$Arg200_{LE2}$) (Fig. 3h and Supplementary Fig. 15). Gly35, which corresponds to the "fifth residue" of the Lsi1 structure, is covered by the bulky $His63_{TM2}$. SF of AtTIP2;1 is characterized by the fact that the conserved $Arg200_{LE2}$ adopts a unique position and that an additional hydrophilic residue $His131_{LC}$ extended from loop C contributes to SF. Lsi1 has Thr157 at the corresponding position of $His131_{LC}$ in the AtTIP2;1 structure (Supplementary Fig. 15). Thr157 in Lsi1 has hydrogen bonding interaction with Thr223 and exposes its methyl group to the pore. Moreover, MD simulation suggested that $Si(OH)_4$ remains at the position of Thr157 in the crystal structure when the Thr157 is displaced (Supplementary Fig. 15e, f). Therefore, Thr157 in Lsi1 is distinct from the AtTIP2;1 structure and may not directly contribute to substrate selectivity.

Among five residues defining SF of the $Lsi1_{cryst}$ structure, $Thr65_{TM1}$ is unique and likely plays an essential role in the specificity of the transport substrate (Fig. 3a). Among the NIP subfamily (NIP I, II, and III), NIP I subgroup has a bulky $Trp_{TM2}$ in SF, whereas NIP II and III subgroups have small amino acid residues $Gly_{TM2}$ or $Ala_{TM2}$ or $Ser_{TM2}$ (Fig. 8). Therefore, the "fifth residue" of SF corresponding to $Thr65_{TM1}$ in Lsi1 is likely to face the channel in NIP II and III subgroups but not in the NIP I subgroup. In the NIP III subgroup, the "fifth residue" is $Thr_{TM1}$ in monocot and dicot species, except $Gly_{TM1}$ in legumes, whereas it is $Gly_{TM1}$ or $Ala_{TM1}$ in NIP I and II subgroups (Fig. 8). Also, Thr181 is conserved in NIP III subgroup except for leguminosae (Supplementary Fig. 1). Thus $Thr65_{TM1}$ and Thr181 identified in the present study are unique to the NIP III subgroup. The *Equisetum* Si channel group, which also possesses the Si transport activity, has $Ala_{TM1}$, $Ser_{TM2}$, and $Thr_{TM5}$[29] (Fig. 8). Hydroxyl groups of $Ser_{TM2}$ and $Thr_{TM5}$ in the *Equisetum* Si channel may compensate for the substrate specificity or interactions with water molecules provided by the hydroxyl groups of $Thr65_{TM1}$ and $Ser207_{TM5}$ in Lsi1.

Human aquaglyceroporin hAQP10 is permeable to silicic acid[30], and its structure in the closed state has been reported[31]. The RMSD value between the hAQP10 and $Lsi1_{cryst}$ structures is 1.7 Å, which is slightly larger than those calculated between Lsi1 and the structures SoPIP2;1 and AtTIP2;1. The relatively larger RMSD value arises from different structures in loops and different orientations of TM helices (Supplementary Fig. 16a). Nevertheless, SF of hAQP10 is very similar to that of Lsi1 (Fig. 3a, f), consistent with the fact that hAQP10 is permeable to $Si(OH)_4$. SF of hAQP10 is wide and amphiphilic ($Thr35_{TM1}$/$Gly62_{TM2}$/$Gly202_{TM5}$/$Ile211_{LE1}$/$Arg217_{LE2}$). SF of hAQP10 contains many water molecules, including two water molecules Wat3 and Wat9, in the Lsi1 structure, which create the polar face (Fig. 3f and Supplementary Fig. 16c). There are two notable differences in SF between hAQP10 and Lsi1. First, the pore diameter of the hAQP10 is wider in SF (Supplementary Fig. 16b). Second, Gly210 and $Ile211_{LE1}$ provide a row of carbonyls in the pore in the hAQP10 structure, but they are two Gly residues (Gly215 and $Gly216_{LE1}$) in the Lsi1 structure. The side chain of $Ile211_{LE1}$ renders the channel hydrophobic and displaces TM5 of hAQP10 up to 5 Å towards the pore's distal side relative to the Lsi1 structure. Therefore, SF's hydrophilicity and the tilting angles of TM helices are different between the structures hAQP10 and

Lsi1 (Fig. 3f and Supplementary Fig. 4e). The bulky Ile211$_{LE1}$ residue also affects the orientation of the carbonyls. The carbonyl of Gly210 in hAQP10 rotates by 40° towards Arg217$_{LE2}$ compared to Gly215 in the Lsi1 structure. This rotation weakens the hydrogen bond to waters and narrows the region of the pore through which silicic acid may pass. Given that SF of Lsi1 is ideal for silicic acid permeation, the selectivity for silicic acid of hAQP10 may be different from that of Lsi1.

We have shown that the unique TM helix orientations and SF of Lsi1 are essential for silicic acid transport. Lsi1 has presumably acquired these features during its evolution from canonical AQPs that could not transport silicic acid. The evolution of plant AQP family proteins that permeate substrates other than water and glycerol, such as Lsi1 and AtTIP2;1, seems to have involved drastic modification of SF from canonical AQPs. As described, even between the evolutionarily distant species of human (hAQP10) and rice (Lsi1), a shared structure exists in which water molecules create a polar face in SF to transport silicic acid. Such a common structure has likely evolved from convergent evolution.

Lsi1 is also permeable to carcinogenic arsenite[11], the primary form of As in the paddy field. Arsenite is also present in the form of a non-charged molecule and has a similar size as silicic acid[11]. Rice is a staple food for half of the world population but can accumulate high As through Lsi1[11]. Since rice is the primary dietary source of As, it is crucial to reduce As in rice grain for human health. However, compared with silicic acid, usually, arsenite shows broader specificity[12]. For example, two T65I$_{TM1}$ and T181Q substitutions were identified to decrease or abolish Ge-transport activity while As-transport activity was substantially retained (Fig. 5). Also, Lsi1 with a S207I$_{TM5}$ substitution[10] does not transport Ge but transports As. Silicic acid is a tetrahedral molecule that forms four hydrogen bonds, whereas arsenite is a trigonal pyramid that forms three hydrogen bonds (Fig. 1a). Since SF closely matches the transport substrates in the dehydrated form as observed in Lsi1$_{cryst}$ and GlpF[14], or even in the KcsA potassium channel[32], the larger number of possible hydrogen-bonding interactions with the channel as well as the tetrahedral stereochemistry may be the reasons why selectivity for silicic acid is stricter than that for arsenite. While other factors determining transport substrate specificity remain to be investigated, the structure of Lsi1 obtained in this study could serve as a blueprint for rational designs of transgenic crops that specifically take up silicic acid but not arsenite through manipulating the selectivity of Lsi1. Such modification will contribute to safe food production in the future.

## Methods

**Protein expression and purification of Lsi1.** The *Lsi1* gene from rice (*Oryza sativa* cv. Nipponbare) was cloned into the pFastBac1 vector for baculovirus expression in Sf9 insect cells using standard methods. A TEV protease cleavage site and the octa-His affinity tag were introduced between the C terminus of Lsi1 and EGFP. The functionally active construct of Lsi1 was discovered by examining N- and C-terminal deletion constructs, several point mutations, as well as additional Si permeable AQPs from other organisms. All these constructs were created by using the QuikChange II site-directed mutagenesis method (Stratagene) with primers (Supplementary Table 3) and screened by FSEC[15]. Removing 44 residues from the N-terminus, 24 residues from the C-terminus, with seven point mutations (K50R, C66A, T93V, C139A, K232R, T253V, and K264R), yielded the construct, Lsi1$_{cryst}$, used in the crystallographic studies described here. Among the seven mutations in the construct Lsi1$_{cryst}$, four mutations (C66A, T93V, C139A, and T253V) in TM helices enhanced the thermo-stability of Lsi1 in the detergent micelle. Three lysine residues (K50R, K232R, and K264R) in loop regions are mutated to arginine to reduce the surface entropy, hoping that mutants may improve the crystal packing.

Infected Sf9 cells were harvested by centrifugation (8000 × *g*, 15 min), and were disrupted by an ultrasonic disrupter UD-211 (TOMY). After centrifugation (3000 × *g*, 10 min), the supernatant was ultra-centrifuged (200,000 × *g*, 1 h), and membrane fraction was collected and homogenized. The crude membrane fractions were solubilized for 1 h in a buffer containing 500 mM NaCl, 20 mM Tris-HCl pH 8.0, 6% (w/v) glycerol, 1.8% (w/v) *n*-dodecyl-β-D-maltopyranoside (DDM),

0.06 mg/ml RNase A. Insoluble material was removed by ultracentrifugation (148,500 × *g*, 1.5 h) and the supernatant was incubated with TALON cobalt affinity resin (Clontech) for 3 h in the presence of 10 mM imidazole. After washing with a buffer containing 15 mM imidazole, 500 mM NaCl, 20 mM HEPES-NaOH pH 7.5, 10% (w/v) glycerol, and 0.02% (w/v) DDM, Lsi1 mutants were eluted by application of a buffer containing 150 mM imidazole, 500 mM NaCl, 20 mM HEPES-NaOH pH 7.5, 10% (w/v) glycerol, and 0.02% (w/v) DDM. The eluates were precipitated in the presence of 22.2% (w/v) PEG 1500, and then dissolved in a buffer composed of 500 mM NaCl, 20 mM HEPES-NaOH pH 7.5, 10% (w/v) glycerol and 1% (w/v) *n*-octyl-β-D-glucoside (OG). The octa-His tag was cleaved with hexa-His-tagged TEV$_{SH}$[33] (3:1 mass ratio of Lsi1 to TEV$_{SH}$) overnight, and the protein was re-chromatographed on a TALON cobalt affinity resin. The tag cleaved Lsi1 was further purified by gel filtration (Superdex 200 Increase 10/300 GL column) in 500 mM NaCl, 20 mM HEPES-NaOH pH 7.5, and 1% (w/v) OG. All steps were performed at 4 °C unless otherwise noted.

**Crystallization.** The purified protein was concentrated to about 10 mg/ml using a 50 kD molecular weight cut-off centrifugal filter device. The Lsi1$_{cyst}$ crystals were obtained at 7 °C by vapor diffusion sitting drop method by mixing 1:1 (v/v) ratio of protein and a reservoir solution containing 39−50% (w/v) PEG 400, 100 mM Gly-NaOH pH 9.5, 1% (w/v) OG, and 0.1% (w/v) cholesteryl hemisuccinate (CHS). Both pyramidal and rod-shaped crystals appeared in the same crystallization drops, but the diffraction limit of the pyramidal crystals was around 7 Å resolution. The rod-shaped crystals were collected and soaked in a solution containing 41% (w/v) PEG 400, 500 mM NaCl, 20 mM HEPES-NaOH pH 7.5, 100 mM Bis-Tris HCl pH 7.0, 2% (w/v) OG, 0.1% (w/v) CHS, then flash-frozen in liquid nitrogen for X-ray diffraction experiment. Some crystals were soaked in a crystallization buffer supplemented with 44 mM Si(OH)$_4$ or a saturated concentration of Ge(OH)$_4$ for 5 min prior to freezing. A fresh Si(OH)$_4$ solution was prepared to avoid the oligomerization of silicic acids.

**Structure determination.** X-ray diffraction data were collected at SPring-8 BL41XU or BL44XU and were processed with XDS[34]. The crystals belong to the space group $P2_1$ (unit-cell parameters $a = 89.5$ Å, $b = 91.4$ Å, $c = 166.1$ Å, and $\beta = 102.1°$). While some reflections remained sharp, the others were diffused with stronger intensities. This unusual pattern was induced by so-called lattice-translocation defects, in which two identical but translated lattices coexist as a single mosaic block in a crystal. We thus corrected the intensities based on the method reported previously[16]. In brief, if the intensity for each unit cell is $I_{unit}$, the total intensity is

$$I_{total} = (2\kappa^2 - 2\kappa + 1)[1 + 2\kappa(1 - \kappa)/(2\kappa^2 - 2\kappa + 1)\cos(2\pi h\mathbf{t_d})]I_{unit}$$

where the translocation vector $\mathbf{t_d}$ and the fraction $\kappa$ were determined to be (1/3, 0, 1/3) and 0.30, respectively. The corrected intensities gave significantly smaller values for R-factor and free R-factor. They were 0.343 for the R-factor and 0.365 for the free R-factor before the correction and 0.245 for the R-factor, and 0.273 for the free R-factor after the correction. The initial phase information was obtained by molecular replacement with Phaser[35] using a homology model of Lsi1 as a search probe, created from the crystal structure of *Archaeoglobus fulgidus* AQP (PDB 3NE2). Two tetramers were manually built using COOT[36] based on the electron density map calculated at a 1.8 Å resolution and refined using phenix.refine[37]. The two Lsi1 tetramers contained eight sodium ions, four OGs, two CHSs, and eleven PEGs. During the refinement process, coordinates, temperature factors, TLS were refined, and non-crystallographic symmetry restraints were not applied. The statistics for refinement was shown in Table 1. Figures were prepared using Cuemol2 (http://www.cuemol.org) or PyMOL (http://www.pymol.org), and the channel of Lsi1$_{cryst}$ was analyzed using HOLE2[38].

**Functional assay in Sf9 cells.** The full-length *Lsi1*, including the octa-His tag and EGFP at the C-terminus of the *Lsi1* construct (CE-Lsi1), was cloned into the pFastBac1 vector. Similarly, codon-optimized GlpF, AQPM, and PfAQP genes were synthesized (Integrated DNA Technologies), amplified by PCR, cloned into the pFastBac1 vector using a seamless cloning method. All mutated *Lsi1* genes (e.g., Thr65$_{TM1}$ variants, CE-Lsi1_ΔNC, and CE-Lsi1$_{cryst}$) used in the functional assay were generated using the QuikChange II site-directed mutagenesis method (Stratagene).

Sf9 cells infected with P3 viruses of interested CE-Lsi1 mutants or other AQPs (CE-GlpF, CE-AQPM, and CE-PfAQP) were grown at 27 °C for 24 h after infection, followed by growing at 20 °C for an additional 24 h. Cell pellets were suspended with 900 μl of PBS buffer (200 mM NaCl, 2.68 mM KCl, 10 mM Na$_2$HPO$_4$, and 2 mM KH$_2$PO$_4$), and then mixed with about 100 μl of PBS buffer or PBS buffer containing 1 mM Ge(OH)$_4$ or 100 μM As(OH)$_3$. For one assay, $1.5 \times 10^7$ cells were used, and the suspension was in a total volume of 1 ml. After a subsequent 5 min incubation, cells were collected and then dried in a rotary evaporator for 2 days. After nitric acid digestion, the concentration of Ge or As of the dried Sf9 cells were determined with ICP-MS (inductively coupled plasma-mass spectrometry 7700X; Agilent Technologies). The entire experiment was performed in triplicate.

To determine the protein expression in the Sf9 cells, $2.0 \times 10^6$ cells were solubilized with 125 mM Tris-HCl, pH 6.8, 4% SDS, 20% glycerol, 1.5 μM aprotinin, 10 μM leupeptin, 10 μg/ml trypsin inhibitor, and 1 mM PMSF, and then sonicated. The protein contents were determined with a BCA protein assay kit (TaKaRa). A 20 μg of total protein from each sample was separated by SDS-PAGE, transferred to PVDF membrane, detected with the anti-green fluorescent protein tag polyclonal antibody-HRP-DirecT (MBL). As a loading control, the solubilized cell was stained by CBB.

**Transport activity assay in *Xenopus* oocytes**. Oocytes for transport activity assay were isolated from *X. laevis*. Procedures for defolliculation, culture conditions, and selection were the same as described previously[6]. The ORFs of native and mutated *Lsi1*, and *GlpF* were amplified from pFastbac1 plasmids containing the genes described above by PCR. The ORFs of these genes were inserted into the *BglII* site of a *Xenopus* oocyte expression vector, pXβG-ev1 with FLAG tag (DYKDDDDK). Capped RNA was synthesized by in vitro transcription with a mMESSAGE mMACHINE High Yield Capped RNA Transcription Kit (Ambion). A volume of 50 nl (1 ng nl$^{-1}$) cRNA or RNase-free water as a negative control was injected into the oocyte. After 1−3 days of incubation in Modified Barth's Saline (MBS) at 18 ℃, oocytes were subjected to the transport activity assay.

To determine the protein expression in the oocytes, membrane protein was collected from the oocytes by centrifugation according to the previous report[10]. A 5 μg of a membrane protein from each sample was separated by SDS-PAGE, transferred to PVDF membrane, probed with the anti-DYKDDDDK tag monoclonal antibody (Invitrogen), and detected with Anti-Mouse IgG HRP Conjugate (Promega) or mouse monoclonal ANTI-FLAG® M2-HRP antibody (Sigma-Aldrich). As a loading control, the membrane was stained by CBB.

For Ge and As transport activity determination, oocytes were incubated in MBS with 1 mM Ge(OH)$_4$ or 100 μM As(OH)$_3$ for 30 min at 18 ℃. At the end of the uptake, the oocytes were washed in ice-cold MBS and digested with HNO$_3$. Ge and As concentrations in the digested solution were determined by ICP-MS as described above. The permeability of oocytes for glycerol and water was determined by a swelling assay. After cRNA injection and initial incubation in control MBS for 2−3 days, oocytes were transferred to a five-fold diluted MBS for water permeability assay. Changes in the oocyte volume were monitored within 180 s at 20 s intervals. For the glycerol permeability, oocytes were transferred to an isotonic solution containing five-fold diluted MBS supplemented with glycerol to adjust the osmolarity (glycerol concentration was 170 mM). Changes in the oocyte volume were recorded as described above. Permeability of glycerol and water was presented as oocyte volume change [d($V$ $V_0^{-1}$) d$t^{-1}$]. In this study, we used two systems (*Xenopus* oocytes and insect cells) for transport analysis, and the functional results obtained were equivalent or similar in either system.

**QM/MM calculation**. We placed silicic acid molecules (The Cambridge Crystallographic Data Center, the deposition number 1406687) by hand so that their oxygen atoms overlapping with water molecules in the crystal structure. This process allowed us to build eight Si molecules (Si1−Si8), and water molecules employed for the modeling were shown in Fig. 6e. Theoretical analysis (QM/MM calculations and the MD simulation) for Lsi1 was performed using the X-ray crystal structure. Hydrogen atoms were generated and energetically optimized with the CHARMM program[39], while the positions of all non-hydrogen atoms were fixed, and all titratable groups were maintained in their standard protonation state at pH 7. We added additional counterions to neutralize the entire system. Atomic partial charges of the amino acid were adopted from the all-atom CHARMM22 parameter set[40].

For QM/MM calculations, we used the Qsite[41] code and employed the restricted density functional-theory method with the B3LYP functional and LACVP* basis sets. The geometries were refined by constrained QM/MM optimization. To avoid the uncertainty associated with the MM force field, we constrained most of the atoms in the surrounding MM region. Namely, the coordinates of the heavy atoms in the MM region were fixed to the original X-ray coordinates. In contrast, those of the H atoms in the MM region were optimized using the OPLS2005 force field. All atomic coordinates in the QM region were fully relaxed (i.e., not fixed) in the QM/MM calculations. The QM region was defined as the modeled Si(OH)$_4$ in the channel (Si1−Si8; Fig. 6d), water molecules (Fig. 6c) in the channel (Fig. 6), and amino acids H-bonded with them (Thr65, Gln84, Gly104, Val203, Ser207, Gly215, Gly216, the sidechain of Asn108, Met177, Thr181, Asn219, and Arg222, and the backbone of His106, Thr156, and Ser217).

**Molecular dynamics simulation**. For the MD simulation, the Lsi1 tetramer was embedded in a lipid bilayer consisting of 314 1-palmitoyl-2-oleyl-sn-glycero-3-phosphocholine (POPC), using the CHARMM-GUI program[42]. Then the system was soaked in 385 Si(OH)$_4$ and 24215 TIP3P water models[43] (~1 M of Si(OH)$_4$). After structural optimization with positional restraints on heavy atoms of the Lsi1 tetramer, the system was heated from 0.1 to 300 K over 5.5 ps with a time step of 0.01 fs, equilibrated at 300 K for 1 ns with a time step of 0.5 fs, and annealed from 300 to 0 K over 5.5 ps with a time step of 0.01 fs. The same procedure was repeated with positional restraints on the heavy atom of the protein backbone. The same procedure was repeated without positional restraints on any atoms. After an

equilibrating MD run for 15 ns, a production run was conducted over 450 ns with a time step of 1.5 fs. The SHAKE algorithm was used for hydrogen constraints[44]. The MD simulation was based on the CHARMM force field for protein residues[40] and lipids[45]. For Si(OH)$_4$, we employed the parameter set reported by Piane et al.[46], except for the parameter set for the O−H bond, which was taken from the generalized Amber force field (GAFF)[47]. The atomic partial charges of Si(OH)$_4$ were determined by fitting the electrostatic potential by using the RESP procedure[48]. They were 1.0631 for Si atom, −0.7103 for O atom, and 0.4445 for H atom, respectively. The electronic wave functions were calculated after geometry optimization with the density functional theory of the B3LYP/6-31G** level by using JAGUAR[49]. The MD simulation was conducted using the MD engine NAMD[50].

**Reporting summary**. Further information on research design is available in the Nature Research Reporting Summary linked to this article.

## Data availability

The coordinates and structure factors for Lsi1$_{cryst}$ have been deposited in the Protein Data Bank (PDB) with accession number 7CJS. The source data for Figs. 2, 5, and 7, and Supplementary Figs. 3, 6−8, 10−12, and 16 have been provided as the Source Data file. Any other data associated with this manuscript are available from the authors at a reasonable request. Source data are provided with this paper.

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

## Acknowledgements

This work was supported by JSPS KAKENHI Grants JP16H06296 (M.S. and J.F.M.), JP21H05034 (M.S. and J.F.M.), JP17H06879 (Y.S.), JP19K16056 (Y.S.), 18H05155 (H.I.), 18H01937 (H.I.), 20H03217 (H.I.), 20H05090 (H.I.), 16H06560 (K.S.) and 18H01186 (K.S.). This work was also supported by JST CREST JPMJCR1656 (H.I.) and the Inter-disciplinary Computational Science Program in CCS, University of Tsukuba. The X-ray diffraction experiment was performed at beamlines 41XU and 44XU of SPring-8 (Hyogo, Japan) with the approval of the Japan Synchrotron Radiation Research Institute (JASRI) (proposals 2016B6621, 2017A6724, 2017B6724, 2018A2530, 2018B2530, 2019A2559 and 2019B2559), and we thank the staff at SPring-8 for their help. We thank S. Yonekura for the discussion, A. Morita, Y. Takahashi, M. Hikasa and S. Rikiishi for experimental assistance.

## Author contributions

M.S. and J.F.M. conceived the project; M.S. and K.M. screened genes by FSEC; Y.S., K.M., L.Y., J.-R.S. and M.S. purified Lsi1. Y.S. and K.M. crystallized Lsi1. M.S. and Y.S. determined the Lsi1 structure. Y.S., N.M.-U., K.M., N.Y., S.H., M.S., and J.F.M. per-formed functional assays; K.S. and H.I. performed theoretical calculations; Y.S., M.S., N.M.-U. and J.F.M. wrote the paper with inputs from all authors.

## Competing interests

The authors declare no competing interests.
