## [Peer Review File · Nature Communications]

Structural basis for high selectivity of a rice silicon channel Lsi1REVIEWER COMMENTS

Reviewer #1 (Remarks to the Author):

In previous work this group has provided ground breaking findings on the molecular identity of transporter and aquaporin-like proteins that mediate Si uptake and homeostasis that is critical for plant resilience to biotic and abiotic stresses. In the present work, the authors provide a detailed and convincing structural assessment of rice Lsi1, a unique aquaporin-like silicic acid channel of the plant-specific Nodulin Intrinsic Protein family. Specifically they have elucidated a high resolution crystal structure of the core portion of the protein that provides atomic details of the transport channel. Additionally, the authors present a model through which the Lsi1 structure mediates the specific transport of silicic acid. This model is based on the geometry and organization of waters and potential hydrogen bond donors within the channel pore. The quality of the work is high, and the resolution of the crystal structure is impressive. This is not an easy accomplishment for this class of proteins, and the authors should be applauded for their accomplishment.

The impact and novelty of the work include:

- It provides the first high resolution structure for a subfamily of plant specific channels that are essential for metalloid transport in higher plants, the Nodulin Intrinsic Protein family. The outcome of this work will influence investigators within this larger plant membrane community.
- From a larger perspective of aquaporin evolution and structure and function, the work illustrates a novel structure that deviates from classical aquaporin and glyceroporin channels, and shows a new and unprecedented model for how amino acid residues within the canonical selectivity filter of this family are rearranged to allow new functional properties.
- From a methodology point of view, the work provides a guide for the potential elucidation of additional structures of members of this subfamily of aquaporins, as well as a high resolution structure that will guide modeling, computational measurements, and biophysical investigation of these proteins.

The main items that should be addressed more thoroughly are the potential permeation and association of silicic acid in the channel based on this high resolution structure. The structure that is elucidated represents a core fragment of the protein that contains the intact pore region but which lacks the transported substrate. The authors attempted to soak crystals with substrate analogs without success. Nevertheless, the impact of the study holds, and I encourage the authors to consider more quantitative

computational approaches with their structure to provide more thorough models regarding channel dynamics and potential Si permeation and specificity.

Specific comments:

1. In the abstract, the authors note that the data provide a structural basis for Si uptake mechanism, but do not provide details how. To address the potential mechanism, have they considered computational approaches such as Steered Molecular Dynamics, which has been used in several investigations of substrate permeation, transport, and energetics?

2. In order to obtain crystals that diffract to high resolution, Lsi1 with truncated N and C termini and seven amino acid substitutions was generated (Lsi1cryst). What rationale did the authors use to design their mutants? How were these mutants generated and screened?

3. The truncations do not affect the pore forming regions, and most amino acid substitutions (with the exception of two thr to val substitutions in TM2 and TM6) are conservative. Moreover, the authors show that Lsi1cryst largely retains transport properties for the Si substrate homolog Ge. Nevertheless, they should comment briefly on the potential effects of these substitutions on the structure. Based on other AQP, are there potential regulatory functions of the termini?

4. One of the important outcomes of the study is the finding of global conformational changes in the orientation of the TM helices in Lsi1 compared to conventional AQP and glyceroporin structures, and the importance of the location of glycine sidechains at pivotal locations to allow the close approach of helical interfaces. Based on the structures of other transmembrane proteins, glycines are critical for helical flexibility and dynamics. Does Lsi1 show greater conformational flexibility compared to other aquaporin structures, and could this be critical for its transport function?

5. The high quality of the Lsi1 crystal structure allows the identification of multiple water molecules in the vestibule and channel of the structure (discussed on lines 121-123). Is this unusual compared to other aquaporin structures?

6. On page 5 the authors comment: "The results reveal a distinct structure of Lsi1cryst from other aquaporins which is associated with the high selectivity of Lsi1 for silicic acid." It is not clear from the global conformational differences and tilt of α -helices discussed in this section (lines 85-109) how this would favor selectivity for Si. Does the helical conformation of conventional AQPs exclude Si? If so, how?

7. In several places, the authors posit potential hypotheses about the mechanistic role of specific structures in Si permeation. These include:

-The waters in the extracellular vestibule that ensure dehydration of Si and lowers the energetic cost of transport (line 123-4). Is there evidence or computational modeling support for this potential function?

-Lines 144-147, Fig. 2C the authors note a “slight distortion” of the selectivity region that could be beneficial for preventing larger molecules. It is difficult to see the distortion that is being referred to and how this would prevent the transport of larger substrates. This should be clarified.

-Lines 168-71, the waters hydrogen bonded to Thr65 serve as hydrogen bond acceptors during Si transport. However mutations that remove the hydroxyl group of threonine (e.g., ala and gly) do not affect transport. It is postulated that another water molecule compensates for this (lines 183-4), but this seems somewhat conjectural, and there are other interpretations as well.

-Lines 234-255, the authors postulate a detailed permeation mechanism based on the position of channel waters that could indicate the position of Si(OH)₄ hydrogen bonds with amino acid side chains and backbone.

While these hypotheses are theoretically plausible, could more detail on their potential role in permeation be brought to light by molecular dynamics approaches with this high resolution structure? Have they tried to use computational docking programs to investigate the potential structure, orientation, and binding free energy of Si(OH)₄ and other substrates in the selectivity filter? Such analyses may strengthen these hypotheses.

8. On lines 135-7, the authors note that the most striking feature of the Lsi1cryst channel is the large number of water molecules in the extracellular half that are not in single file. Why so they consider this the most striking feature of the pore?

9. Figure 3a, the authors note a water hydrogen bonded to Thr 65 in TM1 within the Ar/R. Is this hydrogen bonded to the side chain hydroxyl?

10. Some mutations differentially affect As vs Si/Ge transport (e.g., T65I; T181Q). Why? How might these substrates differ in their interaction and permeation with NIP III proteins? This should be resolved by comparison of As(OH)₃ and Si(OH)₄ docked models of the structure, which would strengthen a

discussion of potential contact amino acids and specificity. This seems pertinent since the authors discuss in general terms the desire to engineer Lsi1 that has greater selectivity for Si over As containing substrates (lines 290-294).

11. The authors propose that the Lsi1 structure presents two polar faces that result in the ability to hydrogen bond to the tetrahedral $\text{Si}(\text{OH})_4$ molecule compared to amphipathic glycerol (lines 204-211). They provide a strong set of gain of functional assays with GlpF and AQP analogs and mutants to support their hypothesis for this model. Have they tried the converse assay with Lsi1 to determine if gain of function of glycerol transport is obtained with substitutions that eliminate the polar face? Glycerol is quite flexible with several conformations that can mimic hydroxylated metalloids (e.g., *Chem Res Toxicol*, 20:1269). Is glycerol excluded from the Lsi1 pore?

12. From the structure shown in Figure 1d there appears to be cross-subunit contacts with residues within the pore that are not discussed in the text. Do these play any structural role in the Lsi1 channel or overall structure/conformation?

13. Lines 265-268, while a role for Thr65^{TM1} is clear from the Lsi1 structure, it may be premature to extrapolate this to other NIP subclasses I and II since the effect of the residues in the ar/R and other regions are difficult to predict.

14. The discussion section is somewhat short, and it seems appropriate for the authors to discuss the Lsi1 structure from a larger perspective of its novelty with respect to aquaporin structure and evolution. Similarly, the abstract should underscore the high resolution quality of their structure, as well as the larger impact and novelty of their work.

Reviewer #2 (Remarks to the Author):

Saitoh et al presents the high-resolution crystal structure of the silicic acid transporter Lsi1 from rice. This is the first structure of a plant aquaporin from the Nodulin 26-like intrinsic protein (NIP) subfamily and provides novel insights into the mechanism for silicic acid selectivity. This is an important advancement in the field of plant aquaporin structural biology, and may have implications for the development of sustainable crops with altered metalloids uptake profiles.

Although the structure itself is of high interest, I cannot recommend publication of the manuscript in its current form. Most importantly, its analysis with regard to previous work in the field is lacking in several aspects. Furthermore, the proposed mechanism for silicic acid transport, including the modelling of silicic acid in the pore, would be significantly strengthened by molecular dynamic simulation studies. Moreover, the functional assays need complementing with additional work, including some important controls. More details on these aspects as well as some other suggestions for improvement are given below.

Major comments:

1. The authors should compare the Lsi1-structure with other known plant aquaporin structures, such as SoPIP2;1 and AtTIP2;1. In particular, AtTIP2;1 is interesting since it is permeable to ammonia and would be a useful addition when comparing the structural basis for silicic acid selectivity to other solutes. Also, based on the structure of AtTIP2;1, a fifth residue of the selectivity filter was proposed. The authors should discuss this in light of their own proposed fifth residue.
2. The authors argue that there is a connection between helix tilt and silicic acid selectivity. However, they only compare the structure of Lsi1 to those of AQP1 and GlpF. A more thorough comparison with other structurally characterized AQPs would be beneficial. The authors also argue that there are structural differences in loops A and C that could be important for selectivity in Lsi1. However, structural variability in these two loops is not unique to Lsi1, in fact they often show differences when AQP structures are compared. Again, the authors should include more AQPs in their comparison and expand or, if no longer relevant, remove this statement.
3. As mentioned in the manuscript, human AQP10 has also been suggested to transport silicic acid. The authors should include a structural comparison between Lsi1 to AQP10 and discuss any similarities and differences.
4. The construct used for solving crystal structure of Lsi1 contained a number of point mutations. The authors should clarify the basis for including these particular mutations, visualize their location in the structure and expand the discussion on their effect or lack of effect on the structure and function of Lsi1.
5. In figure 2c, the structure of *P. pastoris* Aqy1 is shown as an example of a water-specific AQP (the PDB code is 3ZOJ, not 3Z5J as stated in the figure legend). It would be better to use AQP1 as done in the other comparisons?

6. The manuscript would be significantly strengthened by including molecular dynamics simulations to test the proposed mechanism for silicic acid transport and location of silicic acid molecules in the channel.

7. The authors should clarify why two different cellular systems were used for the functional assays. Are there any differences that could arise from the choice of method that makes the results difficult to compare with each other? The functional studies of silicic acid transport through gain-of-function mutants should be also complemented with control experiments showing that the respective aquaglyceroporin is fully functional in the assay used, i.e. is able to transport glycerol. Moreover, the water and glycerol permeability of Lsi1 and, when relevant for the selectivity discussion, Lsi1 mutants should be tested.

Minor comments:

8. The authors state that the higher number of water molecules in the extracellular vestibule is important for the dehydration of silicic acid to be transported through the channel. However, it is not clear what they are comparing, is it the number of water molecules in the extracellular and cytoplasmic vestibule? If so, how does this compare to for example aquaglyceroporins? Also, how does this fit with AQPs being bidirectional transporters, passively transporting solutes along its concentration gradient in any direction?

9. Several of the figures would benefit from the size of the labels being reduced, for example Fig 1d and Fig 4.

10. The text refers to Figure 2e in a couple of places, I assume this should be 2d.

11. The refinement statistics for the highest resolution shell should be given in Table 1. Also, the entry for multiplicity does not look right. Why is given as a range and what is the number for the highest resolution shell? The table also gives average B-factors for ligands but as far as I can tell these are not mentioned in the manuscript. Any ligands built should be included in methods section, even if they are not important for the conclusions.

12. The manuscript would benefit from being read by a native English speaker.

REVIEWER COMMENTS

Response to the Reviewer #1

Reviewer #1 (Remarks to the Author):

In previous work this group has provided ground breaking findings on the molecular identity of transporter and aquaporin-like proteins that mediate Si uptake and homeostasis that is critical for plant resilience to biotic and abiotic stresses. In the present work, the authors provide a detailed and convincing structural assessment of rice Lsi1, a unique aquaporin-like silicic acid channel of the plant-specific Nodulin Intrinsic Protein family. Specifically they have elucidated a high resolution crystal structure of the core portion of the protein that provides atomic details of the transport channel. Additionally, the authors present a model through which the Lsi1 structure mediates the specific transport of silicic acid. This model is based on the geometry and organization of waters and potential hydrogen bond donors within the channel pore. The quality of the work is high, and the resolution of the crystal structure is impressive. This is not an easy accomplishment for this class of proteins, and the authors should be applauded for their accomplishment.

The impact and novelty of the work include:

- It provides the first high resolution structure for a subfamily of plant specific channels that are essential for metalloid transport in higher plants, the Nodulin Intrinsic Protein family. The outcome of this work will influence investigators within this larger plant membrane community.
- From a larger perspective of aquaporin evolution and structure and function, the work illustrates a novel structure that deviates from classical aquaporin and glyceroporin channels, and shows a new and unprecedented model for how amino acid residues within the canonical selectivity filter of this family are rearranged to allow new functional properties.
- From a methodology point of view, the work provides a guide for the potential elucidation of additional structures of members of this subfamily of aquaporins, as well as a high resolution structure that will guide modeling, computational measurements, and biophysical investigation of these proteins.

The main items that should be addressed more thoroughly are the potential permeation and association of silicic acid in the channel based on this high resolution structure. The structure that is elucidated represents a core fragment of the protein that contains the intact pore region but which lacks the transported substrate. The authors attempted to soak crystals with substrate analogs without success. Nevertheless, the impact of the study holds, and I encourage the authors to consider more quantitative computational approaches with their structure to provide more thorough models regarding channel dynamics and potential Si permeation and specificity.

(Response)

First of all, we thank the reviewer for his/her careful reading of our manuscript, insightful and positive comments. We have carefully considered the comments raised by the reviewer and revised our manuscript accordingly. Major changes in the revised manuscript are the following.

First, we performed MD simulation to verify the Si(OH)_4 transport in Lsi1, and we also verified the modeled Si(OH)_4 in the pore with quantum mechanics/molecular mechanics (QM/MM) calculation. The results reinforced the Si(OH)_4 transport mechanism and provided new insights into the structural roles of the Lsi1 structure, which likely addresses the concerns from the reviewer (see below). Second, we performed the additional functional assays with *Xenopus* oocytes to verify Lsi1 and the mutants' glycerol transport activities. In the mutants, we manipulated the selectivity filter to GlpF type/hydrophobic. Interestingly, against our expectation, Lsi1 likely also transports glycerol molecules. Third, we extended the discussion section by incorporating the structural comparison of the Lsi1 structure between plant AQPs and hAQP10. These modifications are included in the revised manuscript.

Our point-by-point responses are given in the following text, and we sincerely hope that we have adequately addressed the reviewer's concerns.

Specific comments:

1. In the abstract, the authors note that the data provide a structural basis for Si uptake mechanism, but do not provide details how. To address the potential mechanism, have they considered computational approaches such as Steered Molecular Dynamics, which has been used in several investigations of substrate permeation, transport, and energetics?

(Response)

We performed a 300 ns MD simulation of the Lsi1 structure in the presence of 1 M Si(OH)_4 to verify the transport mechanism. Silicic acid permeated the channel during the simulation, consistent with our proposed mechanism. MD simulation results also suggested three structural bottlenecks through the channel during the silicic acid transport (red arrows in Fig. #1a). Two were found in the selectivity filter; one was consistent with the modeled Si2, and the other was between the Si3 and Si4. The third one was near the Thr181 in between the Si7 and Si8. The simulation suggested that each bottleneck has a cavity where the silicic acid's oxygen atoms cannot occupy. Instead, water molecules, Wat3, Wat9, and Wat17 (Wat17 was not identified in the crystal structure and has hydrogen bonding interaction with Thr181 in the MD simulation), stably exist within these cavities (blue arrows in Fig. #1d). The calculated populations are 60% for Wat3, 94% for Wat9, and 44% for Wat17, respectively, in the MD simulation. These data support that Wat3 and Wat9 occupy the cavity during silicic acid passage while Wat17 will be displaced. The second bottleneck near Wat9 was not identified when Wat9 was omitted during the diameter calculation (Fig. #2), highlighting the high-resolution structure's significance in visualizing most of the water molecules in the channel.

We have included this discussion in the revised manuscript and figures for the MD simulation in the revised manuscript.

Fig. #1 MD simulation of Lsi1 with silicic acid.

Distribution of Si atoms of silicic acid (a) and its overlay with modelled Si(OH)₄ molecules (Si1-Si8) (b), and distribution of Si atoms as well as oxygen atoms (c) of silicic acid, or oxygen atoms of water (d) in the 0-300 ns MD simulation. In (a), structural bottlenecks found in the channel are indicated by red arrows. In (d), Wat3, Wat9, and Wat17 locations are indicated by blue arrows. Distributions of (c) and (d) are merged in (e). (f, g) Close up views of water molecules, Wat3, Wat9, and Wat17, which stably exist in the channel.

Fig. #2 Channel diameters and its profile of the Lsi1_{cryst} structure.

Channel diameters are calculated with (pink) and without (blue) Wat3 and Wat9 in the structure (a). The selectivity filter's close-up views are shown without (b) and with (c) Wat3 and Wat9. The structural bottleneck made by Wat9 is indicated by an arrowhead.

2. In order to obtain crystals that diffract to high resolution, Lsi1 with truncated N and C termini and seven amino acid substitutions was generated (Lsi1_{cryst}). What rationale did the authors use to design their mutants? How were these mutants generated and screened?

(Response)

We truncated the N- and C- terminals of Lsi1 as they are predicted to be disordered. Among the seven mutations in the construct Lsi1_{cryst}, four mutations (C66A, T93V, C139A, and T253V) in TM helices enhanced the thermo-stability of Lsi1 in the detergent micelle. Three lysine residues (K50R, K232R, and K264R) in loop regions are mutated to arginine to reduce the surface entropy, hoping that mutants may improve the crystal packing. Truncations and locations of the mutation are visualized (Fig. #3). We screened the expression level and stability of the mutants by FSEC.

We included this statement in the method section and the figure visualizing mutations' location into the supplementary information.

Fig. #3 The location of the mutations in the construct Lsi1_{cryst}.

Side views of the Lsi1_{cryst}, rainbow-colored with the N terminus in blue. Arrowheads indicate the truncation site of N and C terminus. Point mutations which enhanced thermostability of Lsi1 (C66A, T93V, C139A and T253V) are colored in magenta, and those likely reduce the surface entropy of Lsi1 (K50R, K232R and K264R) are colored in light green.

3. The truncations do not affect the pore forming regions, and most amino acid substitutions (with the exception of two thr to val substitutions in TM2 and TM6) are conservative. Moreover, the authors show that Lsi1_{cryst} largely retains transport properties for the Si substrate homolog Ge. Nevertheless, they should comment briefly on the potential effects of these substitutions on the structure. Based on other AQP, are there potential regulatory functions of the termini?

(Response)

As pointed out by the reviewer, the truncations and amino acid substitutions on the Lsi1_{cryst} hardly affected the Ge transport activity. The role in gating the pore by N-terminal has been reported in other AQP homologs (Kreida, Current Opin Struct Biol, 2015). The construct Lsi1_{cryst} maybe lacked such a function, although gating of Lsi1 has not been reported.

4. One of the important outcomes of the study is the finding of global conformational changes in the orientation of the TM helices in Lsi1 compared to conventional AQP and glyceroporin structures, and the importance of the location of glycine sidechains at pivotal locations to allow the close approach of helical interfaces. Based on the structures of other transmembrane proteins, glycines are critical for helical flexibility and dynamics. Does Lsi1 show greater conformational flexibility compared to other aquaporin structures, and could this be critical for its transport function?

(Response)

In the current crystal structure of Lsi1, the crystallographic asymmetric unit contains two tetramers, meaning that 3D structures of eight polypeptide chains have been determined independently. The TM helices' orientation is identical in all protomers, and slight differences were found in the protomers' water molecules. This suggests that Lsi1 does not have greater conformational flexibility compared to other AQP structures. AQPs,

including Lsi1, are bidirectional passive transporters. Conformational flexibility may not be required, unlike other transmembrane proteins.

5. The high quality of the Lsi1 crystal structure allows the identification of multiple water molecules in the vestibule and channel of the structure (discussed on lines 121-123). Is this unusual compared to other aquaporin structures?

(Response)

The high-resolution structures of other AQPs have shown a large number of water molecules identified on the surface of the proteins or in the extracellular/intercellular vestibules. Therefore, multiple water molecules in the vestibule are not unusual, but those found in the channel are unusual compared to other AQP structures. Mainly, water molecules not being single-file in the selectivity filter are the unique feature of Lsi1 and hAQP10 (See our response to comment 8 for a detailed explanation for this water arrangement.).

6. On page 5 the authors comment: “The results reveal a distinct structure of Lsi1_{cryst} from other aquaporins which is associated with the high selectivity of Lsi1 for silicic acid.” It is not clear from the global conformational differences and tilt of α -helices discussed in this section (lines 85-109) how this would favor selectivity for Si. Does the helical conformation of conventional AQPs exclude Si? If so, how?

(Response)

We agree with the reviewer that a relationship between the global conformational differences and the Lsi1 selectivity for silicic acid is not clear. We want to express here that the residues consisting of the selectivity filter of Lsi1 also contribute to the distinct structure of Lsi1. While we found the helical conformation of Lsi1 is unique, we are not sure that the helical conformation of conventional AQPs excludes Si.

We soften the assertive description in the revised manuscript as described below.

These results reveal a distinct structure of Lsi1_{cryst} from other AQPs, which “can be” associated with the high selectivity of Lsi1 for silicic acid.

7. In several places, the authors posit potential hypotheses about the mechanistic role of specific structures in Si permeation. These include:

–The waters in the extracellular vestibule that ensure dehydration of Si and lowers the energetic cost of transport (line 123-4). Is there evidence or computational modeling support for this potential function?

(Response)

In the MD simulation, we found that silicic acid bound to the vestibule and was dehydrated (Fig. #4). We are not sure that this is relevant to aquaglyceroporins; however, a similar discussion has already been made in the PfAQP structure (Newby *et al.*, Nat Struct Mol

Biol., 2008).

We modified the text in the revised manuscript as described below and cited the paper about PfAQP (Newby *et al.*, Nat Struct Mol Biol., 2008).

“Newby *et al.* have made a similar discussion in the PfAQP structure (Newby *et al.*, Nat Struct Mol Biol., 2008).”

Fig. #4 Silicic acid binding sites in the extracellular vestibule.

Three silicic acid binding sites in the extracellular vestibules indicated by the MD simulation. View direction is from the extracellular side onto the pore. The blue dots represent the Si atoms of $\text{Si}(\text{OH})_4$ found in the MD simulation in 0-300 nsec.

-Lines 144-147, Fig. 2C the authors note a “slight distortion” of the selectivity region that could be beneficial for preventing larger molecules. It is difficult to see the distortion that is being referred to and how this would prevent the transport of larger substrates. This should be clarified.

(Response)

We added an arrowhead in Fig. 2c to indicate the distortion. “The shift of HE provides this channel’s distortion which likely prevents the transport of larger substrates than silicic acid, such as silicic acid oligomer, as they unlikely rotate freely in this distortion.”

This explanation is added in the revised manuscript to clarify how the distortion prevents larger substrates’ transport.

-Lines 168-71, the waters hydrogen bonded to Thr65 serve as hydrogen bond acceptors during Si transport. However mutations that remove the hydroxyl group of threonine (e.g., ala and gly) do not affect transport. It is postulated that another water molecule compensates for this (lines183-4), but this seems somewhat conjectural, and there are other interpretations as well.

(Response)

We agree that this description is assertive, and there are other interpretations as well. We modified it into the following text.

This “can indicate” that a polar environment made by hydroxy of ...

-Lines 234-255, the authors postulate a detailed permeation mechanism based on the position of channel waters that could indicate the position of Si(OH)₄ hydrogen bonds with amino acid side chains and backbone.

While these hypotheses are theoretically plausible, could more detail on their potential role in permeation be brought to light by molecular dynamics approaches with this high resolution structure? Have they tried to use computational docking programs to investigate the potential structure, orientation, and binding free energy of Si(OH)₄ and other substrates in the selectivity filter? Such analyses may strengthen these hypotheses.

(Response)

We verified the modeled silicic acid locations using quantum mechanics/molecular mechanics (QM/MM) calculation (Fig. #5). Displacement of the QM/MM model ranges from 0.5 to 1.0 Å relative to the modeled silicic acid. The number of hydrogen-bonding interactions in the channel is largely unchanged after the simulation, suggesting that silicic acid stably occupied the positions through Si1 to Si8. The QM/MM model and the MD simulation strengthen our proposed mechanism for silicic acid transport.

The results of the QM/MM model have been incorporated in the revised manuscript.

However, we cannot determine the QM/MM model of other substrates such as As(OH)₃ unambiguously. Therefore, we would like to stick to the silicic acid transport in this manuscript and will compare it with other substrates in our future studies.

Fig. #5 QM/MM calculation of Lsi1 with the modeled silicic acid.

The modeled $\text{Si}(\text{OH})_4$ molecules (**a**), and the $\text{Si}(\text{OH})_4$ molecules obtained from the QM/MM calculation (**b**), and their comparison with respect to the displacement of the Si atoms and the number of hydrogen bonds with Lsi1 (**c**).

8. On lines 135-7, the authors note that the most striking feature of the Lsi1cryst channel is the large number of water molecules in the extracellular half that are not in single file. Why so they consider this the most striking feature of the pore?

(Response)

Water molecules are single-file in most of the structures of AQPs and aquaglyceroporins. This water molecule arrangement has been known to be crucial for the rapid transport of waters or preventing proton leakage. The exception is the structures of Lsi1 and hAQP10, both of which are known to be a channel for silicic acid. In the hAQP10 structure paper, this unique water arrangement was not mentioned. Therefore, we compared the Lsi1 structure with hAQP10 and included this discussion in the revised manuscript in response to reviewer 2.

9. Figure 3a, the authors note a water hydrogen bonded to Thr 65 in TM1 within the Ar/R. Is this hydrogen bonded to the side chain hydroxyl?

(Response)

Yes. This water Wat9 has hydrogen bond interactions with the side chain hydroxyl of Thr65_{TM1} and the carbonyl oxygen of Gly88_{TM2}. Fig. 4a in the manuscript shows the hydrogen bonds.

Fig. Wat3 and Wat9 in the channel of the Lsi1_{cryst} structure.

This figure is a copy of Fig. 4a in the revised manuscript.

10. Some mutations differentially affect As vs Si/Ge transport (e.g., T65I; T181Q). Why? How might these substrates differ in their interaction and permeation with NIP III proteins? This should be resolved by comparison of As(OH)₃ and Si(OH)₄ docked models of the structure, which would strengthen a discussion of potential contact amino acids and specificity. This seems pertinent since the authors discuss in general terms the desire to engineer Lsi1 that has greater selectivity for Si over As containing substrates (lines 290-294).

(Response)

As(OH)₃ has three hydroxy groups, fewer than the number of Si(OH)₄, it is likely related to the different substrate transport effect. Both mutations, T65I and T181Q, likely convert the pore facing residue larger, thereby narrowing the pore. The T65I mutation in the QM/MM model results in the ejection of water molecule Wat9 (Fig. #6). The Wat9 has a crucial role during the passage of silicic acid. Therefore, T65I mutation is unfavorable for silicic acid transport. Also, the silicic acid likely has a severe hindrance with the mutated residue T181Q in the QM/MM model (Fig. #6). However, As(OH)₃ likely permeates the pore without directing the hydroxyl group towards T65I or T181Q residues (Fig. #6). These findings may explain the different effects on the As vs. Si transport in some mutations. However, we did not include the docking models with As(OH)₃ in the revised manuscript because we could not determine the QM/MM model of the As(OH)₃ unambiguously.

Fig. #6 Possible $\text{Si}(\text{OH})_4$ and $\text{As}(\text{OH})_3$ docked models in the mutated $\text{Lsi1}_{\text{cryst}}$ structure. T65I (a) or T181Q (b) mutation is introduced into the $\text{Si}(\text{OH})_4$ docked (left) or $\text{As}(\text{OH})_3$ docked (right) QM/MM models. $\text{As}(\text{OH})_3$ docked QM/MM model was obtained based on the $\text{Si}(\text{OH})_4$ docked model with the Lsi1 crystal structure.

11. The authors propose that the Lsi1 structure presents two polar faces that result in the ability to hydrogen bond to the tetrahedral $\text{Si}(\text{OH})_4$ molecule compared to amphipathic glycerol (lines 204-211). They provide a strong set of gain of functional assays with GlpF and AQP analogs and mutants to support their hypothesis for this model. Have they tried the converse assay with Lsi1 to determine if gain of function of glycerol transport is obtained with substitutions that eliminate the polar face? Glycerol is quite flexible with several conformations that can mimic hydroxylated metalloids (e.g., Chem Res Toxicol, 20:1269). Is glycerol excluded from the Lsi1 pore?

(Response)

We have determined the water and glycerol permeability of Lsi1 and the mutants in which the selectivity filter was manipulated to GlpF type/hydrophobic. They include the full GlpF-type mutant (T65G/G88W/S207G/G216F), the partial GlpF type mutant (T65G/G88W), and a hydrophobic mutant (S207I). All mutants showed partially/completely abolished the transport activities compared to the wild-type (Fig. #7). This suggests that the mutations may render the selectivity filter hydrophobic, but they can also disrupt the channel. Consistent with this notion, the full GlpF type mutant became unstable in the detergent micelle (Fig. #8).

We added the following text in the revised manuscript.

“We also examined the effect on the Lsi1 glycerol permeability by converting the selectivity filter into a hydrophobic one. We measured water and glycerol transport activity of wild type Lsi1 and the full GlpF type mutant (T65G_{TM1}, G88W_{TM2}, S207G_{TM5}, and G216F_{LE1}), the partial GlpF-type mutant (T65G_{TM1} and G88W_{TM2}), and a hydrophobic mutant S207I_{TM5}. All mutants showed partially/completely abolished the glycerol transport activity compared to wild-type. These results suggest that these mutations may render the selectivity filter hydrophobic, but they can also disrupt the channel.”

Against our expectation, Lsi1 also transports glycerol. In our initial report for Lsi1 (Ma *et al.*, Nature, 2006), we measured glycerol permeability of Lsi1 in the oocyte in the presence of 2 mM glycerol, which was lower than 170 mM used in the current study. The difference in the glycerol concentration used in the assay may be the reason why we did not observe the transport activity before. As the reviewer mentioned, glycerol is quite flexible with several conformations that can mimic hydroxylated metalloids. There is no surprise that Lsi1 permeates glycerol through the wide and hydrophilic pore.

We corrected the manuscript explaining that Lsi1 permeable to glycerol and this explanation in the discussion.

Fig. #7 Effect of aquaglyceroporin mimic mutations of Lsi1 on the transport activity for glycerol, water, Ge and As.

Transport activity of glycerol (a), water (b), germanic acid (c) and arsenite (d) in *Xenopus* oocytes. In (a-d), different letters above the columns indicate statistically significant differences at $P < 0.05$ by Tukey-Kramer's test. Values are means \pm s.d. ($n = 4$).

Fig. #8 FSEC analysis of aquaglyceroporin mimic Lsi1 mutants.

FSEC analysis of S207I (a), T65G+G88W (b), T65G+G88W+S207G+G216F (c) Lsi1 mutants used for the transport activity measurement.

12. From the structure shown in Figure 1d there appears to be cross-subunit contacts with residues within the pore that are not discussed in the text. Do these play any structural role in the Lsi1 channel or overall structure/conformation?

(Response)

We carefully inspect the structure and found that TM2 and TM5 make a hollow in the cross-subunit contact. This hollow extends laterally towards the adjacent protomer, resembling a “cave” (Fig. #9). The hollow contains three water molecules, and the water-mediated hydrogen bonding network connects to the channel. Thus, these cross-subunit contacts may have a role in stabilizing the tetramer.

Fig. #9 A hollow made by TM2 and TM5 in the cross-subunit contact.

13. Lines 265-268, while a role for Thr65TM1 is clear from the Lsi1 structure, it may be

premature to extrapolate this to other NIP subclasses I and II since the effect of the residues in the ar/R and other regions are difficult to predict.

(Response)

The selectivity filter of Lsi1 and other NIP subgroups has been predicted based on the structures and sequence alignment of AQP homologs. No regard has been given to the role of the “fifth residue” Thr65_{TM1} since the “fifth residue” is covered by a bulky residue on TM2 in other AQP structures. However, the Lsi1 structure has Gly88 on TM2. Therefore, Thr65 is exposed to the pore. The sequence alignment revealed that NIP II and Equisetum Si channels have a small residue on TM2, likewise, Lsi1, indicating exposure of the “fifth residue” to the pore. Since structural prediction does not always work correctly, we have verified the homology model and confirmed that there is no large displacement of the residues, at least, in the selectivity filter (data not shown). Therefore, we believe it is not premature to predict other NIP subgroups’ selectivity filter based on our findings.

Since we believe this is an important finding for the community and the effect of the “fifth residue” on the plant AQP functions should be verified in future studies, we would like to keep this prediction in this paper.

14. The discussion section is somewhat short, and it seems appropriate for the authors to discuss the Lsi1 structure from a larger perspective of its novelty with respect to aquaporin structure and evolution. Similarly, the abstract should underscore the high resolution quality of their structure, as well as the larger impact and novelty of their work.

(Response)

The phylogenetic analysis has shown that the bacterial and animal MIPs have diverged from the two large groups, the aquaporin group and the aquaglyceroporin group, which evolved by gene duplication. However, all high plant MIPs belong to the aquaporin group lacking the MIPs representing the aquaglyceroporin group even though NIP subgroups permeate larger substrates, including glycerol. While several hypotheses exist, there is as yet no consensus about the evolution of NIP (Danielson and Johanson, *Adv Exp Med Biol.*, chapter 2, 2010). Silicic acid channel genes have been identified in angiosperms and pteridophytes. The gene between angiosperms and pteridophytes has low sequence homology. This suggests that the silicic acid channel has evolved since angiosperms and gymnosperms have separately evolved (Trembath-Reichert *et al.*, *PNAS*, 2015).

Because the evolution of NIP is unclear, discussion about the Lsi1 structure concerning aquaporin structure and evolution may not be appropriate in this paper. Instead, we extended the discussion section by comparing the Lsi1 structure with other plant AQPs structures, AtTIP2;1 and SoPIP2;1. AtTIP2;1 is interesting since it is known to permeate ammonium. We also compared the Lsi1 structure with the hAQP10 structure, which is known to permeate silicic acid. We hope this modification is satisfactory with the reviewer.

Response to the Reviewer #2

Reviewer #2 (Remarks to the Author):

Saitoh et al presents the high-resolution crystal structure of the silicic acid transporter Lsi1 from rice. This is the first structure of a plant aquaporin from the Nodulin 26-like intrinsic protein (NIP) subfamily and provides novel insights into the mechanism for silicic acid selectivity. This is an important advancement in the field of plant aquaporin structural biology, and may have implications for the development of sustainable crops with altered metalloid uptake profiles.

Although the structure itself is of high interest, I cannot recommend publication of the manuscript in its current form. Most importantly, its analysis with regard to previous work in the field is lacking in several aspects. Furthermore, the proposed mechanism for silicic acid transport, including the modelling of silicic acid in the pore, would be significantly strengthened by molecular dynamic simulation studies. Moreover, the functional assays need complementing with additional work, including some important controls. More details on these aspects as well as some other suggestions for improvement are given below.

(Response)

First of all, we thank the reviewer for his/her careful reading of our manuscript, insightful and positive comments. We have carefully considered the comments raised by the reviewer and revised our manuscript accordingly.

Our point-by-point responses are given in the following text, and we sincerely hope that we have adequately addressed the reviewer's concerns.

Major comments:

1. The authors should compare the Lsi1-structure with other known plant aquaporin structures, such as SoPIP2;1 and AtTIP2;1. In particular, AtTIP2;1 is interesting since it is permeable to ammonia and would be a useful addition when comparing the structural basis for silicic acid selectivity to other solutes. Also, based on the structure of AtTIP2;1, a fifth residue of the selectivity filter was proposed. The authors should discuss this in light of their own proposed fifth residue.

(Response)

Thank you for your constructive suggestion! In the revised manuscript, we have compared the Lsi1 structure with other known plant AQP structures, SoPIP2;1 (PDB 1Z98, 2B5F) and AtTIP2;1 (PDB 5I32) as indicated. The SoPIP2;1 structure in the open and closed states has been reported. The striking feature of the SoPIP2;1 structure is that loop D changes its conformation to open or occlude the pore in response to phosphorylation or change of pH. In the SoPIP2;1 structure, loop D folds below the pore and occludes it in the closed conformation, whereas loop D flips largely towards TM4 and TM5 of the adjacent protomer, thereby opening the pore in the open conformation. The overall structure of Lsi1 is very similar to that of SoPIP2;1 (The RMSD values are 1.1 Å for the open and 1.2 Å for the closed structures, respectively.), except for loop D (Fig. #1a). Loop D in the Lsi1 structure

flips towards TM2 of the other adjacent protomer with interactions between Arg189 (A) and His101 (B) and between Leu194 (A) and Tyr97 (B) (Fig. #1b). The selectivity filter of the SoPIP2;1 is narrow and hydrophilic (Phe81_{TM2}/His210_{TM5}/Thr219_{LE1}/Arg225_{LE2}), which is consistent with that SoPIP2;1 is the water-specific AQPs (Fig. #2a). However, Thr55, which corresponds to the fifth residue of the Lsi1 structure, is covered by the Phe81_{TM2} and therefore not facing the pore.

On the other hand, AtTIP2;1 is water and ammonia permeable AQPs. While the Lsi1 structure is similar to the AtTIP2;1 structure (The RMSD value is 1.4 Å), notable differences are found in loops A and C. In the Lsi1 structure, loop A is oriented to the distal side of the pore while it orients towards the pore in the AtTIP2;1 structure, and loop C in the Lsi1 structure is displaced up to 5 Å towards the pore relative to the AtTIP2;1 structure (Fig. #3). The selectivity filter of AtTIP2;1 is narrow and hydrophilic (His63_{TM2}/His131_{LC}/Ile185_{TM5}/Gly194_{LE1}/Arg200_{LE2}) (Fig. #2b). Gly35, which corresponds to the fifth residue of the Lsi1 structure, is covered by the bulky His63_{TM2}. The structural feature in the selectivity filter of AtTIP2;1 is characterized by the fact that the conserved Arg200_{LE2} adapts a unique position and that additional hydrophilic residue His131_{LC} extended from loop C constitutes the selectivity filter (Fig #4a-d). Lsi1 has Thr157 at the corresponding position of His131_{LC} in the AtTIP2;1 structure. Thr157 may not directly constitute the selectivity filter because it has hydrogen bonding interaction with Thr223 and extends its methyl group to the pore. However, MD simulation suggested that Si(OH)₄ remains at the position of Thr157 in the crystal structure when the Thr157 is displaced (Fig #4e, f). Therefore, Thr157 in Lsi1 may have another role in the selectivity for Si(OH)₄.

We have incorporated these comparisons into the discussion section as suggested. We have added figures of the selectivity filters of SoPIP2;1 and AtTIP2;1 into Fig. 3g and 3h. Furthermore, we also have added the figures comparing the Lsi1 structure and the SoPIP2;1 and AtTIP2;1 structure into the supplementary information.

Fig. #1 Comparison of the structures of Lsi1 and SoPIP2;1.

Superposition of the structure of Lsi1_{cryst} (cyan) with SoPIP2;1 (PDB 1Z98 close conformation, lime green; PDB 2B5F, open conformation, magenta) (a). Loop D is highlighted in bold sticks. Interaction between the loop D and adjacent monomer's TM2 is shown in (b).

the Lsi1 structure (b). In (b), chains A and B are colored in cyan and light blue, respectively.

Fig. #2 Selectivity filter of SoPIP2;1 and AtTIP2;1.

Selectivity filter of SoPIP2;1 (a) (PDB 1Z98) and AtTIP2;1 (b) (PDB 5I32) are shown in lime green and light purple, respectively. The view direction is the same as Fig. 3.

Fig. #3 Comparison of the structures of Lsi1 and AtTIP2;1.

Superposition of the structure of Lsi1_{cryst} (cyan) with AtTIP2;1 (PDB 5I32, light purple) (a). Loop A and C are highlighted in bold sticks. Close-up view of the loop A (b) and loop C (c).

Fig. #4 Comparison of the selectivity filters of Lsi1 and AtTIP2;1.

The selectivity filter of Lsi1_{cryst} (a and c, cyan), and AtTIP2;1 (b and d, light purple, PDB 5I32), are shown together with the additional residue from loop C. Top views (a and b) and side views (c and d) of the selectivity filter. In (a and b), the view direction is the same as Fig. 3. The Si atom positions of silicic acid (blue dots) during the 0-300 ns MD simulation (e and f). Thr157 in the crystal structure is shown in yellow, and the Si atom positions at T157 is indicated by a red arrow. All Thr157 conformations during the MD simulation is overlaid in (f).

2. The authors argue that there is a connection between helix tilt and silicic acid selectivity. However, they only compare the structure of Lsi1 to those of AQP1 and GlpF. A more thorough comparison with other structurally characterized AQPs would be beneficial. The authors also argue that there are structural differences in loops A and C that could be important for selectivity in Lsi1. However, structural variability in these two loops is not unique to Lsi1, in fact they often show differences when AQP structures are compared. Again, the authors should include more AQPs in their comparison and expand or, if no longer relevant, remove this statement.

(Response)

We compared the Lsi1 structure with additional eleven AQP structures (Fig. #5). They are PfAQP (PDB 3C02), AQPM (PDB 2F2B), hAQP7 (PDB 6QZI), hAQP10 (PDB 6F7H), AQPZ (PDB 1RC2), Aqy1 (PDB 3ZOJ), hAQP2 (PDB 4NEF), hAQP4 (PDB 3GD8), hAQP5 (PDB 3D9S), SoPIP2;1 (PDB 1Z98), and AtTIP2;1 (PDB 5I32). Among them, seven structures (PfAQP, AQPM, hAQP7, hAQP10, AQPZ, hAQP2, and AtTIP2;1) showed the similar tilt in four helices. The other four structures have the tilt in three helices (Aqy1, and SoPIP2;1) or two helices (hAQP4, and hAQP5). Based on these findings, it seems that tilt in four helices in the Lsi1 structure is unique and distinct from other AQP structures. To support this, we have included Fig #5 in the supplementary material in the revised manuscript.

Regarding loops A and C, we found that Lsi1 has the shortest loop A, which may be related to the unique helical orientations of TM1 and TM2. In contrast, the length of loop C varies between the structures. So we removed the statement about loop C in the revised manuscript.

Fig. #5 Comparison of the structures of Lsi1 and other AQPs.

Superposition of the structure of Lsi1_{cryst} (cyan) with other AQPs (green or gray), GlpF (a), PfAQP (b), AQPM (c), hAQP7 (d), hAQP10 (e), AQPZ (f), Aqy1 (g), hAQP2 (h), hAQP4 (i), hAQP5 (j), SoPIP2;1 (k), and AtTIP2;1 (l). View directions are same with Fig. 2a.

3. As mentioned in the manuscript, human AQP10 has also been suggested to transport silicic acid. The authors should include a structural comparison between Lsi1 to AQP10 and discuss any similarities and differences.

(Response)

The RMSD value between the structures Lsi1 and hAQP10 is 1.7 Å, which is slightly larger than those calculated between Lsi1 and the structures SoPIP2;1 and AtTIP2;1. The relatively larger RMSD value arises from different structures in loops and different orientations of TM helices (Fig. #6a). Nevertheless, the selectivity filter of hAQP10 is very similar to that of Lsi1, consistent with the fact that hAQP10 is permeable to Si(OH)₄. The selectivity filter of hAQP10 is wide and amphiphilic (Thr35_{TM1}/Gly62_{TM2}/Gly202_{TM5}/Ile211_{LE1}/Arg217_{LE2}). As seen in the Lsi1 structure, a large number of water molecules exist in the selectivity filter of hAQP10. There are two notable differences in the selectivity filter between hAQP10 and Lsi1. First, pore diameter of the hAQP10 is further larger in the selectivity filter (Fig. #6b, c). Second, Gly210 and Ile211_{LE1} provide a row of carbonyls in the pore in the hAQP10 structure while it is made by two Gly residues (Gly215 and Gly216_{LE1}) in the Lsi1 structure. Due to its large side chain of Ile211_{LE1}, TM5 of hAQP10 is displaced up to 5 Å toward the pore's distal side relative to the Lsi1 structure, therefore tilting angles of TM helices are different between the structures hAQP10 and Lsi1 (Fig. #5e). The bulky Ile211_{LE1} residue also affects the orientation of the carbonyls. Indeed, the carbonyl of Gly210 in hAQP10 rotates by 40° towards Arg217_{LE2} compared to Gly215 in the Lsi1 structure.

We have included this comparison in the discussion section of the revised manuscript and the structural comparison figure in the supplementary information.

Fig. #6 Comparison of the structures of Lsi1 and hAQP10.

a, Superposition of the structure of Lsi1_{cryst} (cyan) with hAQP10 (PDB 6F7H, orange). Dotted circles indicate the areas where TM helices tilt in different orientations. **b**, Channel profile (left) and diameters (right) along the pore for Lsi1_{cryst} and hAQP10. **c**, Cross-section of the channel for Lsi1_{cryst} (left, cyan) and hAQP10 (right, orange).

4. The construct used for solving crystal structure of Lsi1 contained a number of point mutations. The authors should clarify the basis for including these particular mutations, visualize their location in the structure and expand the discussion on their effect or lack of effect on the structure and function of Lsi1.

(Response)

We truncated the N- and C- terminals of Lsi1 as they are predicted to be disordered. Among the seven mutations in the construct Lsi1_{cryst}, four mutations (C66A, T93V, C139A, and T253V) in TM helices enhanced the thermo-stability of Lsi1 in the detergent micelle. Three lysine residues (K50R, K232R, and K264R) in loop regions are mutated to arginine to reduce the surface entropy, hoping that mutants may improve the crystal packing. Truncations and locations of the mutation are visualized (Fig. #7). We screened the expression level and stability of the mutants by FSEC. Because a role in gating the pore by N-terminal has been reported in other AQP homologs (Kreida and Törnroth-Horsefield, Current Opin Struct Biol, 2015), the construct we used maybe lacked such a function of Lsi1.

We have included this statement in the method section and Fig. #7 into the supplementary information; however, we did not expand the discussion on the effect on the structure and function due to the limited space.

Fig. #7 The location of the mutations in the construct Lsi1_{cryst}.

Side views of the Lsi1_{cryst}, rainbow-colored with the N terminus in blue. Arrowheads indicate the truncation site of N and C terminus and point mutations. Point mutations which enhanced thermo-stability of Lsi1 (C66A, T93V, C139A and T253V) are colored in magenta, and those likely reduce the surface entropy of Lsi1 (K50R, K232R and K264R) are colored in light green.

5. In figure 2c, the structure of *P. pastoris* Aqy1 is shown as an example of a water-specific AQP (the PDB code is 3ZOJ, not 3Z5J has stated in the figure legend). It would be better to use AQP1 as done in the other comparisons?

(Response)

We used the Aqy1 structure as an example of a water-specific AQP because there are

fewer water molecules in the AQP1 structure, probably due to its lower resolution.

We thank the reviewer for pointing out our mistake in the PDB code.

6. The manuscript would be significantly strengthened by including molecular dynamics simulations to test the proposed mechanism for silicic acid transport and location of silicic acid molecules in the channel.

(Response)

In response to your comments, we have performed a 300 ns MD simulation of the Lsi1 structure in the presence of 1 M $\text{Si}(\text{OH})_4$ to verify the transport mechanism. Silicic acid permeated the channel during the simulation, consistent with our proposed mechanism. MD simulation results also suggested three structural bottlenecks through the channel during the silicic acid transport (red arrows in Fig. #8a). Two were found in the selectivity filter; one was consistent with the modeled Si2, and the other was between the Si3 and Si4. The third one was near the Thr181 in between the Si7 and Si8 (Fig. #8b). The simulation suggested that each bottleneck has a cavity where the silicic acid's oxygen atoms cannot occupy. Instead, water molecules, Wat3, Wat9, and Wat17 (Wat17 was not identified in the crystal structure and has hydrogen bonding interaction with Thr181 in the MD simulation), stably exist within these cavities (blue arrows in Fig. #8d). The calculated populations are 60% for Wat3, 94% for Wat9, and 44% for Wat17, respectively, in the MD simulation. These data support that Wat3 and Wat9 occupy the cavity during silicic acid passage while Wat17 will be displaced. The second bottleneck near Wat9 was not identified when Wat9 was omitted during the diameter calculation (Fig. #9), highlighting the high-resolution structure's significance in visualizing most of the water molecules in the channel.

We also verified the modeled silicic acid locations by using quantum mechanics/molecular mechanics (QM/MM) calculation (Fig. #10). Displacement of the QM/MM model ranges from 0.5 to 1.0 Å relative to the modeled silicic acid. The number of hydrogen-bonding interactions in the channel is mostly unchanged after the simulation, suggesting that silicic acid stably occupied the positions through Si1 to Si8.

We have included this discussion in the revised manuscript, figures for the MD simulation as Fig. 7, and figures for QM/MM simulation in the supplementary information.

Fig. #8 MD simulation of Lsi1 with silicic acid.

Distribution of Si atoms of silicic acid (a) and its overlay with modelled Si(OH)₄ molecules (Si1-Si8) (b), and distribution of Si atoms as well as oxygen atoms (c) of silicic acid, or oxygen atoms of water (d) in the 0-300 ns MD simulation. In (a), structural bottlenecks found in the channel are indicated by red arrows. In (d), Wat3, Wat9, and Wat17 locations are indicated by blue arrows. Distributions of (c) and (d) are merged in (e). (f, g) Close up views of water molecules, Wat3, Wat9, and Wat17, which stably exist in the channel.

Fig. #9 Channel diameters and its profile of the Lsi1_{cryst} structure.

Channel diameters are calculated with (pink) and without (blue) Wat3 and Wat9 in the structure (a). The selectivity filter's close-up views are shown without (b) and with (c) Wat3 and Wat9. The structural bottleneck made by Wat9 is indicated by an arrowhead.

Fig. #10 QM/MM calculation of Lsi1 with the modeled silicic acid.

The modeled Si(OH)_4 molecules (a), and the Si(OH)_4 molecules obtained from the QM/MM simulation (b), and their comparison with respect to the displacement of the Si atoms and the number of hydrogen bonds with Lsi1 (c).

7. The authors should clarify why two different cellular systems were used for the functional assays. Are there any differences that could arise from the choice of method that makes the results difficult to compare with each other? The functional studies of silicic acid transport through gain-of-function mutants should be also complemented with control experiments showing that the respective aquaglyceroporin is fully functional in the assay used, i.e. is able to transport glycerol. Moreover, the water and glycerol permeability of Lsi1 and, when relevant for the selectivity discussion, Lsi1 mutants should be tested.

(Response)

There is no specific reason for using two different cellular systems for the functional assays. Both of them are widely used in transport activity assay. We used these different systems in two different labs and basically obtained similar results.

However, in response to the reviewer's comments, we further determined the glycerol transport activity of Lsi1 and aquaglyceroporins (GlpF, PfAQP, and AQPM) in *Xenopus* oocytes (Fig. #11). Wild type Lsi1, GlpF, and PfAQP, showed transport activities for glycerol, whereas AQPM was not functional in the oocyte. Western-blotting showed no expression of the PfAQP mutant in the oocyte (Fig. #11a,b). Therefore, the functional studies of silicic acid transport through gain-of-function mutants and their discussion are allowed in GlpF only. While the results of gain-of-function mutants were consistent in both cellular systems (Fig. #11g, h), the results of the other aquaglyceroporins may not be relevant. Moreover, a combination of the functional assay and western-blotting suggested that a direct comparison between the different AQPs is not valid due to the different expression levels. However, in GlpF, the wild type and the Lsi1-type mutant had a similar expression level, whereas the mutant showed the lower transport activity for water and glycerol. This is likely due to the increased hydrophilicity (and size) in the selectivity filter. Notably, the mutant showed higher transport activity for $\text{Ge}(\text{OH})_4$ and $\text{As}(\text{OH})_3$ than the wild-type GlpF, consistent with the assay with insect cells.

We shortened the discussion regarding the Lsi1-type mutants of PfAQP and AQPM and inserted the following text in the revised manuscript.

"We determined glycerol transport activity of Lsi1 and aquaglyceroporins (GlpF, PfAQP, and AQPM) in *Xenopus* oocytes to verify they are fully functional in the assay used. Wild type Lsi1, GlpF, and PfAQP, showed transport activities for glycerol, whereas AQPM was not functional in the oocytes. Successive western blotting showed that the PfAQP mutant was not expressed in the oocyte. Therefore, the functional study of silicic acid transport through gain-of-function mutants and discussion is allowed in GlpF only."

We also determined the water and glycerol permeability of Lsi1 and the mutants in which the selectivity filter was manipulated to be GlpF-type/hydrophobic. They include the full GlpF-type mutant (T65G/G88W/S207G/G216F), the partial GlpF-type mutant (T65G/G88W), and a hydrophobic mutant (S207I). All mutants showed partially/completely abolished transport activity compared to the wild-type (Fig. #12). This suggested that these mutations render the selectivity filter hydrophobic, but they can also disrupt the channel. Consistent with this notion, the full GlpF-type mutant became unstable in the detergent micelle (Fig. #13).

We added the following text in the revised manuscript.

"We also examined the effect on the Lsi1 glycerol permeability by converting the selectivity filter into a hydrophobic one. We determined water and glycerol transport activity of wild-type Lsi1 and the full GlpF type mutant (T65G_{TM1}, G88W_{TM2}, S207G_{TM5}, and G216F_{LE1}), the partial GlpF type mutant (T65G_{TM1} and G88W_{TM2}), and a hydrophobic mutant S207I_{TM5}. All mutants showed partially/completely abolished transport activities compared to wild-type. These results suggest that these mutations render the selectivity filter hydrophobic, but they can also disrupt the channel."

Finally, we re-examined the transport activity of Lsi1 for glycerol by using high

concentration (170 mM) instead of low concentration used in our previous study (Ma *et al.*, Nature, 2006). We found that Lsi1 is also permeable to glycerol at high concentration (Figs. #11, 12). As glycerol is quite flexible with several conformations that can mimic hydroxylated metalloids, there is no surprise that Lsi1 permeates glycerol through the wide and hydrophilic pore.

Based on additional experiment results, we have revised the manuscript accordingly.

Fig. #11 Effect of the selectivity filter mutations of aquaglyceroporins on the transport activity for glycerol, water, Ge, and As.

a and **b**, Western blotting and CBB staining of SDS-PAGE with *Xenopus* oocytes used in the experiments (**c-h**). **c-h**, Transport activity of glycerol (**c** and **d**), water (**e** and **f**), Ge (**g**), and As (**h**) in *Xenopus* oocytes. The substrate uptake experiments were carried out in a

high concentration gradient for **c** and **e** (170 mM glycerol and 5-fold diluted MBS) and in a low concentration gradient for **d** and **f** (90 mM glycerol and 2-fold diluted MBS). In (**c**, **e**, **g**, and **h**), different letters above the columns indicate statistically significant differences at $P < 0.05$ by Tukey-Kramer's test. Values are means \pm s.d. ($n = 4$). In (**d** and **f**), * $P < 0.05$ (two-tailed paired Student's t -test).

Fig. #12 Effect of aquaglyceroporin mimic mutations of Lsi1 on the transport activity for glycerol, water, Ge and As.

Transport activity of glycerol (**a**), water (**b**), germanic acid (**c**) and arsenite (**d**) in *Xenopus* oocytes. In (**a-d**), different letters above the columns indicate statistically significant differences at $P < 0.05$ by Tukey-Kramer's test. Values are means \pm s.d. ($n = 4$).

Fig. #13 FSEC analysis of aquaglyceroporin mimic Lsi1 mutants.

FSEC analysis of S207I (a), T65G+G88W (b), T65G+G88W+S207G+G216F (c) Lsi1 mutants used for the transport activity measurement.

Minor comments:

8. The authors state that the higher number of water molecules in the extracellular vestibule is important for the dehydration of silicic acid to be transported through the channel. However, it is not clear what they are comparing, is it the number of water molecules in the extracellular and cytoplasmic vestibule? If so, how does this compare to for example aquaglyceroporins? Also, how does this fit with AQPs being bidirectional transporters, passively transporting solutes along its concentration gradient in any direction?

(Response)

Sorry for the unclear description, but we meant to say that the larger number of water molecules bound at the extracellular vestibule, compared to the cytoplasmic side, likely ensures the removal of hydrated water molecules from silicic acid. In the MD simulation, we found that silicic acid bound to the vestibule and was dehydrated (Fig. #14). We are not sure that this is relevant to aquaglyceroporins; however, the PfAQP structure paper (Newby *et al.*, Nat Struct Mol Biol., 2008) has made a similar discussion. As the reviewer pointing out, AQPs are bidirectional transporters, which is also true for Lsi1. However, the hydrophilic extracellular vestibule found in the Lsi1 structure seems beneficial for transporting silicic acid into the cytoplasmic side.

We modified the text in the revised manuscript in the following and cited the paper about PfAQP (Newby *et al.*, Nat Struct Mol Biol., 2008). “The large number of water molecules bound at the extracellular vestibule likely ensure the removal of hydrate water molecules from silicic acid at the extracellular side (Fig. 2d). Newby et al. have made a similar discussion in the PfAQP structure (Newby et al., Nat Struct Mol Biol., 2008).”

Fig. #14 Silicic acid binding sites in the extracellular vestibule.

Three silicic acid binding sites in the extracellular vestibules indicated by the MD simulation. View direction is from the extracellular side onto the pore. The blue dots represent the Si atoms of $\text{Si}(\text{OH})_4$ found in the MD simulation in 0-300 nsec.

9. Several of the figures would benefit from the size of the labels being reduced, for example Fig1d and Fig 4.

(Response)

We carefully checked figures and reduced the labels' size in some figures, including Fig. 1d and Fig. 4 in the revised manuscript as suggested.

10. The text refers to Figure 2e in a couple of places, I assume this should be 2d.

(Response)

We carefully read the text and corrected them in the revised manuscript. We apologize for our careless mistakes.

11. The refinement statistics for the highest resolution shell should be given in Table 1. Also, the entry for multiplicity does not look right. Why is given as a range and what is the number for the highest resolution shell? The table also gives average B-factors for ligands but as far as I can tell these are not mentioned in the manuscript. Any ligands built should be included in methods section, even if they are not important for the conclusions.

(Response)

We carefully checked Table 1 and corrected them in the revised manuscript as suggested. We apologize for our careless mistakes. Concerning the ligands, we have added the following text in the method section of the revised manuscript.

“The two Lsi1 tetramers contained eight sodium ions, four OGs, two CHSs, and eleven PEGs.”

12. The manuscript would benefit from being read by a native English speaker.

(Response)

In response to the reviewer's comments, we have asked a native English speaker to check through our manuscript.

REVIEWER COMMENTS

Reviewer #1 (Remarks to the Author):

The authors have submitted a revision of their manuscript documenting the high resolution structure of a protein from the NIP subfamily of aquaporin channels. As noted in the previous review, the work is impactful since it documents a structure that has diverged from classical aquaporins, and that shows how structural aquaporin evolution in plants has led to a new and critical function in metalloid (silicic acid) nutrient transport. The main shortcoming of the previous work lay in the lack of detail or quantitative assessment regarding how this novel structure permits silicic acid permeability.

In their revision, the authors have included computational analysis (QM/MM and molecular dynamics simulations) to attempt to address the lack of functional evidence for silicic acid transport. While this a positive step forward, the analysis is not presented or documented in adequate detail in their manuscript, and the proposed permeation pathway (both water and silicic acid) needs to be evaluated more rigorously and described more clearly. In some places, as noted in the comments below, the functional significance of some of their structural observations seem over interpreted. Finally, the functional data, particularly the new data in the revised manuscript, needs to be reevaluated, clarified, and shortened, and the significance of the work, not only to silicic acid biology but also to plant aquaporin evolution, needs to be discussed.

Specific Comments:

1. In the original review, questions about the steps and rationale that the authors took in the design of their truncated mutant for expression and crystallography were raised. In the revision, they state that they generated these mutants to increase thermal stability and to reduce surface entropy. How did they identify that these were issues, and how did they identify which mutations to make? Did they do this randomly or were these mutations rationally designed? This point is not adequately resolved in the revised manuscript.

2. The description of some of the structural features of Lsi1 that lead to a difference in TM helical tilt, and how this leads to Si selectivity (lines 94-114 in the revised manuscript), are still not clear. The authors do describe interesting and conserved substitutions within the TMs that change the helical packing of Lsi1 compared to other AQP and GlpF structures, but it is still unclear how this would lead to Si selectivity. This was raised in the original review (pt. 6) but the author's response was to change the phrase "is associated with the high selectivity of Lsi1 for silicic acid" to "can be associated..." (line 114). This is not a substantial revision, and the connection (or even a hypothesis) between the unique helical structural properties discussed in this section and Lsi1 Si selectivity remains unclear.

Conversely, the authors note differences in the selectivity filter residues between Lsi1 and AQP1 and GlpF (lines 95-101), but do not describe how these SF changes leads to global conformational changes and altered helical tilt.

They have done modeling and docking of a silicic acid molecule in the selectivity filter (Fig. 6D). Can they interpret this model and connect it to the structural observations noted above to provide functional significance for these structural features?

3. Lines 128-133. The authors assert that a larger (larger compared to what?) number of water molecules likely ensure dehydration of silicic acid entering the channel. Comments about the energetics regarding Further they claim that hydrophobicity and fewer (fewer compared to what?) waters promote rehydration and silicic acid release. This is seems conjectural and structural or computational evidence is not sufficiently provided to support these functional statements. However, the authors seem to have the tools to investigate water dynamics and behavior using molecular dynamics approaches. This needs to be done to provide quantitative assessment and a test of these hypotheses.

4. On lines 143-145 the authors state that the most striking feature of the Lsi1 crystal structure is the larger number of water molecules in the extracellular half of the channel that are not in single file. But they do not comment on why this is striking other than it is different than other AQP structures. What is the predicted significance of this feature? Have they investigated water permeation by steered molecular dynamics to determine how this affects water flow through the channel? This might reveal more about the structural idiosyncrasies of the Lsi1 channel. Outside of the fact that the SF is wider, what is the significance of waters aligned in a non single file manner? As noted in point 3, MD simulations of water permeability could provide insight.

5. In other areas of the manuscript, functional characteristics are attributed to structural features of Lsi1 without adequate support. For example:

- They note a distortion in the pore structure of Lsi1 and assert that this prevents the permeation of silicic acid oligomers. (Lines 154-156).
- Lines 177-180: "The water molecules of the additional polar face likely act as hydrogen bond acceptors during substrate transport..." There is no structural or functional support for this statement.
- Lines 188-190. Based on the novel structure of the Lsi1 SF, the authors propose that Thr65 in TM1 as well as a pore water molecule (water 9) "may play a role in the specificity of transport substrate." (line 201). However substitution of Thr65 for Ala and Gly, which would remove the hydrogen bond capability

of this position results in no change in silicic acid transport properties. While the authors suggest that additional water molecules compensate for this property (lines 192-3), no evidence in support of this statement is provided.

- The function of Threonine 181 in transport does not emerge from the functional data (see point 6).
- As noted in point 7 below, there are questions regarding the computational approaches and the conclusions drawn that need to be clarified.

6. The new functional data provided by the authors in the revised manuscript are confusing in places and do not contribute adequately to the study as written and presented. This part of the manuscript would be improved by shortening, rewriting, provision of key expression controls, and discussion of the results in terms of their structural model and predicted transport pathway in the Discussion section. Specific issues include:

- Differences in transport activity in the assays employed by the authors could be due to interesting factors (structural differences between the proteins) or less interesting factors (misfolding or differences in protein expression or trafficking). It is important to control for these factors where possible. While the authors present Western blot analysis for some assays to demonstrate protein expression (e.g., Supplemental Fig. 7), it is missing from other analyses (Fig. 5 and Supplemental Fig. 8). This is an important control to demonstrate that differences in activity are not due to expression or trafficking/folding issues.
- A series of mutations at the two Thr positions (Thr 65 TM1 and Thr 181 TM4) were generated and analyzed in *Xenopus* oocytes or insect Sf9 cells. On lines 208-210, the authors argue that “the specificity of the transport substrate is modified by manipulating Thr65 and Thr181”. While the data support this to some degree for Thr65, most substitutions of Thr181 (including hydrophobic valine) have no effect on Ge transport. It is not clear from the structure and the manuscript text what function Thr 181 has in the transport pore, and the functional data are not compelling. This data could be shortened and placed in the supplemental figures, or removed.
- The authors carry out gain of function Si transport analyses with three selected aquaglyceroporins, GlpF, PfAQP and AQPM (Lines 212 to 236; Fig. 5 panel e,f, and Supplemental Fig. 7). The most interesting observation is engineering the Lsi1 selectivity filter on GlpF results in gain of silicic acid transport capability. The results with the other two proteins (PfAQP and AQPM) were inconclusive as it was not clear that the mutants were functional proteins. The data would be strengthened by removing the inconclusive results (i.e., all PfAQP and AQPM data), providing Western blot controls in both *Xenopus* and Sf9 cells for all GlpF and Lsi1 wild type and mutant constructs.

- Similarly, the glycerol permeability of Lsi1 wild type and mutants (lines 238-243) provided some new questions that need to be resolved. In the original analysis of Lsi1 (Ma et al., ref. 6), glycerol permeability was not observed for Lsi1. Based on the structural features of the Lsi1 channel in the present study, the authors hypothesize that this is the result of the two polar faces of the channel compared to the amphipathic feature and greater hydrophobicity of the glyceroporin SF. To test this several mutations of Lsi1 to introduce glyceroporin-like substitutions were generated and tested.

However that data are conflicting with previous results, and mutant studies are inconclusive. For example, by using an indirect assay (oocyte swelling) it appears as if Lsi1 now exhibits strong glycerol transport that is not statistically different from the bacterial glycerol channel GlpF (Supplemental Fig.7 panel c). The authors attribute this to the high concentration of glycerol used, but it could also be due to the nature of the assay (oocyte swelling) which is less quantitative than using radiolabeled substrates. Moreover, previous work with human AQP10, which has a similar SF as Lsi1, shows that it is a strong glycerol transporter (ref. 25). In all cases, the Lsi1 mutants have lower glycerol transport than wild type (Supplemental Fig. 8) which the authors attribute to “disruption of the channel” (line 243). Western blot data for expression are not included. In short, this data does not add any new information to Lsi1 specificity for glycerol, and presents data that are at potential odds with previous observations. If glycerol permeability is a critical part of the Lsi1 story, than much more analysis is required. However, I question whether this data is necessary and their inclusion in the present manuscript detracts from the focus on the most important functional biological question: why does Lsi1 transport silicic acid?

- Some assays were done in insect cells and others in Xenopus. While it is understood that both are suitable systems for AQP functional analyses, why were two different systems used? In particular, it seems as if insect cells are more desirable since this is the same system used to produce the protein used for structure elucidation. At the very least, the authors need to clarify why two systems were used and provide a statement or evidence that functional results were equivalent in both systems.

7. As pointed out in the previous review, the authors were not able to obtain structures of Lsi1 with any transport substrates. Based on the suggestions of both reviewers, the authors have used QM/MM modeling and MD simulation (300 ns) to computationally investigate potential modes of silicic acid interaction with the channel, and putative pathways of transport. As noted below, these provide some interesting observations, but details about how these measurements were done, as well as questions regarding what their observations reveal about the permeation pathway, remain.

- The authors make several assertions about the positions of waters in the pore and their roles in transport (lines 126-151, 267 to 300, see also comments in points 3 to 5 above). They should use steered MD with their hydrated structure to investigate the potential pathway and energetics of water movement, and whether the potential properties of water in the unique regions of the Lsi1 pore behave

as they predict. The analysis should provide snapshots of the simulation to provide information on the water permeation trajectory, as well as potential of mean force simulations to identify energy barriers. This seems to be important preliminary step before replacing the water molecules with silicic acid and carrying out a similar MD simulation to investigate Si permeation. Similarly, this may provide information that is useful in the substitution of water with silicic acid for QM/MM and SF docking analyses.

- The MD simulation lacks details the trajectory of silicic acid movement and the description of the results are difficult to follow from a functional perspective. What are the positions of Si molecules at the beginning and end of the simulation? What is the potential trajectory and permeation path of silicic acid, did a permeation event take place during the simulation? The authors conclude that “silicic acid permeated the channel during the simulation, consistent with our proposed mechanism.” A clearer description of the proposed mechanism is needed, as well as snapshots of the substrate and water dynamics during the simulation. The authors note “bottlenecks”, but it is not clear what these represent. Are these barriers to transport? Does silicic acid move past these bottlenecks during their MD simulation? If waters occupy these positions, what are their roles in transport? A MD simulation of water transport, as noted in the previous point, would help determine the dynamics of water in the pore.

With respect to the pathway of transport, why did the authors not consider steered molecular dynamic approaches as noted above? These have been widely used for other AQP structures and could provide details on water and silicic acid movement and permeation trajectories in the pore.

- For the QM/MM, how were silicic acid molecules docked into the pore for these measurements? It seems as if this was done by replacing water molecules in the channel with silicic acid, but the methodology used is not clear. The authors do provide a fairly detailed structural model for silicic acid bound to the selectivity filter (Fig 6b) which is interesting and provides reasoning for the location of specific residues within the SF. They should provide some distances between ligand and protein atoms as well as potential hydrogen bonds so we can see clearly whether the ligands fit well.

- The authors state (lines 260-1) “we verified the modeled silicic acid locations by using QM/MM calculations.” As written, this seems to be an over-statement, since it is not clear how QM/MM “verified” the position of silicic acid in the absence of structural data. In general, the description of the Si on lines 266 through 283, while representing a plausible scenario for silicic acid interaction with the channel, seems to be one of perhaps other interpretations. Strong conclusions seem to be made based on limited information (e.g., lines 278 to 280: “This assignment allowed us not only identifying hydrogen bonding partners of the silicic acids but also to determine orientations of the four positions.”). This should be rewritten to underscore the hypothetical nature of the model.

8. Some assertions made by the authors are not accurate. Examples include:

- In Lines 157-9. The authors note two additional constrictions from glycine carbonyls in Lsi1, and state that these “is unusual since in the structures of other AQPs the selectivity filter constitutes a single narrowest constriction”. A review of the available AQP structures (e.g., AQP0 and others) shows that this statement is not accurate, and that other areas of constriction occur on a case by case basis in AQP structures.

- Lines 166-168. The authors claim that the feature of a fifth residue (Thr 65 in TM1) of the Lsi1 selectivity filter “has not been recognized since in other AQPs bulky hydrophobic residue in TM2 shields the residue equivalent to Thr65). This is actually not accurate since, as the authors note later in the manuscript, the orientation of the Lsi1 selectivity filter is remarkably similar to the arrangement observed in the structure of human AQP10 in which glycine substitutions and a threonine at an equivalent position of TM1 results in a similar wide and hydrophilic SF (the authors note this on lines 181-82, 365-7 and show the similarities of the Lsi1 and hAQP10 SF in Fig. 3).

9. In the revised manuscript, the authors have expanded the discussion section as suggested in the original review, but elements of Lsi1 and its unique functions and evolution should be discussed. Instead, the authors provide detailed analysis of Lsi1 compared to two distant AQP family members in plants (SoPIP2;1 and AtTIP2;1). While this is interesting, they have already covered the differences between Lsi1 and other AQP and glyceroporins thoroughly in the results section and it is unclear why they added two new ones here. They thoroughly describe the differences but do not present the significance of this discussion from a larger perspective of aquaporin structure and evolution in plants.

10. The discussion of Human AQP10 seems more relevant, given the similarities in the two structures, and the fact that both Lsi1 and AQP10 both transport silicic acid. Is this an example of convergent evolution? AQP10 has a pH gate, is this lacking in Lsi1? As a final note, (lines 380 to 381) the authors suggest that “The selectivity for silicic acid of hAQP10 may be lower than Lsi1...” There is no evidence to support this statement.

11. Minor comments and suggestions:

- In figure 1a, please include the van der Waals volumes for each solute.
- In Figure 1d indicate key residues of Ar/R perhaps with shading.

- In supplemental figure 4, please list the backbone rmsd for each superposed structure under each figure in the panel.
- On line 105 indicate which TMs are being referred to.
- Line 124, arginine would not be considered to be a small residue.
- Line 398, do the authors mean “tetrahedral”?
- Have the authors tried crystallography trials in glycerol?
- I did not see a description or citation of the FSEC technique in the Methods.
- Line 488, details of gene synthesis are missing.
- Line 511 “mMESSAGE mMACHINE” not “mMESSAGE mMACHINE”.
- Line 513, “MBS” define abbreviation, here and throughout.
- Fig. 1d, the significance of cross subunit interactions is still missing in the revision.
- Fig. 2A, what do the angstrom annotations represent? Please clarify in the figure legend. Fig. 2e, please indicate the position of the distortion from figure 2c in the Lsi1 image in this panel. What feature of the structure causes this distortion? Indicate the glycerol molecule in GlpF in the legend. The positions of the SF and NPA regions on the structures in Fig. 2e would be helpful.
- Fig. 4, the authors may wish to show the disposition of Thr 181 in an added panel since they discuss it as a symmetry related residue w/r Thr 65.
- Line 795, legend Fig. 6, I do not see blue spheres.

- Fig. 8b, please include the other residues of interest discussed for Lsi1 in this comparison table (e.g., Thr 181).
- Supplemental Fig. 6 In legend, “numbered” instead of “labeled”. Note in legend which residues are shown in the channel.
- Supplemental Figure 9, show close up and superimposition of panels a and b. List the hydrogen bond distances.
- Supplemental Fig. 12 and 13 may not be necessary.

Reviewer #2 (Remarks to the Author):

The authors have substantially improved the manuscript and have adequately responded to concerns and comment raised in my previous review.

My only remaining concern is regarding the oocyte functional assay where the authors state that use Western blots to compare expression of the different constructs. In Supplementary Figure 7 it looks like Lsi1 clearly expresses at a lower level than other constructs (except for the PfAQP mutant), yet this is not taken into consideration when interpreting the results. The equivalent western blot should also be shown for the oocyte experiments with the different Lsi mutants shown in Fig 5, even this does not show any significant differences.

Ideally, the transport values should be corrected for the expression level, for example by comparing the western blot signal in a quantitative manner. At a minimum, the differences that are visible by eye should be acknowledged and taken into consideration when discussing the results.

Finally, the English language is improved but the manuscript would benefit by careful proof-reading.

Reviewer #3 (Remarks to the Author):

This manuscript reports a high-resolution crystal structure of the silicic acid transporter Lsi1 from rice. The structure, together with the biochemistry experiments and computer modeling, provide valuable insights into the mechanism of the protein operation. This work represents an important step forward in the studies of Lsi1 and similar transport proteins. The authors should address the following issues before the manuscript can be accepted for publication:

Lines 128-133: The authors state that large number of water molecules bound at the extracellular vestibule likely ensure the removable of hydrate water molecule and that fewer water molecules as well as the hydrophobic protein environment at the intracellular vestibule seem beneficial for solute rehydration and release. These claims need more quantitative supports other than the cited Ref. 21, because:

(1) These claims are just based on the crystal structure, which may not represent the natural environment where the protein functions. In fact, in the authors MD simulations, the intracellular vestibule appears to be well solvated, too (Figs. 7d-7e).

(2) One may argue that the hydrophobic protein environment at the intracellular vestibule presents a barrier for solute release.

Lines 212-251: In this section, the authors carried out gain-of-function mutagenesis experiments, aiming to test their hypothesis on Lsi1's high selectivity of silicic acid over the smaller glycerol. Unfortunately, although the results lent support to the reversed selectivity in GlpF, they did not really explain why Lsi1 selects silicic acid over glycerol. Rewriting this section is recommended. (See also the comments at the end of this report, please.)

Also, it will be nice if the authors can summarize in a small table the relative selectivity between silicic acid, glycerol, water, and other relevant molecules by Lsi1 (and by other proteins in comparisons). This will be helpful to researchers in a wider community.

Lines 259-264, 532-549: The QM/MM calculations have a number of problems.

(1) It is not clear if only one or multiple $\text{Si}(\text{OH})_4$ molecules are presented in the pore in each given geometry optimization. My guess is one, but this was not explicit from the descriptions.

(2) For each of the Si1 to Si8 positions, it is not clear from the text if multiple starting geometries were used and if so, did the optimizations converge to the same or equivalent locations and orientations.

(3) Importantly, it appears that only the $\text{Si}(\text{OH})_4$ is treated at the QM level. Consequently, the critical interactions, such as the H-bonds between $\text{Si}(\text{OH})_4$ and its surroundings, are modeled at the MM level.

(4) The presence of pore water molecules may significantly impact the binding pose and affinity of Si(OH)₄, but they were removed.

As such, what one can learn from these QM/MM calculations are really limited. It will be much better to include critical pore residues and pore water molecules in the QM region, but this can substantially increase the computational costs. If a larger QM region is not feasible, it is advised to remove the current QM/MM calculations from the manuscript.

Lines 283-284: The authors report the observation of Si(OH)₄ permeating the channel during the 300 ns classical MD simulations. That is interesting. Can the authors provide more quantitative analyses on this? For example, how frequently does such an event occur? On average how long does it take for a Si(OH)₄ to pass through the pore? Does a Si(OH)₄ enter and exit the pore from the same end? What about water?

Note that due to the employed periodic boundary, there is no driving force for Si(OH)₄ to pass through the channel in the model system. Therefore, the simulations should be understood as equilibration (or solvation) of the protein in the membrane-solvent environment rather than the transport of Si(OH)₄ across membrane by Lsi1.

Line 293: The “calculated population” should be defined.

Lines 333-338: The displacement of Thr157 by Si(OH)₄ as seen in the MD simulations suggests the flexibility of the Thr157 side chain. However, claiming that Thr157 has a role in Si(OH)₄ selectivity is probably an overstatement in the absence of further supporting data.

The MD simulation should be able to provide more useful information about the pore water, which is vital to the Lsi1 operation. For example, on average how many water molecules are in the pore at a given time? Does Si(OH)₄ completely lose its first-solvation-shell water when entering the channel? How often do the water molecules in the Wat3, Wat9, Wat17 binding sites are replaced (e.g., by other water molecules from the bulk)? Does such water exchange correlate with Si(OH)₄ entering/exiting the pore from either side?

Perhaps the authors can simulate the same model system but with glycerol in the solvent and see how glycerol binds into the pore. This should allow direct comparisons between the binding of silicic acid and the binding of glycerol, potentially providing critical tests on the authors' hypothesis on their relative selectivity in Lsi1 (Lines 212-251). Doing so could further improve the manuscript.

Reviewer #1 (Remarks to the Author):

The authors have submitted a revision of their manuscript documenting the high resolution structure of a protein from the NIP subfamily of aquaporin channels. As noted in the previous review, the work is impactful since it documents a structure that has diverged from classical aquaporins, and that shows how structural aquaporin evolution in plants has led to a new and critical function in metalloids (silicic acid) nutrient transport. The main shortcoming of the previous work lay in the lack of detail or quantitative assessment regarding how this novel structure permits silicic acid permeability.

In their revision, the authors have included computational analysis (QM/MM and molecular dynamics simulations) to attempt to address the lack of functional evidence for silicic acid transport. While this is a positive step forward, the analysis is not presented or documented in adequate detail in their manuscript, and the proposed permeation pathway (both water and silicic acid) needs to be evaluated more rigorously and described more clearly. In some places, as noted in the comments below, the functional significance of some of their structural observations seem over interpreted. Finally, the functional data, particularly the new data in the revised manuscript, needs to be reevaluated, clarified, and shortened, and the significance of the work, not only to silicic acid biology but also to plant aquaporin evolution, needs to be discussed.

(Response)

We thank the reviewer for his/her positive comments on the manuscript. We have carefully considered the comments raised by the reviewer and revised our manuscript accordingly. We show our point-by-point responses in the following text, and we sincerely hope that we have adequately addressed the reviewer's concerns.

Specific Comments:

1. In the original review, questions about the steps and rationale that the authors took in the design of their truncated mutant for expression and crystallography were raised. In the revision, they state that they generated these mutants to increase thermal stability and to reduce surface entropy. How

did they identify that these were issues, and how did they identify which mutations to make? Did they do this randomly or were these mutations rationally designed? This point is not adequately resolved in the revised manuscript.

(Response)

Screening mutants enhancing thermostability or decreasing surface entropy are common strategies to improve crystallinity (Derewenda, 2010, Acta Crystallogr D Biol Crystallogr.). We used a homology model of Lsi1 to rationalize the mutations. We screened mutants with good thermostability, expression, and monodispersity and then combined them step-by-step.

2. The description of some of the structural features of Lsi1 that lead to a difference in TM helical tilt, and how this leads to Si selectivity (lines 94-114 in the revised manuscript), are still not clear. The authors do describe interesting and conserved substitutions within the TMs that change the helical packing of Lsi1 compared to other AQP and GlpF structures, but it is still unclear how this would lead to Si selectivity. This was raised in the original review (pt. 6) but the author's response was to change the phrase "is associated with the high selectivity of Lsi1 for silicic acid" to "can be associated..." (line 114). This is not a substantial revision, and the connection (or even a hypothesis) between the unique helical structural properties discussed in this section and Lsi1 Si selectivity remains unclear.

Conversely, the authors note differences in the selectivity filter residues between Lsi1 and AQP1 and GlpF (lines 95-101), but do not describe how these SF changes leads to global conformational changes and altered helical tilt.

They have done modeling and docking of a silicic acid molecule in the selectivity filter (Fig. 6D). Can they interpret this model and connect it to the structural observations noted above to provide functional significance for these structural features?

(Response)

The selectivity filter residues in Lsi1 are fine-tuned by the unique TM helix arrangement, resulting in ideal hydrogen bonds between the silicic acid

molecule and the selectivity filter. These observations indicate that the unique residues of the selectivity filter and the helical arrangement of Lsi1 are linked to the high silicic acid selectivity.

Indeed, superimposed GlpF structure on Lsi1 shows that GlpF's selectivity filter causes steric hindrance with modeled silicic acid molecule, indicating that residues in the selectivity filter of Lsi1 are inevitable changes to transport silicic acid molecules.

Since many residues involve global conformational changes, we may not provide a complete explanation that satisfies the reviewer. However, the changes in the amino acid residues discussed in the manuscript would be one of the factors that contribute to the global conformational changes.

In the revised manuscript, we modified the last sentence of this section as below.

“These results reveal a distinct structure of Lsi1_{cryst} from other AQPs, presumably associated with the high selectivity of Lsi1 for silicic acid since the orientation of TM helices links to the orientation of residues facing the channel.”

3. Lines 128-133. The authors assert that a larger (larger compared to what?) number of water molecules likely ensure dehydration of silicic acid entering the channel. Comments about the energetics regarding Further they claim that hydrophobicity and fewer (fewer compared to what?) waters promote rehydration and silicic acid release. This is seems conjectural and structural or computational evidence is not sufficiently provided to support these functional statements. However, the authors seem to have the tools to investigate water dynamics and behavior using molecular dynamics approaches. This needs to be done to provide quantitative assessment and a test of these hypotheses.

(Response)

We admit that our hypothesis is still conjunctive, and further calculations may strengthen our hypothesis. We can say now that the energetic cost for desolvation/solvation of the transported silicic acid may differ between the intracellular and extracellular vestibules (Fig #1).

However, because this is not the focus of this manuscript, we will do a quantitative assessment in our future studies. We have modified the manuscript avoiding assertive descriptions as described below.

“The extracellular vestibule is hydrophilic and bounds more significant water molecules, whereas the intercellular side is hydrophobic and bounds fewer water molecules. The contrasting environments may affect the energetic barrier to pass the selectivity filter in removing hydrated water molecules or promoting hydration and releasing the transported silicic acid.”

Fig. #1 Desolvation of Si(OH)₄ molecules on the vestibules when Si(OH)₄ enters from (a) the extracellular side and (b) the intracellular side in the 450 ns equilibrium MD simulation. Black line shows the trajectory (the z coordinate) of Si(OH)₄ (measured in the left axis). Blue and cyan lines show numbers of H-bonded water molecules and H-bonded amino acids of the protein, respectively (measured in the right axis). The vestibules and channel region are colored in light green and khaki, respectively.

4. On lines 143-145 the authors state that the most striking feature of the Lsi1 crystal structure is the larger number of water molecules in the extracellular half of the channel that are not in single file. But they do not comment on why this is striking other than it is different than other AQP structures. What is the predicted significance of this feature? Have they investigated water permeation by steered molecular dynamics to determine how this affects water flow through the channel? This might reveal more about the structural idiosyncrasies of the Lsi1 channel. Outside of the fact that the SF is wider, what is the significance of

waters aligned in a non single file manner? As noted in point 3, MD simulations of water permeability could provide insight.

(Response)

Lsi1 has evolved to pass through silicic acid selectively and thus has acquired a broader and more hydrophilic selectivity filter than canonical AQP family proteins. This evolution likely resulted in a loss of the non-single file of water molecules in the selectivity filter region. The well-oriented single-file water in canonical AQP family proteins prevents proton transfer via the grotthuss mechanism (de Groot and Grubmüller, 2001, *Science*; Tajkhorshid, 2002, *Science*, Kosinska Eriksson *et al.*, 2013, *Science*). In addition, strongly correlated movements of the well-oriented single-file water prevent fast proton transfer (Kosinska Eriksson *et al.*, 2013, *Science*). Since the water molecules in the selectivity filter region of Lsi1 are not single-file, the mechanism that prevents fast proton transport seems to be lost in Lsi1. We hypothesized that Lsi1 might have acquired its proton exclusion mechanism in the selectivity filter region to compensate for the loss of the proton exclusion function in the selectivity filter region. We investigated the hydrogen-bonding network of water molecules in the structure and found that our crystal structure captures the state in which disable proton transport via the grotthuss mechanism. Since Wat9 orientation is fixed by Lsi1 in a direction that prevents it from receiving protons from other water molecules, Wat4, 5, and 6 in the selectivity filter are not connected to Wat7 and 8, which are located on the NPA region side, by a defect-free hydrogen-bonding network. Therefore, the proton transport by the grotthuss mechanism is blocked. This may be a new mechanism to partially compensate for the lack of proton transport inhibition caused by acquiring the broader and more hydrophilic selectivity filter.

We analyzed the MD water permeation, but what we could say from it is that water molecules permeated the channel very frequently compared to silicic acid. While steered molecular dynamics may determine how the waters affect water flow through the channel, we would like to do it in our future studies.

We have inserted the following sentences to show the predicted significance of water molecules. We thank the reviewer's comment by which we can finally reach an idea that prevents proton transfer in Lsi1. "In chain A, three water

molecules in the selectivity filter (Wat4, Wat5, and Wat6) and two water molecules beneath it (Wat7 and Wat8) are well separated ($3.4\text{-}\text{\AA}$) along the channel, which no longer contributes to the hydrogen bond interactions. Since Wat9 donates hydrogen bonds to the carbonyls, it cannot transfer protons to water molecules Wat5 and Wat8. The breakage of the connectivity between selectivity filter waters and other water molecules may prevent proton translocation via the grotthuss mechanism. Strongly correlated movements of the well-oriented single-file water in orthodox AQP family proteins prevent proton transfer via the grotthuss mechanism (Kosinska Eriksson et al., 2013, Science). Since the water molecules in the selectivity filter of Lsi1 are not single-file, Lsi1 may have a different mechanism that prevents fast proton transport.”

5. In other areas of the manuscript, functional characteristics are attributed to structural features of Lsi1 without adequate support. For example:

- They note a distortion in the pore structure of Lsi1 and assert that this prevents the permeation of silicic acid oligomers. (Lines 154-156).

(Response)

We admit that rigorous experiments may be necessary to clarify our hypothesis. However, we would like to keep a possible function inferred from the structure in the present manuscript. We hope that the research community may reveal it by presenting such a possibility. We have modified the manuscript as below:

The shift of HE provides this channel’s distortion which “could” prevent the transport of larger substrates...

- Lines 177-180: “The water molecules of the additional polar face likely act as hydrogen bond acceptors during substrate transport...” There is no structural or functional support for this statement.

(Response)

Thank you for your comments, but we have explained the structural basis for Wat3 and Wat9 acting as hydrogen bond acceptors during silicic acid transport in the manuscript. We have explained how Wat3 and Wat9 orient their oxygen atoms toward the channel lumen (Lines 175-183) and how these water molecules make hydrogen bonds with the modeled silicic acid molecules (Lines 257-261). The statement is inferred from the high-resolution crystal structure, which is also consistent with the MD snapshots (Supplementary Fig.10).

- Lines 188-190. Based on the novel structure of the Lsi1 SF, the authors propose that Thr65 in TM1 as well as a pore water molecule (water 9) “may play a role in the specificity of transport substrate.” (line 201). However substitution of Thr65 for Ala and Gly, which would remove the hydrogen bond capability of this position results in no change in silicic acid transport properties. While the authors suggest that additional water molecules compensate for this property (lines 192-3), no evidence in support of this statement is provided.

(Response)

We admit that there is no evidence for the additional water molecules compensating the property of Thr65, and there may be other interpretations. Therefore, we avoided assertive description and made a most reasonable explanation inferred from the structure and functional analysis results. We would like to point out that water molecule often compensates for an effect of the mutation. Please see the Asp to Cys mutation in photosystem II (Kuroda et al., 2021, BBA Bio).

We have modified the revised manuscript as below:

“One possible interpretation is” that a polar environment made by hydroxyl...

- The function of Threonine 181 in transport does not emerge from the functional data (see point 6).

(Response)

Please see the response to point 6.

- As noted in point 7 below, there are questions regarding the computational approaches and the conclusions drawn that need to be clarified.

6. The new functional data provided by the authors in the revised manuscript are confusing in places and do not contribute adequately to the study as written and presented. This part of the manuscript would be improved by shortening, rewriting, provision of key expression controls, and discussion of the results in terms of their structural model and predicted transport pathway in the Discussion section. Specific issues include:

- Differences in transport activity in the assays employed by the authors could be due to interesting factors (structural differences between the proteins) or less interesting factors (misfolding or differences in protein expression or trafficking). It is important to control for these factors where possible. While the authors present Western blot analysis for some assays to demonstrate protein expression (e.g., Supplemental Fig. 7), it is missing from other analyses (Fig. 5 and Supplemental Fig. 8). This is an important control to demonstrate that differences in activity are not due to expression or trafficking/folding issues.

(Response)

Although we added new functional data in response to both reviewer's concerns in the previous manuscript, we agree that some do not contribute adequately to the study. Thus, we have shortened and rewrote this part in the revised manuscript. We have included Western blot analysis as Supplementary Fig. 7 in the revised manuscript. In addition, we have investigated effects of the expression or trafficking/folding issues by FSEC and found that it does not affect the conclusion. The important point is that we tried to test whether these proteins have transport activity or not, but did not compare the transport activity quantitatively based on the protein level.

- A series of mutations at the two Thr positions (Thr 65 TM1 and Thr 181 TM4) were generated and analyzed in *Xenopus* oocytes or insect Sf9 cells. On lines

208-210, the authors argue that “the specificity of the transport substrate is modified by manipulating Thr65 and Thr181”. While the data support this to some degree for Thr65, most substitutions of Thr181 (including hydrophobic valine) have no effect on Ge transport. It is not clear from the structure and the manuscript text what function Thr 181 has in the transport pore, and the functional data are not compelling. This data could be shortened and placed in the supplemental figures, or removed.

(Response)

The community of AQPs has believed the selectivity filter mainly determines the substrate specificity since it creates the narrowest part of the channel. However, other factors have not been considered extensively. The significance of the selectivity filter is confirmed in the Lsi1 structure by the mutations of Thr65_{TM1}. Thr181 is a unique residue in a pseudo-c2-symmetrically related to Thr65_{TM1}, and the fact that T181Q mutation changes the substrate specificity is a novel finding in the community. Therefore, we would like to keep the descriptions and functional assays for Thr181.

Since the function of Thr181 is not clear and functional data are not compelling, we modified the manuscript text of this section as below:

“It should be noted that the effect of Thr181 mutations is moderate compared to Thr65_{TM1} and its function is still not clear. Nevertheless, the results suggest that the specificity of transport substrate is modified by manipulating the two Thr residues identified in the present study.”

- The authors carry out gain of function Si transport analyses with three selected aquaglyceroporins, GlpF, PfAQP and AQPM (Lines 212 to 236; Fig. 5 panel e,f, and Supplemental Fig. 7). The most interesting observation is engineering the Lsi1 selectivity filter on GlpF results in gain of silicic acid transport capability. The results with the other two proteins (PfAQP and AQPM) were inconclusive as it was not clear that the mutants were functional proteins. The data would be strengthened by removing the inconclusive results (i.e., all PfAQP and AQPM data), providing Western blot controls in both *Xenopus* and Sf9 cells for all GlpF and Lsi1 wild type and mutant constructs.

(Response)

We have removed inconclusive results in Supplementary Fig. 7 in the previous manuscript but would like to keep the results of wild-type PfAQP and AQPM in Fig. 5. This study is the first report on PfAQP and AQPM transport activities for silicic acid and arsenite, and they are functional with a similar expression level in Sf9 cells.

We have modified the manuscript text to avoid confusion and included Western blot controls in the new Supplementary Fig.7 in the revised manuscript, based on the reviewer's comment.

- Similarly, the glycerol permeability of Lsi1 wild type and mutants (lines 238-243) provided some new questions that need to be resolved. In the original analysis of Lsi1 (Ma et al., ref. 6), glycerol permeability was not observed for Lsi1. Based on the structural features of the Lsi1 channel in the present study, the authors hypothesize that this is the result of the two polar faces of the channel compared to the amphipathic feature and greater hydrophobicity of the glyceroporin SF. To test this several mutations of Lsi1 to introduce glyceroporin-like substitutions were generated and tested.

However that data are conflicting with previous results, and mutant studies are inconclusive. For example, by using an indirect assay (oocyte swelling) it appears as if Lsi1 now exhibits strong glycerol transport that is not statistically different from the bacterial glycerol channel GlpF (Supplemental Fig.7 panel c). The authors attribute this to the high concentration of glycerol used, but it could also be due to the nature of the assay (oocyte swelling) which is less quantitative than using radiolabeled substrates. Moreover, previous work with human AQP10, which has a similar SF as Lsi1, shows that it is a strong glycerol transporter (ref. 25). In all cases, the Lsi1 mutants have lower glycerol transport than wild type (Supplemental Fig. 8) which the authors attribute to “disruption of the channel” (line 243). Western blot data for expression are not included. In short, this data does not add any new information to Lsi1 specificity for glycerol, and presents data that are at potential odds with previous observations. If glycerol permeability is a critical part of the Lsi1 story, than much more analysis

is required. However, I question whether this data is necessary and their inclusion in the present manuscript detracts from the focus on the most important functional biological question: why does Lsi1 transport silicic acid?

(Response)

We agree with the reviewer that mutant studies are inconclusive. We have shortened the text and removed data in the revised manuscript.

- Some assays were done in insect cells and others in *Xenopus*. While it is understood that both are suitable systems for AQP functional analyses, why were two different systems used? In particular, it seems as if insect cells are more desirable since this is the same system used to produce the protein used for structure elucidation. At the very least, the authors need to clarify why two systems were used and provide a statement or evidence that functional results were equivalent in both systems.

(Response)

Another reviewer asked the same question in the first round of the review process. We answered as follows, which convinced the reviewer. "There is no specific reason for using two different cellular systems for the functional assays. Both are widely used in transport activity assay. We used these different systems in two different labs and obtained similar results."

7. As pointed out in the previous review, the authors were not able to obtain structures of Lsi1 with any transport substrates. Based on the suggestions of both reviewers, the authors have used QM/MM modeling and MD simulation (300 ns) to computationally investigate potential modes of silicic acid interaction with the channel, and putative pathways of transport. As noted below, these provide some interesting observations, but details about how these measurements were done, as well as questions regarding what their observations reveal about the permeation pathway, remain.

- The authors make several assertions about the positions of waters in the pore and their roles in transport (lines 126-151, 267 to 300, see also comments in

points 3 to 5 above). They should use steered MD with their hydrated structure to investigate the potential pathway and energetics of water movement, and whether the potential properties of water in the unique regions of the Lsi1 pore behave as they predict. The analysis should provide snapshots of the simulation to provide information on the water permeation trajectory, as well as potential of mean force simulations to identify energy barriers. This seems to be an important preliminary step before replacing the water molecules with silicic acid and carrying out a similar MD simulation to investigate Si permeation. Similarly, this may provide information that is useful in the substitution of water with silicic acid for QM/MM and SF docking analyses.

(Response)

We admit several assertions about water positions during the $\text{Si}(\text{OH})_4$ transport in the previous manuscript. As noted in our response at point 4, we analyzed the water permeation trajectory, showing no bottleneck for the water permeation in the channel. Also, we could not perform the steered molecular dynamics due to increased computational costs. Instead, we have added MD snapshots showing the interactions between water and silicic acid (Supplementary Fig. 10). We extended the QM/MM calculation in which the QM region includes the critical residues and pore water molecules. The new QM/MM calculation agreed with our modeled silicic acid molecules (Supplementary Fig. 9). We hope these new data are satisfactory to the reviewer.

- The MD simulation lacks details the trajectory of silicic acid movement and the description of the results are difficult to follow from a functional perspective. What are the positions of Si molecules at the beginning and end of the simulation? What is the potential trajectory and permeation path of silicic acid, did a permeation event take place during the simulation? The authors conclude that “silicic acid permeated the channel during the simulation, consistent with our proposed mechanism.” A clearer description of the proposed mechanism is needed, as well as snapshots of the substrate and water dynamics during the simulation. The authors note “bottlenecks”, but it is not clear what these represent. Are these barriers to transport? Does silicic acid move past these bottlenecks during their MD simulation? If waters occupy these positions, what are their roles in transport? A MD simulation of water transport, as noted in the

previous point, would help determine the dynamics of water in the pore. With respect to the pathway of transport, why did the authors not consider steered molecular dynamic approaches as noted above? These have been widely used for other AQP structures and could provide details on water and silicic acid movement and permeation trajectories in the pore.

(Response)

We have added the trajectory of silicic acid movement (Supplementary Fig.11) and MD snapshots (Supplementary Fig. 10) in the revised manuscript. We have included the positions of $\text{Si}(\text{OH})_4$ molecules at the beginning and end of the simulation in our response (Fig.#2). We have modified the descriptions for the MD simulation as much as possible based on the reviewer's comment.

The bottleneck represents the locations where silicic acid cannot move freely, and therefore they are presumably the barriers. However, a few silicic acid molecules can move past these bottlenecks and permeated the channel during the simulation. We have described this point in the revised manuscript.

However, we would like to investigate the simulation details, including the steered MD, in our future studies.

Fig. #2 The positions of $\text{Si}(\text{OH})_4$ molecules at the beginning (left) and end (right) of the simulation.

- For the QM/MM, how were silicic acid molecules docked into the pore for

these measurements? It seems as if this was done by replacing water molecules in the channel with silicic acid, but the methodology used is not clear. The authors do provide a fairly detailed structural model for silicic acid bound to the selectivity filter (Fig 6b) which is interesting and provides reasoning for the location of specific residues within the SF. They should provide some distances between ligand and protein atoms as well as potential hydrogen bonds so we can see clearly whether the ligands fit well.

(Response)

As pointed out by the reviewer, we modeled the silicic acid molecules by replacing water molecules in the channel with hydroxy groups of $\text{Si}(\text{OH})_4$. We have included a figure showing some distances between ligand and protein atoms in this response (Fig. #3). We have described the details of the methodology in the revised manuscript based on the reviewer's comment.

Fig.#3 The modeled silicic acid Si2 and its nearby residues.

- The authors state (lines 260-1) “we verified the modeled silicic acid locations by using QM/MM calculations.” As written, this seems to be an over-statement, since it is not clear how QM/MM “verified” the position of silicic acid in the absence of structural data. In general, the description of the Si on lines 266 through 283, while representing a plausible scenario for silicic acid interaction with the channel, seems to be one of perhaps other interpretations. Strong

conclusions seem to be made based on limited information (e.g., lines 278 to 280: “This assignment allowed us not only identifying hydrogen bonding partners of the silicic acids but also to determine orientations of the four positions.”). This should be rewritten to underscore the hypothetical nature of the model.

(Response)

We agree that it was unclear how we verified the QM/MM simulation in the previous manuscript. We have provided more details by comparing our model and the QM/MM simulation in the revised manuscript. We have modified the panels and tables by showing the displacement of the Si atoms, the number of hydrogen bonding interactions, and the superposed structure. In addition, we have modified descriptions in the revised manuscript as much as possible based on the reviewer’s comment.

For instance, a description on lines 278 to 280 in the previous manuscript, “This assignment allowed us not only...” was modified into the following text in the revised manuscript. “This assignment identifies **plausible** hydrogen bonding partners of the silicic acids and defines orientations of the former four positions. **However, there may be other interpretations because we hypothesized based on the structure without any substrates.** The predicted silicic acids....”

8. Some assertions made by the authors are not accurate. Examples include:

- In Lines 157-9. The authors note two additional constrictions from glycine carbonyls in Lsi1, and state that these “is unusual since in the structures of other AQPs the selectivity filter constitutes a single narrowest constriction”. A review of the available AQP structures (e.g., AQP0 and others) shows that this statement is not accurate, and that other areas of constriction occur on a case by case basis in AQP structures.

(Response)

We removed the sentence starting from “This is unusual since...” based on the reviewer’s comment.

• Lines 166-168. The authors claim that the feature of a fifth residue (Thr 65 in TM1) of the Lsi1 selectivity filter “has not been recognized since in other AQPs bulky hydrophobic residue in TM2 shields the residue equivalent to Thr65). This is actually not accurate since, as the authors note later in the manuscript, the orientation of the Lsi1 selectivity filter is remarkably similar to the arrangement observed in the structure of human AQP10 in which glycine substitutions and a threonine at an equivalent position of TM1 results in a similar wide and hydrophilic SF (the authors note this on lines 181-82, 365-7 and show the similarities of the Lsi1 and hAQP10 SF in Fig. 3).

(Response)

The amino acid residue on TM1 has not been recognized as a constituent of the selectivity filter. There are two reasons for this: first, the bulky amino acid residue from TM2 covers the TM1 in canonical AQP structures. Second, the residue corresponding to Thr65 in Lsi1 has been overlooked even after the report of the hAQP10 structure. No one has reported the significance of the residue nor verified its function in substrate permeability before us.

We have modified the following sentence on Lines 166-168 based on the reviewer’s comment. “Significance of Thr65_{TM1} in the selectivity filter has not been recognized since, in canonical AQPs, bulky hydrophobic residue in TM2....”

9. In the revised manuscript, the authors have expanded the discussion section as suggested in the original review, but elements of Lsi1 and its unique functions and evolution should be discussed. Instead, the authors provide detailed analysis of Lsi1 compared to two distant AQP family members in plants (SoPIP2;1 and AtTIP2;1). While this is interesting, they have already covered the differences between Lsi1 and other AQP and glyceroporins thoroughly in the results section and it is unclear why they added two new ones here. They thoroughly describe the differences but do not present the significance of this discussion from a larger perspective of aquaporin structure and evolution in plants.

(Response)

In the first revision, the reviewer pointed out that the discussion section was short, and we should discuss the novelty of Lsi1 in terms of AQP structure and evolution. We were afraid that it would be premature to discuss evolution in this paper because the evolutionary origin of the NIP subfamily is uncertain. Instead, we discussed the structure with distant AQP family members in plants, as requested by another reviewer.

While we are not sure that it is satisfactory to the reviewer, we have included a short discussion regarding the evolution of Lsi1 in the revised manuscript.

“We have shown that the unique TM helix orientations and selectivity filter of Lsi1 are essential for silicic acid transport. Lsi1 has presumably acquired these features during its evolution from canonical AQPs that could not transport silicic acid. The evolution of plant AQP family proteins that permeate substrates other than water and glycerol, such as Lsi1 and AtTIP2;1, seems to have involved drastic modification of SF from canonical AQPs. As described, even between the evolutionarily distant species of human (hAQP10) and rice (Lsi1), a typical structure exists in which water molecules create a polar face in SF to transport silicic acid. Such a common structure has likely evolved from the convergent evolution.”

10. The discussion of Human AQP10 seems more relevant, given the similarities in the two structures, and the fact that both Lsi1 and AQP10 both transport silicic acid. Is this an example of convergent evolution? AQP10 has a pH gate, is this lacking in Lsi1? As a final note, (lines 380 to 381) the authors suggest that “The selectivity for silicic acid of hAQP10 may be lower than Lsi1...” There is no evidence to support this statement.

(Response)

Plants are thought to have acquired the *Lsi1* gene during evolution. Despite the difference in the evolutionary lineage, the mammalian aquaglyceroporin hAQP10 is similar to Lsi1 in its selectivity filter consisting of five amino acid residues and two water molecules. This seems to be a common convergent solution for the different AQP genes to transport larger hydrophilic substrates.

Although hAQP10 has been shown to have a pH gate, there is no report showing a pH gate in Lsi1. We compared the structures of hAQP10 and Lsi1 and found that Lsi1 did not have the same pH gate as hAQP10.

The selectivity filter of hAQP10 is broader and more hydrophobic than that of Lsi1. Therefore, the selectivity filter of hAQP10 may not be identical to that of Lsi1. Since no data are available for comparing them directly, we modified the description (lines 377-381) as follows in the revised manuscript.

“Indeed, the carbonyl of Gly210..., and this rotation **weakens hydrogen bond to waters** and narrows the hallow through which the hydroxy group of the silicic acid passes. **Given that the selectivity filter of Lsi1 is ideal for the silicic acid permeation, selectivity for silicic acid of hAQP10 may be different from that of Lsi1.**”

11. Minor comments and suggestions:

- In figure 1a, please include the van der Waals volumes for each solute.
- In Figure 1d indicate key residues of Ar/R perhaps with shading.
- In supplemental figure 4, please list the backbone rmsd for each superposed structure under each figure in the panel.

(Response)

We have modified the figures in the revised manuscript based on the reviewer's comment.

- On line 105 indicate which TMs are being referred to.

(Response)

“TMs” in the previous manuscript was referred to TMs **between TM1 and TM2, and between TM4 and TM5. We have modified the revised manuscript to identify which TMs were referred.**

- Line 124, arginine would not be considered to be a small residue.

(Response)

We have modified the following text.

However, the selectivity filter of Lsi1 “contains three” small residues (Gly_{TM2}/Ser_{TM5}/Gly_{LE1}/Arg_{LE2})¹⁰.

- Line 398, do the authors mean “tetrahedral”?

(Response)

We have corrected the word into “tetrahedral”.

- Have the authors tried crystallography trials in glycerol?

(Response)

We have tried to crystallize Lsi1_{cryst} in the presence of glycerol but have not yet found suitable conditions for structural analysis.

- I did not see a description or citation of the FSEC technique in the Methods.

(Response)

We have appropriately cited the paper on FSEC technique in the Methods section as reference 15.

- Line 488, details of gene synthesis are missing.

(Response)

We have added details of gene synthesis in the revised manuscript (Line 488, synthesized “(Integrated DNA Technologies).”

- Line 511 “mMESSAGE mMACHINE” not “mMESSAGE mMACHINE”.
- Line 513, “MBS” define abbreviation, here and throughout.

(Response)

We have modified the text in the revised manuscript based on the reviewer's comment.

- Fig. 1d, the significance of cross subunit interactions is still missing in the revision.

(Response)

We have confirmed that the cross-subunit interaction forms a side cavity with three water molecules at the channel entrance, affecting substrate permeation. We are not sure about its significance, and we will investigate it in our future studies.

- Fig. 2A, what do the angstrom annotations represent? Please clarify in the figure legend. Fig. 2e,

(Response)

We have added the following text in the figure legend.

“Arrowheads indicate loop A and blue arrows represent the shift of TM helices (in Å).”.

please indicate the position of the distortion from figure 2c in the Lsi1 image in this panel.

(Response)

As the view directions are different between Fig. 2c and e, we do not show the position of the distortion in Fig. 2e to avoid confusion. Instead, we have shown the rotational relationship between them in the revised manuscript.

What feature of the structure causes this distortion?

(Response)

Since the amino acid residues of the selectivity filter from TM2 and TM5 are small, HE shifts inwards towards the channel. This may be the reason for the distortion of the permeation pathway.

Indicate the glycerol molecule in GlpF in the legend.

The positions of the SF and NPA regions on the structures in Fig. 2e would be helpful.

(Response)

We have modified the figure and its legend based on the reviewer's comment.

- Fig. 4, the authors may wish to show the disposition of Thr 181 in an added panel since they discuss it as a symmetry related residue w/r Thr 65.

(Response)

We acknowledge the reviewer's suggestion, and we have added a new panel in Fig.4 and its legend.

- Line 795, legend Fig. 6, I do not see blue spheres.

(Response)

We have corrected the figure legend in the revised manuscript as follows:

“Water molecules in the channel are shown as spheres and numbered in blue.”

We apologized for the careless mistake.

- Fig. 8b, please include the other residues of interest discussed for Lsi1 in this comparison table (e.g., Thr 181).

(Response)

We have included Thr181 in the figure in the revised manuscript.

- Supplemental Fig. 6 In legend, “numbered” instead of “labeled”. Note in legend which residues are shown in the channel.

(Response)

We have modified the legend based on the reviewer's comment.

- Supplemental Figure 9, show close up and superimposition of panels a and b. List the hydrogen bond distances.

(Response)

We have modified the figure based on the reviewer's comment

- Supplemental Fig. 12 and 13 may not be necessary.

(Response)

We have made the figures to answer another reviewer's comments. Since the reviewer was satisfied with our revision, we would like to keep them in the revised manuscript.

Reviewer #2 (Remarks to the Author):

The authors have substantially improved the manuscript and have adequately responded to concerns and comment raised in my previous review.

My only remaining concern is regarding the oocyte functional assay where the authors state that use Western blots to compare expression of the different constructs. In Supplementary Figure 7 it looks like Lsi1 clearly expresses at a lower level than other constructs (except for the PfAQP mutant), yet this is not taken into consideration when interpreting the results. The equivalent western blot should also be shown for the oocyte experiments with the different Lsi mutants shown in Fig 5, even this does not show any significant differences.

Ideally, the transport values should be corrected for the expression level, for example by comparing the western blot signal in a quantitative manner. At a minimum, the differences that are visible by eye should be acknowledged and taken into consideration when discussing the results.

Finally, the English language is improved but the manuscript would benefit by careful proof-reading.

(Response)

We are delighted to hear that our responses adequately addressed the concerns from the reviewer, and our modifications have substantially improved the manuscript.

We have included Western blot analysis for all mutants in the revised manuscript (Supplementary Fig.7). In addition, we have roughly investigated the effects of the expression or trafficking/folding issues by FSEC and found that it did not affect the conclusion. However, it is not clear from the western blotting whether the difference in expression reflects a functional channel or not. For example, there are two bands of Lsi1 (Supplementary Fig. 8a), but we cannot clarify which one (or both) is functional. Thus, we are afraid that the corrected data from western blotting does not make sense. Instead, we have mentioned the differences in the expression level in the revised manuscript.

For instance, in lines 193-196 of the revised manuscript, we described as below.

“The transport activities for both Ge and As were unaffected by the substitutions of Thr65^{TM1} to Ala, Gly, and Ser (Fig. 5a, b), although expression of the Ala mutant was lower than wildtype or the other two mutants in *Xenopus* oocytes (Supplementary Fig. 7a, b).”

Finally, we have asked a native English speaker to check through our manuscript.

Reviewer #3 (Remarks to the Author):

This manuscript reports a high-resolution crystal structure of the silicic acid transporter Lsi1 from rice. The structure, together with the biochemistry experiments and computer modeling, provide valuable insights into the mechanism of the protein operation. This work represents an important step forward in the studies of Lsi1 and similar transport proteins. The authors should address the following issues before the manuscript can be accepted for publication:

(Response)

First of all, we thank the reviewer for his/her careful reading of our manuscript, insightful and positive comments. We have carefully considered the comments raised by the reviewer and revised our manuscript accordingly.

We give our point-by-point responses in the following text, and we sincerely hope that we have adequately addressed the reviewer's concerns.

Lines 128-133: The authors state that large number of water molecules bound at the extracellular vestibule likely ensure the removable of hydrate water molecule and that fewer water molecules as well as the hydrophobic protein environment at the intracellular vestibule seem beneficial for solute rehydration and release. These claims need more quantitative supports other than the cited Ref. 21, because:

(1) These claims are just based on the crystal structure, which may not represent the natural environment where the protein functions. In fact, in the authors MD simulations, the intracellular vestibule appears to be well solvated, too (Figs. 7d-7e).

(2) One may argue that the hydrophobic protein environment at the intracellular vestibule presents a barrier for solute release.

(Response)

We admit that our hypothesis is still conjunctive, and we agree that quantitative supports may strengthen our hypothesis. We can now say that the energetic costs for desolvation/solvation of the transported silicic acid may differ between the intracellular and extracellular vestibules (Fig #1). However, because this is

not the focus of this manuscript, we will do a quantitative assessment in our future studies. We have modified the manuscript avoiding assertive descriptions as described below.

“The extracellular vestibule is hydrophilic and bounds more significant water molecules, whereas the intercellular side is hydrophobic and bounds fewer water molecules. The contrasting environments may affect the energetic barrier to pass the selectivity filter in removing hydrated water molecules or promoting hydration and releasing the transported silicic acid.”

Fig. #1 Desolvation of Si(OH)₄ molecules on the vestibules when Si(OH)₄ enters from (a) the extracellular side and (b) the intracellular side in the 450 ns equilibrium MD simulation. Black line shows the trajectory (the z coordinate) of Si(OH)₄ (measured in the left axis). Blue and cyan lines show numbers of H-bonded water molecules and H-bonded amino acids of the protein, respectively (measured in the right axis). The vestibules and channel region are colored in light green and khaki, respectively.

Lines 212-251: In this section, the authors carried out gain-of-function mutagenesis experiments, aiming to test their hypothesis on Lsi1's high selectivity of silicic acid over the smaller glycerol. Unfortunately, although the results lent support to the reversed selectivity in GlpF, they did not really explain why Lsi1 selects silicic acid over glycerol. Rewriting this section is recommended. (See also the comments at the end of this report, please.)

(Response)

We have previously shown that Lsi1 transports silicic acid but not glycerol at a low (and more physiological) concentration of 2 mM (Ma *et al.*, 2006, Nature). This study proposed the basis for Lsi1's high selectivity of silicic acid over the glycerol and tried to prove it experimentally. However, the wild-type Lsi1 transported glycerol at a higher concentration (170 mM) of the glycerol, and the mutants showed partially/completely abolished activity for the glycerol. The result suggests that Lsi1 has high selectivity for silicic acid over glycerol, presumably under physiological conditions (or low osmotic pressure), but it transports glycerol at high osmotic pressure. All the mutations did not improve the high substrate selectivity for silicic acid over glycerol or disrupted the channel.

We agree that some data does not adequately explain why Lsi1 selects silicic acid over glycerol. We have shortened and rewritten this part in the revised manuscript based on the reviewer's comment.

Also, it will be nice if the authors can summarize in a small table the relative selectivity between silicic acid, glycerol, water, and other relevant molecules by Lsi1 (and by other proteins in comparisons). This will be helpful to researchers in a wider community.

(Response)

We cannot show the "relative" selectivity because the substrate selectivity has not been investigated quantitatively or investigated in the different assay systems. Instead, we have summarized a table for transport activities as supplementary table 1 in the revised manuscript. We hope this table is helpful to researchers in the broader community.

Table 1 Substrate permeability of AQP family proteins

	Silicic acid	Arsenite	Boric acid	water	glycerol	Ammonia	Reference
Lsi1	+	+	+	+	+/-	?	This study, [1-3]
Mammalian AQP1	?	?	?	+	-	-	[4-5]
Aqy1	?	?	?	+	?	?	[6]
GlpF	-	+	?	+	+	?	This study, [4]
PfAQP	-	+	?	+	+	?	This study, [7]
AQPM	-	-	?	+	+	?	This study, [4]
hAQP10	+	?	?	+	+	?	[8-9]
SoPIP2;1	?	?	?	+	?	?	[4]
AtTIP2;1	?	?	?	+	?	+	[5]

Plus indicates permeable, and minus indicates impermeable.

Lines 259-264, 532-549: The QM/MM calculations have a number of problems.

(1) It is not clear if only one or multiple Si(OH)₄ molecules are presented in the pore in each given geometry optimization. My guess is one, but this was not explicit from the descriptions.

(2) For each of the Si1 to Si8 positions, it is not clear from the text if multiple starting geometries were used and if so, did the optimizations converge to the same or equivalent locations and orientations.

(3) Importantly, it appears that only the Si(OH)₄ is treated at the QM level. Consequently, the critical interactions, such as the H-bonds between Si(OH)₄ and its surroundings, are modeled at the MM level.

(4) The presence of pore water molecules may significantly impact the binding pose and affinity of Si(OH)₄, but they were removed.

As such, what one can learn from these QM/MM calculations are really limited. It will be much better to include critical pore residues and pore water molecules in the QM region, but this can substantially increase the computational costs. If a larger QM region is not feasible, it is advised to remove the current QM/MM calculations from the manuscript.

(Response)

We admit that the previous QM/MM calculation had some problems. We have performed a new QM/MM calculation with silicic acid molecules, pore residues, and water molecules in the QM region based on the reviewer's comment (Fig. #2). In the QM/MM calculation, (1) Only one $\text{Si}(\text{OH})_4$ molecule is presented in the pore in each geometry optimization. (2) We did not use multiple starting geometries. (3, 4) The new QM/MM model of silicic acid molecules was well agreed with the modeled silicic acid molecule (Supplementary Fig. 9). This result indicates that pore water molecules did not significantly affect $\text{Si}(\text{OH})_4$ binding and affinity.

Fig.#2 The QM region (in stick presentation) used in the QM/MM calculation.

Lines 283-284: The authors report the observation of $\text{Si}(\text{OH})_4$ permeating the channel during the 300 ns classical MD simulations. That is interesting. Can the authors provide more quantitative analyses on this? For example, how frequently does such an event occur? On average how long does it take for a $\text{Si}(\text{OH})_4$ to pass through the pore? Does a $\text{Si}(\text{OH})_4$ enter and exit the pore from the same end? What about water?

Note that due to the employed periodic boundary, there is no driving force for Si(OH)₄ to pass through the channel in the model system. Therefore, the simulations should be understood as equilibration (or solvation) of the protein in the membrane-solvent environment rather than the transport of Si(OH)₄ across membrane by Lsi1.

(Response)

We agree with the reviewer that a quantitative analysis may provide further details in Si(OH)₄ permeating the channel. We have carefully inspected the simulation. Two silicic acid molecules permeated the channel from the intracellular to the extracellular side during the simulation. 11.6 ± 2.6 water molecules and 0.7 ± 0.7 Si(OH)₄ molecules existed in the channel region, where we found 16 water molecules in the crystal structure. We have included the description and analysis (Supplementary Fig. 10f) in the revised manuscript.

We have also described the Si(OH)₄ permeation in the text and have included the permeated Si(OH)₄ trajectories and MD snapshots in the revised manuscript (Supplementary Fig. 10, 11). However, the water trajectories did not tell other than that they permeated the channel very frequently.

Line 293: The “calculated population” should be defined.

(Response)

The “calculated population” indicates how often water molecules W3, W9, and W17 occupy the cavities throughout the MD simulation.

We have modified the revised manuscript as below, “We calculated how often these water molecules occupy the cavities throughout the MD simulation. They were 60% for Wat3, 94% for Wat9, and 44% for Wat17.”

Lines 333-338: The displacement of Thr157 by Si(OH)₄ as seen in the MD simulations suggests the flexibility of the Thr157 side chain. However, claiming that Thr157 has a role in Si(OH)₄ selectivity is probably an overstatement in the absence of further supporting data.

(Response)

We agree that the significance of Thr157 may be an overstatement without any supporting data. We have modified the manuscript in lines 335-338 into the following sentences to avoid an overstatement.

“Thr157 has hydrogen bonding interaction with Thr223 and exposes its methyl group to the pore. Moreover, MD simulation suggested that ... displaced (Supplementary Fig. 15e, f). Therefore, Thr157 in Lsi1 is distinct from the AtTIP2;1 structure, and it may not directly contribute to the substrate selectivity.”

The MD simulation should be able to provide more useful information about the pore water, which is vital to the Lsi1 operation. For example, on average how many water molecules are in the pore at a given time? Does Si(OH)₄ completely loss its first-solvation-shell water when entering the channel? How often do the water molecules in the Wat3, Wat9, Wat17 binding sites are replaced (e.g., by other water molecules from the bulk)? Does such water exchange correlate with Si(OH)₄ entering/exiting the pore from either side?

(Response)

Thank you for your comment. As we have already noted above, we further analyzed the MD simulation. On average, 11.6 ± 2.6 water molecules and 0.7 ± 0.7 Si(OH)₄ molecules existed in the channel region (Fig.#3), where we found 16 water molecules in the crystal structure.

Fig. #3 No. of water molecules (blue) and Si(OH)₄ molecules (red) found in the channel region during the simulation. The channel region is defined as $-7.6 \text{ \AA} < \text{pore position} < 10 \text{ \AA}$, and the pore position is shown in Fig. 2d. The number is obtained from the moving average over the 2-ns window.

We have also analyzed the exchange rate of Wat3, Wat9, and Wat17 (Table 2). Their exchange rate suggested that Wat9 bound to the site longer ($\tau = 1.5 \text{ ns}$) than Wat3 and Wat17 ($\tau \sim 0.3 \text{ ns}$). When the Si(OH)₄ moved across the Wat3 and Wat17 sites, Wat3/Wat7 unbound. On the other hand, the Wat9 was independent of the Si(OH)₄ movement and remained binding to the site. We did not find any correlation between the exchange of Wat3, Wat9, and Wat17 and the Si(OH)₄ entering/exiting the pore from either side.

Table 2 Average number n_{ave} and average exchange time τ of water molecules on the site.

	Wat3¹	Wat9²	Wat17³
n_{av}	0.597	0.943	0.437
$\tau \text{ (ns)}^4$	0.30	1.50	0.31

¹ Hydrogen bonded with C=O of Gln84. We defined the hydrogen bond when the O_{Wat}-O distance $< 3.2 \text{ \AA}$.

² Hydrogen bonded with C=O of Gly88.

³ Hydrogen bonded with O of the sidechain of Thr181.

⁴ Calculated by $\tau = T n_{\text{ave}} / N_{\text{wat}}$, where T is the simulation time 450 ns, and N_{wat} is the number of unique water molecules on the site during the simulation.

We have included the new analysis on the MD simulation in the revised manuscript.

Perhaps the authors can simulate the same model system but with glycerol in the solvent and see how glycerol binds into the pore. This should allow direct comparisons between the binding of silicic acid and the binding of glycerol, potentially providing critical tests on the authors' hypothesis on their relative selectivity in Lsi1 (Lines 212-251). Doing so could further improve the manuscript.

(Response)

We agree that an MD simulation with glycerol would provide strong evidence for the substrate specificity of Lsi1 for silicic acid over glycerol. However, we could not perform the simulation due to increased computational costs. We want to investigate it in our future study, and we thank the reviewer's excellent suggestion.

REVIEWERS' COMMENTS

Reviewer #1 (Remarks to the Author):

The authors have submitted a new version of their manuscript in an attempt to address concerns raised in the previous review regarding the Lsi1 function and the lack of clarity regarding the computational analyses of Lsi1 silicic acid transport. The revised manuscript improved the manuscript considerably. While the work falls short of providing a structure of the protein with transport ligand, a novel structure and an interesting model for silicic acid permeability for a plant specific aquaporin family have never-the-less emerged.

However, some of the original criticisms of interpretation and writing still remain. The authors need address these. Also, analyses that they presented in their rebuttal letter, which provide greater clarity, are not included in the revised manuscript. These need to be included in their revised manuscript and explicitly discussed.

1. Description of waters associated with Lsi1cryst

- Line 132-34. In their rebuttal letter, the author's model regarding the numbers of waters in the vestibule, and their role in silicic acid dehydration is addressed. They also provide a MD analysis of silicic acid desolvation observed during their simulation (Fig. #1, pg. 4 of the rebuttal letter). However, this description and this data they do not adequately represented in the revised manuscript. As a result the statement: "The contrasting environments may affect the energetic barrier to pass SF in removing hydrated water molecules..." remains speculative and insubstantial. This description would be improved by providing the information and including the data from their rebuttal letter as an additional supplemental figure. E.g., "This hypothesis is supported by MD simulations of silicic desolvation as shown in Supplemental Fig X...".
- Lines 136 to 143, the significance of the difference in water orientation described here (particularly waters 3 and 9) becomes more apparent on lines 184 to 185. The authors may wish to explicitly note (either in Results or Discussion) that the lower temperature factors of these waters supports their model for their role in transport.
- Other minor revisions: Line 130 and 131, "binds" not "bounds", delete the term "significant". Line 135, the term "noncrystallographic related tetramers" I unclear, please clarify.

2. The functional role of Thr65, and whether it is essential, remains unclear in the revised manuscript. While Thr65 remains highly conserved in NIP III pores, and occupies part of the SF, it is not accurate to conclude that it “constitutes” the SF (line 205) or that it “likely plays an essential role” (line 347). The authors unequivocally show that removal of the hydrogen bonding capacity of the threonine side chain, by replacement with alanine, results in no effect on Lis1 transport activity. This argues the opposite of their conclusion, the conservation of threonine 65 is not required for silicic acid transport. The loss of function mutants result from the substitution of amino acids with bulkier atoms in their side chains. Could it be that the main requirement at this position is steric? Also, from the author’s silicic acid docked model in Fig. 6B, it appears as if Thr65 does not contact substrate, and so it does not appear to “constitute” the SF. The descriptions on line 205 and 347 should be revised or clarified.

3. Lines 209-210, “slightly decrease or increase” are these significant differences or not? What is the proposed role for Thr 181? Is shape or steric properties? Need to describe this more clearly.

4. The focus on the gain of function mutations in GlpF has improved the manuscript. However, the authors need to revise the text on lines 229 to 230, since gain of function mutants were not pursued with AQPM and PfaQP. (i.e., line 230 should simply read “structurally characterized aquaglyceroporin GlpF in an attempt to mimic...”).

5. The authors have tried to revise the manuscript to provide clarity on the methodology and outcome of the QM/MM modeling of silicic acid occupancy of the transport pore. They need to address the following issues in writing and detail.

- The authors must be more precise in their language, and accordingly recognize the limitations of their computational modeling. As noted in the last review, computational modeling is not a “verification” of the transport mechanism for this protein. Rather it provides a test or investigation of one plausible possibility consistent with the structural properties and dynamic modeling of the pore. The term “verified” (i.e., to demonstrate something is a fact) is not accurate or appropriate, and the authors need remove this term from lines 254 and 280 and revise the description.

- In the previous review, questions were raised regarding how the silicic acid molecules were computationally docked or placed in the pore. This is critical since all analyses were done with this starting point. How did the authors decide where to place the silicic acid? While the authors provide a table (Fig. 6e) regarding which waters were chosen for each silicic acid placed in the pore, they do not

indicate how these were chosen and the silicic acid molecules computationally docked. Did they do it by hand? A docking program? What is the source of the ligand file used for silicic acid? The authors need to provide clearer detail in the materials and methods (Lines 546 to 552).

- The most interesting part of this analysis is the proposed docked structure of silicic acid in the SF. In their rebuttal letter, the authors provide an excellent model showing Si₂ bound to the selectivity filter residue residues together with calculated hydrogen bond distances, but this is not included in their revised manuscript. The authors should consider placing it in Fig. 6 as a nice complement to Fig. 6B since it clearly shows the side chain/water/substrate contacts.

- Related to this point, greater detail and clarity regarding the positions of the other silicic acid molecules after QM/MM compared to water should be provided in Supplemental Fig. 9. This is the level of detail needed for their silicic acid model in Supplemental Fig. 9. This is more useful than the tables they provide in Fig. 6e or SFig 9d. A detailed depiction of the final QM/MM simulation structure (with all Si atoms and predicted side chain contacts) compared to the water bound structure may be more informative than the images in SFig 9a and b.

- The superposition of the water chains shown in SFig 9c is difficult to see and interpret since the waters are similar in color and the Figure legend is poorly described and annotated. What does this image show, and what is its significance?

- Lines 265 to 268, this description is unclear. In particular, it is not apparent what the “hollows” represent in this description. Also, it is not clear what the authors are asserting on lines 276 to 278, and this seems speculative.

6. Supplemental Fig 5, legend indicate which water molecule we are looking at. The images need annotation, where is loop C?

7. The description and analysis of the MD simulation is much improved and clearer, particularly the additions to the Results section on 281 to 309. However, Fig. 7 must be updated as well with the new analyses provided by the authors for greater impact in the presentation of the findings:

- Figure 7 in the manuscript is unchanged from the last submission. As noted in the previous review, this figure is difficult to interpret since it does not show time dependent information regarding silicic acid permeation. They do provide the MD trajectory for silicic acid movement in Supplemental Figure 11. The MD trajectory data shown in Fig 11a is much more impactful than the average Si density in Figure 7. The

authors need to include Supplemental Fig. 11A as a panel in Fig. 7. This would provide greater clarity and a time resolved component regarding Si behavior during the simulation.

- To clarify the term “bottleneck”, perhaps clarify in the text as: “where silicic acid densities were low during the simulation and silicic acid cannot move freely”. Also may be informative to place these bottle necks on the simulation pore coordinate in Supplemental Fig. 11a.

- As requested in the previous review, the authors provide an image showing the positions of silicic acid at beginning and end of the simulations (Fig. #2 in rebuttal). For some reason they did not put this in the revised manuscript. This should be included either in Fig. 7 or Supplemental Fig. 10 or 11, and discussed.

8. In their rebuttal letter, the authors have responded to the question regarding why two systems (Xenopus and insect cells) were used for transport analysis, but they have not modified their manuscript accordingly. They should place a short statement in the methods section that the functional results obtained were equivalent or similar in either system.

9. Minor recommended revisions for clarity:

- Abstract Line 32, “composed” instead of “composing.”
- Line 72, remove “the” in: “we failed to obtain the crystals...” in the intro.
- Line 73 “mutant” instead of “mutated”.
- Line 79, list activity as a % of wild type (e.g., 60%).
- Line 81 “of” instead of “by”
- Line 89-90, “restricts the extracellular side” do the authors mean “with a constriction on the extracellular side”?
- Fig 2 panel A and B, change arrow colors to red for better visibility.
- Lines 114-15 Revise to read : “a structure of Lsi1cryst that is distinct from”
- Lines 163-4, revise to read “transport of substrates larger than silicic acid...”
- Lines 168 to 169, the use of “wider” twice in the sentence is unclear. Do they mean that GlpF is wider than AQP1, and that Lsi1 is wider than GlpF? This needs to be rewritten.
- In Fig. 3, put the diameters of the SF (from hole) under each for comparison.
- Lines 144 and lines 170 to 71 The authors may wish to state “A striking feature...” instead of “The striking feature...” (i.e., there are many striking features in their structure!).
- Line 145, do they mean the “extracellular half of the pore”?

- Line 172 should be revised to read “a water molecule”
- Line 173 should be revised to read “a bulky hydrophobic residue”.
- Line 178 “from a nearby...”.
- Line 192-3, recommend “site-directed mutants” instead of variants
- Line 233, delete “the”, Line 236, delete “the” throughout this line.
- Line 261, replace “modeled Si2 can occupy...” with “modeled Si2 occupies...”
- Fig 6, explain what “Ext” refers to in the legend.
- Line 274 recommend “provides orientations” rather than “defines orientations”.
- Line 282 Clarify by stating “during the MD simulation, an average of 11.6 ...”. Also, use the term “occupied” instead of “existed” in this sentence.
- Lines 265 to 268, replace the word “hollow” not sure what they mean.
- Line 288, “that the oxygen...” rather than “where the oxygen...”
- Line 292-3, “Wat3/7 were dissociated from this site” instead of “Wat3/7 unbound”.
- Line 294, “bound” instead of “binding”
- Line 298 “interactions”
- Line 303, it is not clear which cavity is being referred to in the statement: “nearby the cavity”
- Line 305 revise to read “that includes Wat9”.
- Line 329, should read: “therefore does not face the pore.”
- Line 330, “is a water and ammonia permeable AQP...”
- Line 338, “that an additional...”
- Line 339, “contributes to” instead of “constitutes”.
- Line 361, Specify what “them” means in the statement: “between them”.
- Line 371, remove “further”.
- Line 381, “hallow” is not the correct term. “narrows the region of the pore through...”
- Lines 380 to 382, While the authors make a good case for differences between hAQP10 and Lsi1 with respect to their SF, there is no evidence that Lsi1 is more ideal than hAQP10 with respect to Si permeability.
- Line 389, “common” or “shared” instead of “typical”.
- Line 390, delete “the” in “the convergent”.

- Line 398, recommended rewording “while As-transport activity was substantially retained (Fig. 5). Also, “Lsi with a ...”
- Line 400 “Silicic acid is a tetrahedral molecule that forms...”
- Line 404, “tetrahedral” instead of “tetragonal”.

Reviewer #2 (Remarks to the Author):

Overall, the authors have responded adequately to my comment and I am happy with the manuscript content. There are however some minor issues, mainly language related, that I would recommend fixing before publications, see below. I strongly encourage the authors to ask for a final read-through of the manuscript by a native English speaker as there are several mistakes and I may not have caught them all.

Minor revisions:

line 89. "The channel pore exists in each monomer and restricts..." Should be rephrased, I guess the meaning is that the channel narrows on the extracellular side?

line 96: "A few residues unique to the Si permeable AQPs can explain such shifts well." Unclear, rephrase.

line 130 and 131: " change "bounds" to "binds"

lines: 150-155: the section is repetitive and does not explain the proton exclusion mechanism adequately. Please rephrase.

line 258: "is unchanged mainly after the the", I believe it should be "is mainly unchanged after the"

line 275: "...because we hypothesised based on the structure without any substrates" should be "...because we base our hypothesis on the structure..."

line 288: "...a cavity where the oxygen atoms..." should be "...a cavity which the oxygen atoms..."

line 292: "Wat3/Wat7", should it be "Wat3/Wat17"?

line 293: remove the in "the Wat9 was independent"

line 295-296: % of what? Occupancy?

line 303: "Wat3 and Wat9 occupy nearby the cavity" - what do they occupy, positions?

line 305: "included Wat9" should be "including Wat9"

lines 308-309: "highlighting the high-resolution structure's significance in visualizing most water molecules in the channel" - unclear, rephrase. I assume it's the importance of the high-resolution structure's ability to visualize most water molecules that is meant, but the language is not completely correct.

line 330: "AtTIP2;1 is water and ammonia permeable AQPs" - remove AQPs at the end.

line 338 "adapts a unique position" should be "adopts a unique position", "that additional hydrophilic residue... constitutes SF" should be "that an additional hydrophilic residue... constitutes part of SF".

line 353-354: "Thr181 is conserved in the Si-channel" - conserved in NIP III subgroup?

line 361: "The RMSD value between them" - between what?

line 371: "further larger", change to "wider"

line 381: "selectivity for silicic acid" should be "the selectivity for silicic acid"

Reviewer #3 (Remarks to the Author):

The authors addressed most of my concerns. There are only two minor issues remained:

1. The newly added Table 1 in the Supplementary Document: Please explain what a question mark means. Also please briefly explain the “+/-” for Lsi1 in the footnote.
2. The authors provided more details about the QM/MM calculations, and the results in Fig. 9 in the Supplementary Document seem reasonable. Thus, I agree that water may not significantly affect the substrate binding poses. However, I do not think one can claim that the binding affinities are not substantially affected, because there are not binding energies in the comparison.

Reviewer #1 (Remarks to the Author):

The authors have submitted a new version of their manuscript in an attempt to address concerns raised in the previous review regarding the Lsi1 function and the lack of clarity regarding the computational analyses of Lsi1 silicic acid transport. The revised manuscript improved the manuscript considerably. While the work falls short of providing a structure of the protein with transport ligand, a novel structure and an interesting model for silicic acid permeability for a plant specific aquaporin family have never-the-less emerged.

However, some of the original criticisms of interpretation and writing still remain. The authors need address these. Also, analyses that they presented in their rebuttal letter, which provide greater clarity, are not included in the revised manuscript. These need to be included in their revised manuscript and explicitly discussed.

1. Description of waters associated with Lsi1cryst

(Response)

We have added Fig.#1 of the rebuttal letter as Supplementary Fig. 6a,b and modified descriptions in the revised manuscript, based on the reviewer's comment. The term non-crystallographic related tetramers were modified to non-crystallographic symmetry related tetramer.

· Line 132-34. In their rebuttal letter, the author's model regarding the numbers of waters in the vestibule, and their role in silicic acid dehydration is addressed. They also provide a MD analysis of silicic acid desolvation observed during their simulation (Fig. #1, pg. 4 of the rebuttal letter). However, this description and this data they do not adequately represented in the revised manuscript. As a result the statement: "The contrasting environments may affect the energetic barrier to pass SF in removing hydrated water molecules..." remains speculative and insubstantial. This description would be improved by providing the information and including the data from their rebuttal letter as an additional supplemental figure. E.g., "This hypothesis is supported by MD simulations of silicic desolvation as shown in Supplemental Fig X...".

- Lines 136 to 143, the significance of the difference in water orientation described here (particularly waters 3 and 9) becomes more apparent on lines 184 to 185. The authors may wish to explicitly note (either in Results or Discussion) that the lower temperature factors of these waters supports their model for their role in transport.
- Other minor revisions: Line 130 and 131, “binds” not “bounds”, delete the term “significant”. Line 135, the term “noncrystallographic related tetramers” unclear, please clarify.

2. The functional role of Thr65, and whether it is essential, remains unclear in the revised manuscript. While Thr65 remains highly conserved in NIP III pores, and occupies part of the SF, it is not accurate to conclude that it “constitutes” the SF (line 205) or that it “likely plays an essential role” (line 347). The authors unequivocally show that removal of the hydrogen bonding capacity of the threonine side chain, by replacement with alanine, results in no effect on Lis1 transport activity. This argues the opposite of their conclusion, the conservation of threonine 65 is not required for silicic acid transport. The loss of function mutants result from the substitution of amino acids with bulkier atoms in their side chains. Could it be that the main requirement at this position is steric? Also, from the author’s silicic acid docked model in Fig. 6B, it appears as if Thr65 does not contact substrate, and so it does not appear to “constitute” the SF. The descriptions on line 205 and 347 should be revised or clarified.

(Response)

We have modified the description on line 205 as follows, “These results suggested that Thr65_{TM1}, together with Wat9, constitute SF and play a key role in the specificity of transport substrate” Also, we have modified the description on line 347 as follows, “... may compensate for the substrate specificity or interactions with water molecules”

3. Lines 209-210, “slightly decrease or increase” are these significant differences or not? What is the proposed role for Thr 181? Is shape or steric properties? Need to describe this more clearly.

(Response)

As seen in Fig. 5c, there is significant differences in T181S and T181N mutants compared to wild type, but T181V has a similar transport activity with wild type. The role for Thr181 is not key determinant of the substrate transport. However, considering that T181 is conserved within NIPIII silicic acid channel and that wild type showed the highest silicic acid selectivity among the mutants, the size and nature of the Thr residue are most suitable for the high silicic acid selectivity.

4. The focus on the gain of function mutations in GlpF has improved the manuscript. However, the authors need to revise the text on lines 229 to 230, since gain of function mutants were not pursued with AQPM and PfAQP. (i.e., line 230 should simply read “structurally characterized aquaglyceroporin GlpF in an attempt to mimic...”).

(Response)

We have modified the manuscript based on the comment.

5. The authors have tried to revise the manuscript to provide clarity on the methodology and outcome of the QM/MM modeling of silicic acid occupancy of the transport pore. They need to address the following issues in writing and detail.

- The authors must be more precise in their language, and accordingly recognize the limitations of their computational modeling. As noted in the last review, computational modeling is not a “verification” of the transport mechanism for this protein. Rather it provides a test or investigation of one plausible possibility consistent with the structural properties and dynamic modeling of the pore. The term “verified” (i.e., to demonstrate something is a fact) is not accurate or appropriate, and the authors need remove this term from lines 254 and 280 and revise the description.

(Response)

We have changed the term “verified” to “investigated” in the revised manuscript.

· In the previous review, questions were raised regarding how the silicic acid molecules were computationally docked or placed in the pore. This is critical since all analyses were done with this starting point. How did the authors decide where to place the silicic acid? While the authors provide a table (Fig. 6e) regarding which waters were chosen for each silicic acid placed in the pore, they do not indicate how these were chosen and the silicic acid molecules computationally docked. Did they do it by hand? A docking program? What is the source of the ligand file used for silicic acid? The authors need to provide clearer detail in the materials and methods (Lines 546 to 552).

(Response)

We have placed silicic acid molecules downloaded from the Cambridge Structural Database (The Cambridge Crystallographic Data Centre, the deposition number 1406687) by hand so that their oxygen atoms overlapping with water molecules in the crystal structure. This process allowed us to build eight Si molecules (Si1-Si8), and water molecules employed for this modeling were provided in Fig. 6e. We have incorporated this description in the methods section of the revised manuscript.

· The most interesting part of this analysis is the proposed docked structure of silicic acid in the SF. In their rebuttal letter, the authors provide an excellent model showing Si2 bound to the selectivity filter residue residues together with calculated hydrogen bond distances, but this is not included in their revised manuscript. The authors should consider placing it in Fig. 6 as a nice complement to Fig. 6B since it clearly shows the side chain/water/substrate contacts.

(Response)

Since Si2 is a manually docked model, we would not like to show the distances, which may confuse the reader. So we did not include the figure in Fig.6.

· Related to this point, greater detail and clarity regarding the positions of the other silicic acid molecules after QM/MM compared to water should be provided in Supplemental Fig. 9. This is the level of detail needed for their silicic acid model in Supplemental Fig. 9. This is more useful than the tables they provide

in Fig. 6e or SFig 9d. A detailed depiction of the final QM/MM simulation structure (with all Si atoms and predicted side chain contacts) compared to the water bound structure may be more informative than the images in SFig 9a and b.

(Response)

We are not sure which comparison is more useful for the readers. Instead, we have deposited all Si models as supplementary data so that everyone can compare them in detail.

· The superposition of the water chains shown in SFig 9c is difficult to see and interpret since the waters are similar in color and the Figure legend is poorly described and annotated. What does this image show, and what is its significance?

(Response)

There is no water chain in SFig9c. This figure intends to show a comparison of QM/MM calculation and manually modeled silicic acid.

· Lines 265 to 268, this description is unclear. In particular, it is not apparent what the “hollows” represent in this description. Also, it is not clear what the authors are asserting on lines 276 to 278, and this seems speculative.

(Response)

We have modified the manuscript by avoiding term hollows and removed the sentence on lines 276 to 278.

6. Supplemental Fig 5, legend indicate which water molecule we are looking at. The images need annotation, where is loop C?

(Response)

We have added annotation to show loop C in the revised figure.

7. The description and analysis of the MD simulation is much improved and clearer, particularly the additions to the Results section on 281 to 309. However, Fig. 7 must be updated as well with the new analyses provided by the authors for greater impact in the presentation of the findings:

- Figure 7 in the manuscript is unchanged from the last submission. As noted in the previous review, this figure is difficult to interpret since it does not show time dependent information regarding silicic acid permeation. They do provide the MD trajectory for silicic acid movement in Supplemental Figure 11. The MD trajectory data shown in Fig 11a is much more impactful than the average Si density in Figure 7. The authors need to include Supplemental Fig. 11A as a panel in Fig. 7. This would provide greater clarity and a time resolved component regarding Si behavior during the simulation.

- To clarify the term “bottleneck”, perhaps clarify in the text as: “where silicic acid densities were low during the simulation and silicic acid cannot move freely”. Also may be informative to place these bottle necks on the simulation pore coordinate in Supplemental Fig. 11a.

- As requested in the previous review, the authors provide an image showing the positions of silicic acid at beginning and end of the simulations (Fig. #2 in rebuttal). For some reason they did not put this in the revised manuscript. This should be included either in Fig. 7 or Supplemental Fig. 10 or 11, and discussed.

(Response)

We have moved Supplementary Fig. 11A in the previous manuscript to Fig.7. We have also moved Fig. #2 of the rebuttal letter to Supplementary Fig. 11. We have modified the description for the bottleneck in the revised manuscript.

8. In their rebuttal letter, the authors have responded to the question regarding why two systems (Xenopus and insect cells) were used for transport analysis, but they have not modified their manuscript accordingly. They should place a short statement in the methods section that the functional results obtained were equivalent or similar in either system.

(Response)

We have modified the manuscript based on the comment.

9. Minor recommended revisions for clarity:

(Response)

We have modified all issues raised by the reviewer in the revised manuscript.

- Abstract Line 32, “composed” instead of “composing.”
- Line 72, remove “the” in: “we failed to obtain the crystals...” in the intro.
- Line 73 “mutant” instead of “mutated”.
- Line 79, list activity as a % of wild type (e.g., 60%).
- Line 81 “of” instead of “by”
- Line 89-90, “restricts the extracellular side” do the authors mean “with a constriction on the extracellular side”?
- Fig 2 panel A and B, change arrow colors to red for better visibility.
- Lines 114-15 Revise to read : “a structure of Lsi1cryst that is distinct from”
- Lines 163-4, revise to read “transport of substrates larger than silicic acid...”
- Lines 168 to 169, the use of “wider” twice in the sentence is unclear. Do they mean that GlpF is wider than AQP1, and that Lsi1 is wider than GlpF? This needs to be rewritten.
- In Fig. 3, put the diameters of the SF (from hole) under each for comparison.
- Lines 144 and lines 170 to 71 The authors may wish to state “A striking feature...” instead of “The striking feature...” (i.e., there are many striking features in their structure!).
- Line 145, do they mean the “extracellular half of the pore”?
- Line 172 should be revised to read “a water molecule”
- Line 173 should be revised to read “a bulky hydrophobic residue”.
- Line 178 “from a nearby...”.
- Line 192-3, recommend “site-directed mutants” instead of variants
- Line 233, delete “the”, Line 236, delete “the” throughout this line.
- Line 261, replace “modeled Si2 can occupy...” with “modeled Si2 occupies...”
- Fig 6, explain what “Ext” refers to in the legend.
- Line 274 recommend “provides orientations” rather than “defines orientations”.

- Line 282 Clarify by stating “during the MD simulation, an average of 11.6 ...”. Also, use the term “occupied” instead of “existed” in this sentence.
- Lines 265 to 268, replace the word “hollow” not sure what they mean.
- Line 288, “that the oxygen...” rather than “where the oxygen...”
- Line 292-3, “Wat3/7 were dissociated from this site” instead of “Wat3/7 unbound”.
- Line 294, “bound” instead of “binding”
- Line 298 “interactions”
- Line 303, it is not clear which cavity is being referred to in the statement: “nearby the cavity”
- Line 305 revise to read “that includes Wat9”.
- Line 329, should read: “therefore does not face the pore.”
- Line 330, “is a water and ammonia permeable AQP...”
- Line 338, “that an additional...”
- Line 339, “contributes to” instead of “constitutes”.
- Line 361, Specify what “them” means in the statement: “between them”.
- Line 371, remove “further”.
- Line 381, “hallow” is not the correct term. “narrows the region of the pore through...”
- Lines 380 to 382, While the authors make a good case for differences between hAQP10 and Lsi1 with respect to their SF, there is no evidence that Lsi1 is more ideal than hAQP10 with respect to Si permeability.
- Line 389, “common” or “shared” instead of “typical”.
- Line 390, delete “the” in “the convergent”.
- Line 398, recommended rewording “while As-transport activity was substantially retained (Fig. 5). Also, “Lsi with a ...”
- Line 400 “Silicic acid is a tetrahedral molecule that forms...”
- Line 404, “tetrahedral” instead of “tetragonal”.

Reviewer #2 (Remarks to the Author):

Overall, the authors have responded adequately to my comment and I am happy with the manuscript content. There are however some minor issues, mainly language related, that I would recommend fixing before publications, see below. I strongly encourage the authors to ask for a final read-through of the manuscript by a native English speaker as there are several mistakes and I may not have caught them all.

(Response)

We have addressed all issues raised by the reviewer and have asked a native English speaker to check our manuscript. We have incorporated all modifications into the revised manuscript.

Minor revisions:

line 89. "The channel pore exists in each monomer and restricts..." Should be rephrased, I guess the meaning is that the channel narrows on the extracellular side?

line 96: "A few residues unique to the Si permeable AQPs can explain such shifts well." Unclear, rephrase.

line 130 and 131: " change "bounds" to "binds"

lines: 150-155: the section is repetitive and does not explain the proton exclusion mechanism adequately. Please rephrase.

(Response)

We have deleted repetitive descriptions and rephrase and shortened the description as below.

"It has been proposed that strongly correlated movements of the well-oriented single-file water in orthodox AQP family proteins prevent proton transfer via the grothuss mechanism²². Lsi1 must have a different mechanism that prevents

fast proton transport since the water molecules in SF of Lsi1 are not single-file. The breakage of the hydrogen bond interactions between the SF waters and nearby water molecules may prevent proton translocation. “

line 258: "is unchanged mainly after the the", I believe it should be "is mainly unchanged after the"

line 275: "...because we hypothesised based on the structure without any substrates" should be "...because we base our hypothesis on the structure..."

line 288: "...a cavity where the oxygen atoms..." should be "...a cavity which the oxygen atoms..."

line 292: "Wat3/Wat7", should it be "Wat3/Wat17"?

line 293: remove the in "the Wat9 was independent"

line 295-296: % of what? Occupancy?

line 303: "Wat3 and Wat9 occupy nearby the cavity" - what do they occupy, positions?

line 305: "included Wat9" should be "including Wat9"

lines 308-309: "highlighting the high-resolution structure's significance in visualizing most water molecules in the channel" - unclear, rephrase. I assume it's the importance of the high-resolution structure's ability to visualise most water molecules that is meant, but the language is not completely correct.

line 330: "AtTIP2;1 is water and ammonia permeable AQPs" - remove AQPs at the end.

line 338 "adapts a unique position" should be "adopts a unique position", "that additional hydrophilic residue... constitutes SF" should be "that an additional hydrophilic residue... constitutes part of SF".

line 353-354: "Thr181 is conserved in the Si-channel" - conserved in NIP III subgroup?

(Response)

Thr181 is only conserved in NIP III subgroups except for leguminosae.

line 361: "The RMSD value between them" - between what?

line 371: "further larger", change to "wider"

line 381: "selectivity for silicic acid" should be "the selectivity for silicic acid"

Reviewer #3 (Remarks to the Author):

The authors addressed most of my concerns. There are only two minor issues remained:

1. The newly added Table 1 in the Supplementary Document: Please explain what a question mark means. Also please briefly explain the “+/-” for Lsi1 in the footnote.

(Response)

We added the following footnote in the revised manuscript.

“Plus/minus for Lsi1 means Lsi1 transports glycerol at a high concentration of 170 mM but does not transport at a low or physiological concentration of 2 mM. A question mark indicates no study has reported for the permeability.”

2. The authors provided more details about the QM/MM calculations, and the results in Fig. 9 in the Supplementary Document seem reasonable. Thus, I agree that water may not significantly affect the substrate binding poses. However, I do not think one can claim that the binding affinities are not substantially affected, because there are not binding energies in the comparison.

(Response)

We fully agree with the reviewer. Since we do not know the binding energies, we do not claim the binding affinities in this paper.